# On Traceability in $\ell_p$ Stochastic Convex Optimization

**Sasha Voitovych**[*]    **Mahdi Haghifam**[†]    **Idan Attias**[‡]

**Gintare Karolina Dziugaite**[§]    **Roi Livni**[¶]    **Daniel M. Roy**[‖]

## Abstract

In this paper, we investigate the necessity of traceability for accurate learning in stochastic convex optimization (SCO) under $\ell_p$ geometries. Informally, we say a learning algorithm is *m-traceable* if, by analyzing its output, it is possible to identify at least $m$ of its training samples. Our main results uncover a fundamental tradeoff between traceability and excess risk in SCO. For every $p \in [1, \infty)$, we establish the existence of an excess risk threshold below which every sample-efficient learner is traceable with the number of samples which is a *constant fraction* of its training sample. For $p \in [1, 2]$, this threshold coincides with the best excess risk of differentially private (DP) algorithms, i.e., above this threshold, there exist algorithms that are not traceable, which corresponds to a sharp phase transition. For $p \in (2, \infty)$, this threshold instead gives novel lower bounds for DP learning, partially closing an open problem in this setup. En route to establishing these results, we prove a sparse variant of the fingerprinting lemma, which is of independent interest to the community.

## 1 Introduction

Tracing or membership inference informally asks whether it is possible, using only the output of a learning algorithm, to distinguish samples in the training set from held-out samples. The existence of a tracer that identifies training examples reveals that the model has memorized specific examples rather than purely captured the underlying distribution [SSSS17; CCNSTT22]. In particular, understanding tracing has an important role in generalization theory, where an algorithm that is not traceable is known to generalize well beyond its training data [SZ20]. Tracing is also an important technical tool in, e.g., differential privacy (DP), where tracing attacks are the workhorse behind tight lower bounds for the risk-privacy trade-offs [BUV14]. From a privacy standpoint, even the leakage of a single example is viewed as catastrophic. However, from a generalization theory standpoint, we want to understand the exact relationship between an algorithm's generalization performance and the *number* of traceable examples.

To reason rigorously about tracing, following [DSSUV15], we define the problem of tracing as follows. Let $\mathcal{A}_n$ be a learning algorithm that, given a training set $S_n = (Z_1, \dots, Z_n)$ of

---

[*]SV and MH are equal contribution authors.

[*]Institute for Data, Systems, and Society, Massachusetts Institute of Technology

[†]Khoury College of Computer Sciences, Northeastern University

[‡]University of Illinois at Chicago; Toyota Technological Institute at Chicago

[§]Google DeepMind

[¶]Department of Electrical Engineering, Tel Aviv University

[‖]Department of Statistical Sciences, University of Toronto; Vector Institute

39th Conference on Neural Information Processing Systems (NeurIPS 2025).

$n$ i.i.d. samples from some underlying distribution $\mathcal{D}$, outputs a learned model $\hat{\theta}$. Then, a *tracer* $\mathcal{T}$ is a hypothesis tester that, given the model $\hat{\theta}$ and a candidate point $Z$, outputs IN if it believes $Z$ was in $S_n$, or OUT otherwise. Formally, for some small soundness parameter $\xi \in (0, 1)$ and $m \leq n$, we require $\mathcal{T}$ to satisfy:

$$\Pr_{\substack{S_n \sim \mathcal{D}^{\otimes n} \\ Z \sim \mathrm{Unif}(S_n)}} [\mathcal{T}(\hat{\theta}, Z) = \mathrm{IN}] \geq \frac{m}{n} \qquad \Pr_{\substack{S_n \sim \mathcal{D}^{\otimes n} \\ Z \sim \mathcal{D} \perp\!\!\!\perp S_n}} [\mathcal{T}(\hat{\theta}, Z) = \mathrm{OUT}] \geq 1 - \xi$$

When such a tracer exists, we say that $\mathcal{A}_n$ is $(\xi, m)$-traceable. Equivalently, $m$ is the expected number of samples in the training set $S_n$ for which the tracer outputs IN, and we refer to $m$ as *recall*. (See Definitions 2.3 and 2.4.)

A fundamental problem in learning theory is investigating how an algorithm's generalization ability interacts with the information it retains about the training samples (including, in our language, its traceability). The common wisdom is that any information about the training set in a learned model is in tension with generalization [XR17; BMNSY18; SZ20]. On the other hand, non-traceable algorithms, such as *differentially private* algorithms, are often unable to reach optimal excess risk. The central question we study in this paper is: *what is the exact tradeoff between the number of traceable examples and achievable excess risk?*

This question was considered, first, in the context of mean estimation of a $d$-dimensional vector, in the seminal work of [BUV14; DSSUV15]. It was also studied in the context of *Stochastic Convex Optimization* (SCO) [SSSS09] in the work of [ADHLR24]. The work of [DSSUV15] studied the tradeoff between excess risk and the number of traceable examples, when a mechanism publishes an estimate of the mean that is *accurate* in every coordinate (i.e., the output of the algorithm has error of $\alpha$ with respect to $\ell_\infty$ norm to the true mean). At a high level, they showed that for every algorithm that has accuracy better than that achievable by a private algorithm, $\Omega(1/\alpha^2)$ examples are traceable on some hard instance. Notice that for the task of mean estimation in $\ell_\infty$ norm, the statistical sample complexity is $\Theta(\log(d)/\alpha^2)$. Thus, for every algorithm, the preceding result only shows that it is possible to trace out a $1/\log(d)$ fraction of the input samples. In contrast, [ADHLR24] exhibited an SCO problem in $\ell_2$ geometry for which a *constant* fraction of the training samples are traceable. An important open problem, then, is to further explore and understand in which setups we expect a constant fraction of the training sample to be traceable.

In this work, we investigate this question, of traceability, in the fundamental learning setup of *Stochastic Convex Optimization* for general $\ell_p$ geometries. We show that, in this general learning setup, when private learning is not possible, *there is no meaningful gap* between sample complexity and traceability. That is, in every geometry, there exist a hard problem for which every (sample-efficient) algorithm is traceable with a recall which is a constant fraction of its sample size. Due to connections between SCO and mean estimation problems, our results also extend to the latter settings; in particular, we close the $\log(d)$ gap in the setting of [DSSUV15] and show that optimal traceability is *dimension-dependent*.

SCO is an ideal testbed for this problem: (1) as in modern machine learning practices, first-order methods are known to achieve optimal sample-complexity rates in this setting [Fel16; HRS16; AKL21], and (2) within this framework, we can design provable methods that mitigate tracing, such as DP algorithms [CMS11; BST14; BFTG19]. Therefore, by studying the problem of traceability in SCO, we also deepen our understanding of the interaction between privacy risks and sample-optimal learning.

To present our results, we recall the basic setup of SCO. An SCO problem is characterized via a triple $\mathcal{P} = (\mathcal{Z}, \Theta, f)$, where $\mathcal{Z}$ is the data space, $\Theta \subset \mathbb{R}^d$ is the parameter space, which must be convex, and $f : \Theta \times \mathcal{Z} \to \mathbb{R}$ is a loss function such that $f(\cdot, z)$ is convex for all $z \in \mathcal{Z}$. In SCO, data points are drawn from an underlying distribution $\mathcal{D}$ over $\mathcal{Z}$, unknown to the learner. The objective of the learner is to minimize the *expected risk* based on observed samples. Then, a learning algorithm $\mathcal{A}_n : \mathcal{Z}^n \to \Theta$ receives a sample $S_n = (Z_1, \ldots, Z_n)$ of $n$ data points from $\mathcal{Z}^n$ and returns a (perhaps randomized) output in $\Theta$. Then, for $\mathcal{D} \in \mathcal{M}_1(\mathcal{Z})$, expected risk is defined as $F_\mathcal{D}(\theta) := \mathbb{E}_{Z \sim \mathcal{D}}[f(Z, \theta)]$. For an SCO problem to be learnable, one often assumes that the loss function $f$ is *Lipschitz* and the *diameter* of the space $\Theta$ is bounded, both of which can be measured w.r.t. different norms. These bounds govern the behavior of learnability, but they can be measured in different geometries. A

canonical class of SCO problems is induced by the $\ell_p$ norms, in which case we assume that $\Theta$ has bounded $\ell_p$-diameter and $f(\cdot, z)$ is $\ell_p$-Lipschitz, for a fixed $p \in [1, \infty]$.

## 1.1 Contributions

In this paper, we establish a fundamental tradeoff between traceability and excess risk for algorithms in the context of SCO in general geometries. Some settings in which tracing is not possible are already well-understood: in the excess risk regime where DP [DMNS06] is possible, no samples can be traced. Due to this observation, the problem of traceability is only meaningful outside the DP risk regime. More formally, let us define minimax statistical and DP excess risks in $\ell_p$ SCO. Specifically, for a family of $\ell_p$ Lipschitz problems $\mathcal{L}_p^d$, we let

$$\alpha_{\text{stat}}(p, n) = \max_{\mathcal{P} \in \mathcal{L}_p^d} \quad \min_{\mathcal{A}_n} \quad \max_{\mathcal{D}} \left\{ \mathbb{E}\left[F_{\mathcal{D}}(\mathcal{A}_n(S_n))\right] - \inf_{\theta \in \Theta} F_{\mathcal{D}}(\theta) \right\} \qquad \text{and} \qquad (1)$$

$$\alpha_{\text{DP}}(p, n) = \max_{\mathcal{P} \in \mathcal{L}_p^d} \quad \min_{(\varepsilon, \delta)\text{-DP-}\mathcal{A}_n} \quad \max_{\mathcal{D}} \left\{ \mathbb{E}\left[F_{\mathcal{D}}(\mathcal{A}_n(S_n))\right] - \inf_{\theta \in \Theta} F_{\mathcal{D}}(\theta) \right\}, \qquad (2)$$

where, for concreteness, we take $\varepsilon = 0.01$ and $\delta = 1/n^2$ in the above.

**Main Contribution:** We show that every sample-efficient algorithm that achieves an excess risk outside the DP regime (that is, $\alpha = o(\alpha_{\text{DP}})$) is traceable with recall proportional to the number of samples. The precise statement of our main contribution varies based on the geometry of SCO:

**Tracing when $p = 1$.** For the case $p = 1$, we show that any learner whose excess risk is better, by a small polynomial factor, than the best risk attainable by a DP algorithm with constant $\varepsilon$ and $\delta = 1/n^2$ must be traceable. Moreover, we give an essentially optimal lower bound on the number of samples that can be traced. In more detail, we show that there exists an $\ell_1$-SCO problem such that, if an algorithm achieves risk of

$$\alpha \lesssim \frac{\alpha_{\text{DP}}}{d^{0.01} \log^2(n)},$$

then $\Omega(\log(d)/\alpha^2)$ of the training samples can be traced (see Theorem 2.6). We note that the choice of the constant 0.01 above is arbitrary. It is instructive to compare our results to [DSSUV15]. While the settings of mean estimation and SCO are generally different, our lower bound for $\ell_1$ geometry also extends to mean estimation in $\ell_\infty$ norm (see Corollary 2.9 for a formal argument). In both settings, the sample complexity scales like $\log(d)/\alpha^2$, however, [DSSUV15] showed traceability of only $1/\log(d)$ fraction of the samples. On the other hand, in our work we show that there is no meaningful gap between sample complexity and traceability, and every sample-efficient algorithm outside the DP regime must memorize a constant fraction of its sample. Notably, our results also imply that traceability is *dimension-dependent* in this setup.

**Tracing when $p \in (2, \infty)$.** For $\ell_p$ SCO with $p \geq 2$, in Theorem 2.7, we show that for

$$\alpha \lesssim \frac{\underline{\alpha}_{\text{DP}}}{\log(n)},$$

where $\underline{\alpha}_{\text{DP}}$ is set as $\underline{\alpha}_{\text{DP}} = \Theta\left(d/n^2\right)^{1/p}$ we can construct an SCO problem such that, if a learner achieves a risk of $\alpha$, then $\Omega(1/\alpha^p)$ of its samples are traceable. Note that, the non-private sample complexity of learning for $p > 2$ is precisely $\Theta(1/\alpha^p)$ in the relevant parameter regime,[**] i.e., the number of traced out samples is of the order of the sample complexity. Note that, the optimal DP risk in this setup constitutes an open problem, and the quantity $\underline{\alpha}_{\text{DP}}$ above need not be the optimal DP risk in this setting. Nevertheless, this quantity can be shown to be a *lower bound* on the optimal DP risk (in the regime $\varepsilon \in \Theta(1)$). We extend this result to other regimes of $\varepsilon$, and another important contribution of our work is proving such DP lower bounds.

---

[**]We point out that, in general, the sample complexity for $p \geq 2$ scales as $1/n^{1/p} \wedge d^{1/2-1/p}/\sqrt{n}$, however, $\alpha_{\text{stat}} \ll \alpha_{\text{DP}}$ only when $d \gtrsim n$. Thus, the question of traceability is only non-vacuous in the overparameterized regime.

| $p$ | Recall | Range of $\alpha$ | Sample complexity | Minimax DP rate | Refs. |
|---|---|---|---|---|---|
| $1$ | $\frac{\log(d)}{\alpha^2}$ | $\left( \sqrt{\frac{\log(d)}{n}},\ \frac{d^{0.49}}{n\sqrt{\log(1/\xi)}} \right)$ | $\frac{\log(d)}{\alpha^2}$ | $\sqrt{\frac{\log(d)}{n}} + \frac{\sqrt{d}}{\varepsilon n}$ | Thm. 2.6 |
| $(1,2]$ | $\frac{1}{\alpha^2}$ | $\left( \sqrt{\frac{1}{n}},\ \frac{\sqrt{d}}{n\sqrt{\log(1/\xi)}} \right)$ | $\frac{1}{\alpha^2}$ | $\frac{1}{\sqrt{n}} + \frac{\sqrt{d}}{\varepsilon n}$ | Thm. 2.5 |
| $[2,\infty)$ | $\frac{1}{\alpha^p}$ | $\left( \min\left\{ \frac{1}{n^{1/p}}, \frac{d^{1/2-1/p}}{\sqrt{n}} \right\},\ \left( \frac{d}{n^2\log(1/\xi)} \right)^{1/p} \right)$ | $\frac{1}{\alpha^p}$ † | Open | Thm. 2.7 |

**Table 1:** Summary of traceability results. All results are stated up to constants. The sample complexity bounds are implied by Theorem A.1. Minimax DP rates are known due to [BFTG19; BGN21; AFKT21; GLLST23] and are displayed up to log factors and with $\delta = 1/n^2$. (†) Although, in general, the sample complexity in this setting is a minimum of two terms, within the stated range of $\alpha$, the term $1/\alpha^p$ dominates.

In particular, we provide an improved lower bound on DP-SCO under $\ell_p$ geometries for $p > 2$ in the high dimensional regime, i.e., $d \geq \varepsilon n$, which is arguably the most interesting regime as it is more relevant for the modern ML applications. Specifically, we show, in Theorem 2.8, that for all $\varepsilon < 1$ and small $\delta$ we can construct a problem such that for every $(\varepsilon, \delta)$-DP algorithm, $\mathcal{A}_n$, there exists a data distribution such that:

$$\mathbb{E}_{S_n \sim \mathcal{D}^{\otimes n}, \hat{\theta} \sim \mathcal{A}_n(S_n)} \left[ F_{\mathcal{D}}(\hat{\theta}) \right] - \inf_{\theta \in \Theta} F_{\mathcal{D}}(\theta) \gtrsim \left( \frac{d}{n^2 \varepsilon^2} \right)^{1/p}.$$

In particular, the above implies that when $d \geq \varepsilon n$, the risk due to privacy dominates the statistical risk. This result improves upon all previous best bounds in the literature when $d \geq \varepsilon n$ [ABGMU22; LLL24]. In particular, Theorem 3.1 of [LLL24] gives a lower bound $\sqrt{d/n^2\varepsilon^2}$, which is weaker than our lower bound for every $p > 2$. Corollary 4 of [ABGMU22] gives a lower bound of $\min\left\{ \left( \frac{1}{\varepsilon n} \right)^{\frac{1}{p}}, \frac{d^{1-1/p}}{\varepsilon n} \right\}$ which is weaker than our lower bound for $d \geq \varepsilon n$.

**Tracing when $p \in (1,2]$.** For each $p \in (1,2)$, we show that there exists an $\ell_p$ SCO problem such that, if an algorithm achieves excess risk of $\alpha \lesssim \frac{\alpha_{\mathrm{DP}}}{\log^2(n)}$, then $\Omega(1/\alpha^2)$ of its samples can be traced (see Theorem 2.5). This result uncovers a fundamental dichotomy between traceability and privacy in $\ell_p$ SCO. It is known that $p \in (1,2]$, $\Theta(1/\alpha^2)$ is precisely the sample complexity of learning $\ell_p$-Lipschitz problems [AWBR09]. We note that $\alpha_{\mathrm{DP}}$ for $p = 2$ is known due to [BST14]. However, as we discuss in Appendix B.1, combining [BST14] with the tracing results of [DSSUV15] does not give the optimal tracing of $\Theta(1/\alpha^2)$ samples.

### 1.1.1 Traceability beyond SCO: PAC Learning

A natural question is whether a similar phenomenon holds true for other learning setups. Consider the setting of binary classification PAC learning. We show that, for every class with VC dimension bounded by $d_{\mathsf{vc}}$, the recall of every tracer is in $O(d_{\mathsf{vc}} \log^2(n))$, i.e., it is at most a small fraction of the training sample provided $n \gg d_{\mathsf{vc}}$. Since many such classes, including the class of thresholds, are not privately learnable [BNSV15; ALMM19; BLM20], the sharp transitions between privacy and traceability does not hold in PAC classification. We also point out that for the class of thresholds, we can remove the $\log^2(n)$ factor from the recall upper bound. See Appendix H.

### 1.2 Technical contributions

Our technical contributions are elaborated on in Section 2.3. In essence, our technical novelties are twofold. First, we present a novel *sparse fingerprinting lemma* that, intuitively, shows that learners over sparse domains must be correlated to their samples. The key novelty of this result is that the *correlation is inversely proportional to the sparsity parameter*. This feature is not present in prior work, since fingerprinting lemmas are most often applied for learners/estimators over a hypercube domain.

Second, armed with this new fingerprinting result, we present a generic conversion result using a notion of a *subgaussian trace value*, which converts any lower bound on *correlation*

with the samples into a *number* of samples that can be traced. While it is well-appreciated by prior work that, conceptually, a fingerprinting lower bound implies a traceability lower bound, proving results for our setting of SCO involves complicated sparse domains embedded into $\ell_p$ balls. This makes it more technically challenging to prove the necessary concentration phenomena holds for a tracer over the corresponding domain, which motivates us to restrict our attention to tracers that induce a *subgaussian process* over the domain.

## 1.3 Related Work

Our work is most similar in spirit to [DSSUV15; ADHLR24]. Our work builds on top of these results on a number of fronts. A key distinct aspect of our approach is the difference in the structure of hard problems and the new sparse fingerprinting lemma. Also, our generic traceability theory of *subgussian trace value* (Section 2.3) provides an abstract treatment of the approach in [DSSUV15]. Our approach allows to seamlessly convert fingerprinting lemmas into traceability results and even non-private sample complexity lower bound.

Our work also makes progress towards closing the gap regarding the optimal excess error for $\ell_p$ DP-SCO for $p > 2$. The best known upper bounds for DP-SCO in $\ell_p$ geometry for $p > 2$ are due to [BGN21; GLLST23], and Theorem 2.8 is the best lower bound.

To put our *sparse fingerprinting lemma* into the context of prior work, it can be seen to generalize the results of [SU17] to sparse sets. Another "sparse fingerprinting lemma" in the literature is given by [CWZ23]. Our results are distinct by the way sparsity enters the lemmas: in [CWZ23] the *mean vector* is sparse (and data is dense), and in our case, the mean is dense and the *data vectors* are sparse. The proof techniques also differ substantially. Our sparse fingerprinting lemma is also an example of a fingerprinting lemma for the setting where the coordinates of the data vector are not independent, similar to [KMS22; LT24]. Additional related work is discussed in Appendix B.

## 2 Problem Setup and Main Results

We begin by some definitions. For a (measurable) space $\mathcal{R}$, $\mathcal{M}_1(\mathcal{R})$ denotes the set of all probability measures on $\mathcal{R}$. In SCO, an $\alpha$-learner is defined to be a learner whose expected *excess risk* is bounded by $\alpha$. A formal definition is given below.

**Definition 2.1** ($\alpha$-learner)**.** Fix $\alpha > 0$, $n \in \mathbb{N}$ and SCO problem $(\Theta, \mathcal{Z}, f)$. We say $\mathcal{A}_n : \mathcal{Z}^n \to \mathcal{M}_1(\Theta)$ is an $\alpha$-learner for $(\Theta, \mathcal{Z}, f)$ iff for every $\mathcal{D} \in \mathcal{M}_1(\mathcal{Z})$, we have $\mathbb{E}_{S_n \sim \mathcal{D}^{\otimes n}, \hat{\theta} \sim \mathcal{A}_n(S_n)} \left[ F_{\mathcal{D}}(\hat{\theta}) \right] - \inf_{\theta \in \Theta} F_{\mathcal{D}}(\theta) \leq \alpha$.

In our work, we focus on learning *Lipschitz-bounded* families of problems, which are defined below. For every $p \in [1, \infty]$, let $\mathcal{B}_p(r) = \{\theta \in \mathbb{R}^d : \|\theta\|_p \leq r\}$ be the unit ball in $\ell_p$ norm.

**Definition 2.2** (Lipschitz-bounded problems)**.** Fix $p \in [1, \infty]$, and let $d < \infty$ be a natural number. We let $\mathcal{L}_p^d$ denote the set of all $\ell_p$-Lipschitz-bounded SCO problems in $d$ dimensions. Namely, $\mathcal{P} = (\Theta, \mathcal{Z}, f) \in \mathcal{L}_p^d$ iff (i) $\Theta \subset \mathcal{B}_p(1)$, and (ii) for every $\theta_1, \theta_2 \in \Theta$ and $z \in \mathcal{Z}$, we have $|f(z, \theta_1) - f(z, \theta_2)| \leq \|\theta_1 - \theta_2\|_p$.

### 2.1 Tracing

The key notion we study here is *tracing*, and we next introduce our framework for traceability. We consider families of tracers that assign each candidate point a real-valued score capturing how likely it is to have been seen during training. Then, the tracer converts these scores into binary IN or OUT decisions by thresholding the score. Intuitively the score corresponds to the likelihood of the event that the learner saw a data point during training.

**Definition 2.3** (Tracer)**.** Fix data space $\mathcal{Z}$ and parameter space $\Theta$. A tracer's strategy is a tuple of $\mathcal{T} = (\phi, \mathcal{D})$ where $\phi : \Theta \times \mathcal{Z} \to \mathbb{R}$ and $\mathcal{D} \in \mathcal{M}_1(\mathcal{Z})$.

**Definition 2.4** (($\xi, m$)-traceability)**.** Let $n \in \mathbb{N}$, $\xi \in (0, 1)$, and $m \in \mathbb{N}$. We say a learning algorithm $\mathcal{A}_n$ is ($\xi, m$)-traceable if there exists a tracer $(\phi, \mathcal{D})$ and $\lambda \in \mathbb{R}$ such that, if $(Z_0, Z_1, \ldots, Z_n) \sim \mathcal{D}^{\otimes(n+1)}$ and $\hat{\theta} \sim \mathcal{A}_n(Z_1, \ldots, Z_n)$, we have (i) Soundness: $\Pr\left(\phi(\hat{\theta}, Z_0) \geq \lambda\right) \leq \xi$, and (ii) Recall: $\mathbb{E}\left[\left|\{i \in [n] : \phi(\hat{\theta}, Z_i) \geq \lambda\}\right|\right] \geq m$.

## 2.2 Main Results

### 2.2.1 Traceability of $\alpha$-Learners

In this section, we discuss our traceability results for accurate learners in $\ell_p$ geometries. First, we will state a result that applies to $p \in [1, 2)$, and then present its slight refinement for $p = 1$. We will then present our result for $p \geq 2$. See Appendices F.1 to F.3 for proofs.

**Theorem 2.5.** *There exists a universal constant $c > 0$ such that, for all $p \in [1, 2)$, if $d$, $n$, $\xi \in (0, 1/e)$, and $\alpha > 0$ are such that*

$$\frac{c}{\sqrt{n}} \leq \alpha \leq \min\left\{ c \cdot \sqrt{\frac{d}{n^2 \log(1/\xi)}}, \frac{1}{6} \right\}, \tag{3}$$

*then there exist an $\ell_p$ SCO problem that every $\alpha$-learner is $(\xi, m)$-traceable with $m \in \Omega\left(\alpha^{-2}\right)$.*

Note that the upper bound on $\alpha$ in Equation (3) is precisely the optimal DP excess risk for $\varepsilon \in \Theta(1)$ and $p \in [1, 2]$ [AFKT21; BGN21], and the lower bound is precisely the optimal non-private risk (except $p = 1$; see Theorem A.1). Moreover, for $p \in (1, 2]$, the lower bound on $m$ *exactly* matches the statistical sample complexity.

As mentioned above, for $p = 1$, the lower bound on recall in Theorem 2.5 is less than sample complexity by a factor of $\log(d)$. This prompts us to establish the following refinement

**Theorem 2.6.** *There exists a universal constant $c > 0$ such that, if $d$ is large enough and $n$, $\xi \in (0, 1/e)$, and $\alpha > 0$ are such that*

$$c \cdot \sqrt{\frac{\log(d)}{n}} \leq \alpha \leq \min\left\{ c \cdot \frac{d^{0.49}}{n\sqrt{\log(1/\xi)}}, \frac{1}{8} \right\}, then \tag{4}$$

*there exists a $\ell_1$ SCO problem that every $\alpha$-learner is $(\xi, m)$-traceable with $m \in \Omega\left(\log(d)/\alpha^2\right)$.*

Note that the upper bound in Equation (4) is slightly stronger than in Equation (3); however, the lower bound on recall now matches the sample complexity of learning in $\ell_1$ geometry. We now present a result for $p \geq 2$.

**Theorem 2.7.** *There exists a universal constant $c > 0$ such that, for all $p \in [2, \infty)$, if $d$, $n$, $\xi \in (0, 1/e)$, and $\alpha > 0$ are such that*

$$\frac{1}{6} \cdot \min\left\{ \frac{1}{n^{1/p}}, \frac{d^{\frac{1}{2} - \frac{1}{p}}}{\sqrt{n}} \right\} \leq \alpha \leq \min\left\{ c \cdot \left(\frac{d}{n^2 \log(1/\xi)}\right)^{1/p}, \frac{1}{6} \right\}, then \tag{5}$$

*there exist an $\ell_p$ SCO problem such that every $\alpha$-learner is $(\xi, m)$-traceable with $m \in \Omega\left(1/(6\alpha)^p\right)$.*

For $p \in (2, \infty)$, our results have a different implication, showing that all sufficiently accurate learners need to memorize a number of samples on the order of the sample complexity. However, in this case, the upper bound in Equation (5) need not be the optimal DP risk. Instead, it provides a *lower bound* on the optimal DP risk, as we will see next.

### 2.2.2 Improved DP-SCO Lower Bound for $p > 2$

**Theorem 2.8.** *Let $p \in [2, \infty)$. There exist a universal constant $c > 0$ and an $\ell_p$ SCO problem $\mathcal{P} = (\Theta, \mathcal{Z}, f)$ such that every $(\varepsilon, \delta)$-DP learner of $\mathcal{P}$ with $\varepsilon \leq 1$ and $\delta \leq c/n$ satisfies,*

$$\alpha \geq c \cdot \min\left\{ \left(\frac{d}{\varepsilon^2 n^2}\right)^{\frac{1}{p}}, \frac{d^{1-1/p}}{\varepsilon n}, 1 \right\}.$$

### 2.2.3 Consequences for mean estimation

Consider the setting of mean estimation in $\ell_\infty$ norm as in [DSSUV15]. Our results in Theorem 2.6 extend almost verbatim to this setting.

**Corollary 2.9.** *Let $\mathcal{Z} = \{\pm 1\}^d$, and suppose an estimator is given such that, given access to i.i.d. samples $Z_1, \ldots, Z_n \in \mathcal{Z}$, outputs $\hat{\mu}$ with $\mathbb{E} \|\hat{\mu} - \mathbb{E}[Z_1]\|_\infty \leq \alpha/2$. Then, there exists a universal constant $c > 0$ such that, if $d$ is large enough and $n, \xi \in (0, 1/e)$, and $\alpha > 0$ satisfy Equation (4), then the estimator $\hat{\mu}$ is $(\xi, m)$-traceable with $m \in \Omega\left(\log(d)/\alpha^2\right)$.*

## 2.3 Roadmap of the proof

Our proofs rely on introducing two key technical elements that allow us to generalize tracing techniques to general $\ell_p$ setups. The first element is a generic conversion result involving a complexity notion which we term *the subgaussian trace value* of a problem. As we show in the proof of Theorem 2.8, we can use the subgaussian trace value to prove traceability results over general domains, establish DP sample complexity lower bounds, and, even to recover *non-private* sample complexity lower bounds. While the connection between the first two aspects is well-known [FS17], we find the ability of trace value to recover *non-private* lower bounds surprising.

Our second technical contribution concerns techniques for lower bounding the subgaussian trace value, which we accomplish through several novel *fingerprinting lemmas*. Previous works used the standard fingerprinting lemma, where the learner observes points on a hypercube, to lower bound DP and traceability in $\ell_2$ geometry. However, when moving to general $\ell_p$ geometries, this setup no longer captures the hardest settings to learn. For instance, for $p > 2$, canonical instances of hard problems involve data drawn from sparse sets [AWBR09]. We thus prove new fingerprinting lemmas that enable us to leverage our framework in such settings. These fingerprinting lemmas are then applied to carefully constructed instances of hard problems, and we show that every accurate learner of these problems is traceable.

## 3 General framework: subgaussian trace value

We next describe more formally the framework of subgaussian tracers. For a random variable $X$, the subgaussian norm of $X$ is the quantity $\|X\|_{\psi_2} := \inf\{t : \mathbb{E}\left[\exp(X^2/t^2)\right] \leq 2\}$ [Ver18]. We use the following definition of a subgaussian process:

**Definition 3.1** (Subgaussian process)**.** We call an indexed collection of random variables $\{X_\theta\}$ a $\sigma$-subgaussian process w.r.t a metric space $(\Theta, \|\cdot\|)$ if for every $\theta, \theta' \in \Theta$, we have (i) $\|X_\theta - X_{\theta'}\|_{\psi_2} \leq \sigma \|\theta - \theta'\|$, and (ii) $\|X_\theta\|_{\psi_2} \leq \sigma \operatorname{diam}_{\|\cdot\|}(\Theta)$.

For origin symmetric convex body $\Theta$, let $\|\cdot\|_\Theta$ denote the Minkowski norm w.r.t. $\Theta$, that is $\|x\|_\Theta := \inf\{\lambda > 0 : x \in \lambda\Theta\}$. If $\Theta$ is not convex or not origin symmetric, we let $\|\cdot\|_\Theta$ be the Minkowski norm w.r.t. convex hull of $(\Theta \cup -\Theta)$. Note that $\|\cdot\|_\Theta$ is the minimal norm to contain $\Theta$ in its unit ball.

**Definition 3.2** (Subgaussian tracer)**.** Fix $\kappa \in \mathbb{R}$ to be a constant, and let $\Theta$ be a convex body. We let $\mathfrak{T}_\kappa$ be the class of subgaussian tracers at scale $\kappa > 0$, that is, a tracer $(\phi, \mathcal{D}) \in \mathfrak{T}_\kappa$ iff

(i) $\{\phi(\theta, Z)\}_{\theta \in \Theta}$ where $Z \sim \mathcal{D}$ is a 1-subgaussian process w.r.t. $(\Theta, \|\cdot\|_\Theta)$.

(ii) $|\phi(\theta, z)| \leq \kappa$ for all $\theta \in \Theta$ and $z \in \mathcal{Z}$.

**Definition 3.3** (Subgaussian trace value)**.** Fix $n \in \mathbb{N}$, $\alpha \in [0, 1]$, and $\kappa \in \mathbb{R}$. Consider an arbitrary SCO problem $\mathcal{P} = (\Theta, \mathcal{Z}, f)$. Let $\mathfrak{T}_\kappa$ be as in Definition 3.2. Then, we define the subgaussian trace value of problem $\mathcal{P}$ by

$$\operatorname{Tr}_\kappa(\mathcal{P}; n, \alpha) = \inf_{\alpha\text{-learner} \mathcal{A}_n} \sup_{\mathcal{T} = (\phi, \mathcal{D}) \in \mathfrak{T}_\kappa} \mathbb{E}_{S_n = (Z_1, \ldots, Z_n) \sim \mathcal{D}^{\otimes n}, \hat{\theta} \sim \mathcal{A}_n(S_n)} \left[\frac{1}{n} \sum_{i \in [n]} \phi(\hat{\theta}, Z_i)\right].$$

where the inf is taken over all $\mathcal{A}_n$ that achieve excess risk $\leq \alpha$ on $\mathcal{P}$ with $n$ samples.

**Traceability via subgaussian trace value.** The subgaussian trace value characterizes the average score the pair $(\phi, \mathcal{D})$ assigns to the data points in the training set. However, the definition of recall in Definition 2.3 requires characterizing the *number* of samples in the training set that takes a large value. The former can be converted into the latter, provided the sum of squared scores of samples is not too large. A formal statement, which is a

consequence of Paley–Zygmund inequality, can be found in Lemma A.11. In the next lemma, we show how to control the sum of squares of the $\phi(\hat{\theta}, Z_i)$ using the subgaussian assumption.

**Lemma 3.4.** *Fix $n, d \in \mathbb{N}$. Suppose $\Theta \subset \mathbb{R}^d$ is a subset of a unit ball in some norm $\|\cdot\|$. Let $\phi \colon \Theta \times \mathcal{Z} \to \mathbb{R}$ and $\mathcal{D} \in \mathcal{M}_1(\mathcal{Z})$ be such that, as $Z \sim \mathcal{D}$, $\{\phi(\theta, Z)\}$ is a $\sigma$-subgaussian process w.r.t. $(\Theta, \|\cdot\|)$. Let $(Z_1, \ldots, Z_n) \sim \mathcal{D}^{\otimes n}$. Then, there is a constant $C > 0$, such that*

$$\Pr\left[\sup_{\theta \in \Theta} \sqrt{\sum_{i=1}^{n} [\phi(\theta, Z_i)]^2} \leq C\sigma\left(\sqrt{n} + \sqrt{d} + t\right)\right] \geq 1 - 4\exp(-t^2), \quad \forall t \geq 0.$$

Equipped with this lemma, in the next theorem, we show that if, the subgaussian trace value of a problem is large, then every $\alpha$-learner is traceable.

**Theorem 3.5.** *Fix $n \in \mathbb{N}$, $d \in \mathbb{N}$, $\kappa > 0$ and $\alpha \in [0, 1]$. Consider an arbitrary SCO problem $\mathcal{P} = (\Theta, \mathcal{Z}, f)$. Let $T = \mathrm{Tr}_\kappa(\mathcal{P}; n, \alpha)$ be the subgaussian trace value of $\mathcal{P}$. Then, for some constant $c > 0$, every $\alpha$-learner $\mathcal{A}_n$ is $(\xi, m)$-traceable with*

$$\xi = \exp(-cT^2), \quad m = c\left[\frac{n^2 T^2}{n+d} - \frac{16\kappa^2 n}{\exp(n+d)}\right].$$

**Privacy lower bounds via subgaussian trace value.** In the next theorem, we show that the notion of subgaussian trace value directly lower bounds the best privacy parameters achievable by a DP algorithm. The proof is based on [FS17].

**Theorem 3.6.** *There exists a universal constant $c > 0$, such that the following holds. Fix $p \in [1, \infty)$, $n \in \mathbb{N}$, $d \in \mathbb{N}$, $\alpha \in [0, 1]$, $\kappa > 0$ $\varepsilon > 0$, and $\delta \in [0, 1]$. Consider an arbitrary SCO problem $\mathcal{P} = (\Theta, \mathcal{Z}, f)$ in $\mathbb{R}^d$. Let $T = \mathrm{Tr}_\kappa(\mathcal{P}; n, \alpha)$ be the subgaussian trace value of problem $\mathcal{P}$. Then, for every $(\varepsilon, \delta)$-DP $\alpha$-learner $\mathcal{A}_n$, we have $\exp(\varepsilon) - 1 \geq c\left(T - 2\delta\kappa\right)$.*

**Non-private sample complexity via subgaussian trace value.** Surprisingly, if we directly use *subgaussian trace value*, we can recover optimal sample complexity bounds *for all $p \in [1, \infty)$ and all regimes of $(d, \alpha)$*, thus unifying traceability with private and non-private sample complexity lower bounds. While we detail the argument formally in Appendix G, we consider here a helpful example of $\ell_2$ geometry. First, it can be shown that we always have $\mathrm{Tr}(\mathcal{P}; n, \alpha) \lesssim \sqrt{d/n}$ for arbitrary problem $\mathcal{P}$ (see Proposition G.1). Also, we will later show that, for every $\alpha > 0$, there exist an $\ell_2$ problem $\mathcal{P}$ with $\mathrm{Tr}(\mathcal{P}; n, \alpha) \gtrsim \sqrt{d}/n\alpha$ (see Theorem 5.1). Combining these two inequalities gives $n \gtrsim 1/\alpha^2$, which is optimal.

## 4 The sparse fingerprinting lemma

By introducing the notion of subgaussian trace value, we have reduced the problems of traceability and privacy lower bounds to the question of lower bounding the subgaussian trace value. Now, we discuss the techniques to lower bound subgaussian trace value. The proofs can be found in Appendix D. Due to space limitation, we only discuss the details for the case of $p > 1$ and present the details of $p = 1$ in Appendix E.2.

For $\ell_2$ geometry, one can lower bound subgaussian trace value using the classical fingerprinting lemma in [DSSUV15]. While this strategy leads to traceability results in $\ell_2$ geometry, examples of hard problems for $\ell_p$ geometry with $p > 2$ are those with *sparse* sets $\mathcal{Z}$ (e.g., as in [AWBR09]). This motivates us to prove the following *sparse fingerprinting lemma*, which is another important contribution of our work. For a vector $x \in \mathbb{R}^d$, let $\mathrm{supp}(x)$ be the set of its non-zero coordinates and denote $\|x\|_0 = |\mathrm{supp}(x)|$.

**Definition 4.1** (Sparse distributions family). Fix $d \in \mathbb{N}$, $k \in [d]$ and $\mu \in [-k/d, k/d]^d$. Consider the mixture distribution on $\mathcal{Z}_k = \{z \in \{0, \pm 1\}^d : \|z\|_0 = k\}$ given by, for all $z \in \mathcal{Z}_k$, $\mathcal{D}_{\mu,k}(z) = \mathbb{E}_{J \sim \mathsf{unif}(\binom{[d]}{k})}[P_{\mu,k,J}(z)]$, where

$$P_{\mu,k,J}(z) = \mathbb{1}(\mathrm{supp}(z) = J) \cdot \prod_{j \in J}\left(\frac{1 + (d/k) \cdot \mu^j z^j}{2}\right).$$

Note that, in particular, $\mathbb{E}_{Z \sim \mathcal{D}_{\mu,k}}[Z] = \mu$. Intuitively, one can think of *sampling* from $\mathcal{D}_{\mu,k}$ using the following procedure: (i) sample the support coordinates $J \sim \mathsf{unif}(\binom{[d]}{k})$, (ii) for each $j \in J$, sample $Z^j$ from $\{\pm 1\}$ with mean $\frac{d}{k}\mu^j$ independently, (iii) for each $j \notin J$, set $Z^j = 0$.

With this distribution family at hand, we may state the sparse fingerprinting lemma. For $x, y \in \mathbb{R}^d$ and a subset $R \subseteq [d]$ of coordinates, we use $\langle \cdot, \cdot \rangle_S$ to denote the inner product $\langle x, y \rangle_R := \sum_{i \in R} x_i y_i$. Also, for $\alpha, \beta, \gamma > 0$, let $\mathsf{s\text{-}beta}_{[-\gamma, \gamma]}(\alpha, \beta)$ be the *symmetric* beta-distribution, i.e., beta distribution with parameters $\alpha, \beta$ scaled and shifted to have support $[-\gamma, \gamma]$ (see Definition A.13).

**Lemma 4.2** (Sparse fingerprinting). *Fix $d, n \in \mathbb{N}$ and let $k \in [d]$. For each $\mu \in [-k/d, k/d]^d$, let $\mathcal{Z}_k$ and $\mathcal{D}_{\mu,k}$ be as in Definition 4.1. Let $\pi = \mathsf{s\text{-}beta}_{[-k/d,k/d]}(\beta, \beta)^{\otimes d}$ be a prior and set*

$$\phi_\mu(\theta, Z) := \left\langle \theta, \left( Z - \frac{d}{k}\mu \right) \right\rangle_{\mathrm{supp}(Z)}.$$

*Then, for every learning algorithm $\mathcal{A}_n : \mathcal{Z}^n \to \mathcal{M}_1(\mathbb{R}^d)$ with sample $S_n = (Z_1, \ldots, Z_n)$,*

$$\mathbb{E}_{\mu \sim \pi} \mathbb{E}_{S_n \sim \mathcal{D}_{\mu,k}^{\otimes n}, \hat{\theta} \sim \mathcal{A}_n(S_n)} \left[ \sum_{i=1}^n \phi_\mu(\hat{\theta}, Z_i) \right] = \frac{2\beta d}{k} \mathbb{E}_{\mu \sim \pi} \left\langle \mu, \mathbb{E}_{S_n \sim \mathcal{D}_{\mu,k}^{\otimes n}, \hat{\theta} \sim \mathcal{A}_n(S_n)}[\hat{\theta}] \right\rangle.$$

The key novelty of this lemma is that it provides a way to study the correlation between a learner's output and training samples on sparse sets $\mathcal{Z}_k$. An important and distinctive feature of this result is that the right-hand side scales by a factor of $d/k$, highlighting the fact that sparse problems correspond to greater subgaussian trace values. Intuitively, this stems from the fact that each coordinate is *seen* fewer times by the learning algorithm, meaning it must retain more information from each training sample in order to learn accurately. Additionally, for the special case $k = d$, the result precisely recovers the fingerprinting lemma from [SU17].

# 5 Final steps: bounding the subgaussian trace value for hard problems

Finally, we go over the construction of hard problems. To illustrate the difficulty of problem constructions, we give an example of a problem that requires many samples to learn but nevertheless is not traceable. Consider learning over $\ell_1$ ball with linear loss. Let

$$\Theta = \mathcal{B}_1(1), \quad \mathcal{Z} = \{\pm 1\}^d, \quad f(\theta, Z) = -\langle \theta, Z \rangle. \tag{6}$$

Consider a difficult set of distributions $\{\mathcal{D}_i\}_{i=1}^d$ where $\mathcal{D}_i$ is a product distribution on $\mathcal{Z}$ and has mean $\alpha$ on coordinate $i$ and mean zero on all other coordinates. It can be shown this problem requires $\Theta(\log(d)/\alpha^2)$ samples to learn up to risk of $\alpha/3$, and ERM is an optimal learner. However, after seeing $\Theta(\log(d)/\alpha^2)$ samples from $\mathcal{D}_i$, the ERM takes the value $\hat{\theta} = e_i$ w.h.p., which is also the *population* risk minimizer. In other words, it becomes impossible to trace out any specific samples on which $\hat{\theta}$ was trained.

**Generic construction for $p \in (1, \infty)$.** As mentioned above, to obtain optimal results for $p > 2$, problems constructed need to be sparse, and the main subtlety in our constructions is choosing the sparsity parameter. For some $k \in [d]$ to be chosen later, consider the following $\ell_p$-Lipschitz problem $\mathcal{P}_{k,p}$.

$$\Theta = \mathcal{B}_\infty(d^{-1/p}), \quad \mathcal{Z} = \{z \in \{0, \pm 1\}^d : \|z\|_0 = k\}, \quad f(\theta, z) = -k^{-1/q}\langle \theta, z \rangle. \tag{7}$$

Here, the parameter space $\Theta$ is the largest $\ell_\infty$ ball inscribed into the unit $\ell_p$ ball, and $q$ is the Hölder conjugate of $p$, i.e., $\frac{1}{p} + \frac{1}{q} = 1$. The next step is to show that $\alpha$-learners for the above problem must be correlated with the mean of the unknown data distribution. Let $\mathcal{D}$ be a distribution with mean $\mu$, and suppose $\mathcal{A}_n$ is an $\alpha$-learner for Equation (7). Then,

$$\mathbb{E}_{S_n \sim \mathcal{D}^{\otimes n}, \hat{\theta} \sim \mathcal{A}_n(S_n)} \left[ \left\langle \mu, \hat{\theta} \right\rangle \right] \geq \sup_{\theta \in \Theta} \langle \mu, \theta \rangle - k^{1/q} \cdot \alpha = d^{-1/p} \|\mu\|_1 - k^{1/q}\alpha.$$

Now, we apply the sparse fingerprinting lemma (Lemma 4.2). A key step is choosing the scale $\beta \geq 1$ of the beta-prior. On the one hand, $\beta$ should be small enough to guarantee $\mathbb{E} d^{-1/p} \|\mu\|_1 > k^{1/q}\alpha$, so that the above lower bound is non-vacuous. On the other hand, taking $\beta$ too small decreases the sample complexity of learning the problem, thus, disallowing

the desired level of recall. The optimal choice is $\beta \propto \alpha^{-2} \cdot (k/d)^{1/p}$, as long as this quantity is $\geq 1$. This choice yields

$$\text{Tr}_\kappa(\mathcal{P}_{k,p}; n, \alpha) \geq \mathbb{E}_{S_n \sim \mathcal{D}^{\otimes n}, \hat{\theta} \sim \mathcal{A}_n(S_n)} \left[ \frac{1}{n} \sum_{i=1}^{n} \phi(\hat{\theta}, Z_i) \right] \gtrsim \frac{d^{1-1/p}}{k^{1/2-1/p} n \alpha},$$

where $\kappa \in \Theta(1)$, and, for some universal constant $c > 0$, we let

$$\phi(\theta, Z) := \frac{cd^{1/p}}{\sqrt{k}} \left\langle \theta, \left( Z - \frac{d}{k}\mu \right) \right\rangle_{\text{supp}(Z)}.$$

Note that the $d^{1/p}/\sqrt{k}$ scaling ensures $\phi$ induces a 1-subgaussian process. Finally, it remains to choose a suitable value for $k$, for each pair $(p, \alpha)$. Recall the definition of $\mathcal{P}_{k,p}$ from Equation (7).

**Theorem 5.1.** *Let $\mathcal{P}_{k,p}$ be the family of problems described in Equation (7). There exist universal constants $c_1, c_2 > 0$ such that, for all $\alpha \in (0, 1/6]$ and $d \in \mathbb{N}$, the following subgaussian trace value lower bounds hold for all $p \in [1, \infty)$ and $\kappa \leq c_1\sqrt{d}$:*

*(i) For $p \leq 2$ and $k = d$, we have $\text{Tr}_\kappa(\mathcal{P}_{k,p}; n, \alpha) \geq c_2 \frac{\sqrt{d}}{n\alpha}$.*

*(ii) For $p \geq 2$ and $k = (6\alpha)^p d \vee 1$, we have $\text{Tr}_\kappa(\mathcal{P}_{k,p}; n, \alpha) \geq c_2 \left[ \frac{\sqrt{d}}{n(6\alpha)^{p/2}} \wedge \frac{d^{1-1/p}}{n\alpha} \right]$.*

Using the reduction Theorem 3.5, the above establishes Theorems 2.5 and 2.7.

**Refinement for $p = 1$.** While the above construction also yields a traceability result for $p = 1$, it is suboptimal for the following simple reason: for $k = d$, the problem in Equation (7) only requires $\Theta(1/\alpha^2)$ samples to learn, thus, it is impossible to trace out $\Omega(\log(d)/\alpha^2)$ samples. On the other hand, the problem in Equation (6) requires $\Theta(\log(d)/\alpha^2)$ samples to learn but is not traceable. The intuition we follow here is to modify the construction in Equation (7) to make $\Theta$ "look" more like an $\ell_1$-ball to drive up the sample complexity while still avoiding the counterexample with an ERM learner from the beginning of the section. In particular, we consider the following $\ell_1$-problem,

$$\Theta = \mathcal{B}_1(1) \cap \mathcal{B}_\infty(1/s), \quad \mathcal{Z} = \{\pm 1\}^d, \quad f(\theta, z) = -\langle z, \theta \rangle, \tag{8}$$

for a suitably chosen $s \in [d]$. Note that, if we choose $s \gg 1$, $\Theta$ above is a polytope with much more vertices $(2^s \binom{d}{s})$ than an $\ell_1$ ball $(2d)$, which would intuitively force a learner like an ERM to reveal more information about the training sample. On a technical level, selecting large $s$ improves the subgaussian constant of a tracer; however, selecting $s$ that is too large shrinks the diameter of the set, and thus, the problem becomes easier to learn. We must trade off these two aspects, and carefully set the value of $s$. As it turns out, the optimal choice is $s \propto d^{1-c}$ for any small $c > 0$ in order to establish Theorem 2.6. The remainder of the proof is rather technical and hence is deferred to Appendix F.2.

## 6 Limitations

We conclude by stating an intriguing open problem. We conjecture Theorem 2.8 is tight, and the dichotomy between traceability and SCO also holds for $p > 2$. In particular, we conjecture that the optimal DP-SCO excess risk for $\ell_p$ with $p > 2$ scales as

$$\min\left\{ \frac{d^{1/2-1/p}}{\sqrt{n}}, \left(\frac{1}{n}\right)^{1/p} \right\} + \min\left\{ \frac{d^{1-1/p}}{n\varepsilon}, \left(\frac{d}{\varepsilon^2 n^2}\right)^{1/p} \right\},$$

ignoring $\log(1/\delta)$ factors. If the conjecture is true, we have a complete understanding of traceability in SCO. If it is false, it reveals that there is something fundamentally different about settings with $p > 2$, which would also significantly enrich our understanding of DP-SCO.

## Acknowledgments

The authors would like to thank Mufan Li and Ziyi Liu for their comments on the drafts of this work.

## Funding

This work was completed while Sasha Voitovych was a student at the University of Toronto and supported by an Undergraduate Student Research Award from the Natural Sciences and Engineering Research Council of Canada. Mahdi Haghifam is supported by a Khoury College of Computer Sciences Distinguished Postdoctoral Fellowship. Idan Attias is supported by the National Science Foundation under Grant ECCS-2217023, through the Institute for Data, Econometrics, Algorithms, and Learning (IDEAL). Roi Livni is supported by a Google fellowship, a Vatat grant and the research has been funded, in parts, by an ERC grant (FoG - 101116258). Daniel M. Roy is supported by the funding through NSERC Discovery Grant and Canada CIFAR AI Chair at the Vector Institute.

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

# Appendix Contents

# A  Additional preliminaries

## A.1  Background on SCO

The next proposition summarizes the known minimax rates for learning SCO problems in general geometries. A proof can be found in [NY83; AWBR09; ST10].

**Theorem A.1.** *Fix $p \in [1, \infty]$, $d \in \mathbb{N}$, and $n \in \mathbb{N}$. Let $\alpha_{\mathsf{stat}}(\mathcal{L}_p^d, n)$ be the minimax excess risk rate of learning $\ell_p$-Lipschitz-bounded problems, as defined in Equation* (1). *Then,*

1. *For $p = 1$, we have*
$$\alpha_{\mathsf{stat}}(\mathcal{L}_p^d, n) \in \Theta\left(\sqrt{\frac{\log(d)}{n}}\right).$$

2. *For $1 < p \leq 2$, we have*
$$\alpha_{\mathsf{stat}}(\mathcal{L}_p^d, n) \in \Theta\left(\sqrt{\frac{\log(d)}{n}} \wedge \frac{1}{(p-1)\sqrt{n}}\right)$$

3. *For $2 \leq p < \infty$, we have*
$$\alpha_{\mathsf{stat}}(\mathcal{L}_p^d, n) \in \Theta\left(\frac{d^{1/2-1/p}}{\sqrt{n}} \wedge \frac{1}{n^{1/p}}\right)$$

4. *For $p = \infty$, we have*
$$\alpha_{\mathsf{stat}}(\mathcal{L}_p^d, n) \in \Theta\left(\sqrt{\frac{d}{n}}\right).$$

*Remark* A.2. Notice that, in the overparameterized regime $(d \geq n)$, the minimax excess risk for $p \geq 2$ is $\Theta\left(\left(\frac{1}{n}\right)^{1/p}\right)$ which is dimension-independent. This shows that for $d \geq n$, in all geometries except $p = \{1, \infty\}$, the minimax excess risk is dimension-free. ◁

This proposition implies the following corollary on the minimum number of samples required for $\alpha$-learners.

**Corollary A.3.** *Fix $p \in [1, \infty]$, $d \in \mathbb{N}$, and $\alpha \in (0, 1]$. Let $N_{\mathsf{stat}}(\mathcal{L}_p^d, n)$ be the sample complexity of learning problems $\mathcal{L}_p^d$ up to excess risk $\alpha$, i.e.,*

$$N_{\mathsf{stat}}(\mathcal{L}_p^d, n) = \min\left\{n \colon \alpha_{\mathsf{stat}}(\mathcal{L}_p^d, n) \leq \alpha\right\}.$$

*Then,*

1. *For $p = 1$, we have*
$$N_{\mathsf{stat}}(\mathcal{L}_p^d, n) \in \Theta\left(\frac{\log(d)}{\alpha^2}\right).$$

2. *For $1 < p \leq 2$, we have*
$$N_{\mathsf{stat}}(\mathcal{L}_p^d, n) \in \Theta\left(\frac{\log(d)}{\alpha^2} \wedge \frac{1}{((p-1)\alpha)^2}\right).$$

3. *For $2 \leq p < \infty$, we have*
$$N_{\mathsf{stat}}(\mathcal{L}_p^d, n) \in \Theta\left(\frac{d^{1-2/p}}{\alpha^2} \wedge \frac{1}{\alpha^p}\right).$$

4. *For $p = 1$, we have*
$$N_{\mathsf{stat}}(\mathcal{L}_p^d, n) \in \Theta\left(\frac{d}{\alpha^2}\right).$$

## A.2 Differential Privacy

**Definition A.4.** Let $\varepsilon > 0$ and $\delta \in [0, 1)$. A randomized mechanism $\mathcal{A}_n : \mathcal{Z}^n \to \mathcal{M}_1(\Theta)$ is $(\varepsilon, \delta)$-DP, iff, for every two neighboring datasets $S_n \in \mathcal{Z}^n$ and $S'_n \in \mathcal{Z}^n$ (that is, $S_n, S'_n$ differ in one element), and for every measurable subset $M \subseteq \Theta$, it holds

$$\mathrm{Pr}_{\hat{\theta} \sim \mathcal{A}_n(S_n)} \left( \hat{\theta} \in M \right) \leq e^\varepsilon \cdot \mathrm{Pr}_{\hat{\theta} \sim \mathcal{A}_n(S'_n)} \left( \hat{\theta} \in M \right) + \delta.$$

Algorithms that satisfy DP are not traceable in the sense of Definition 2.4 [KOV17]. The following simple proposition formalizes this observation.

**Proposition A.5.** *Fix $n \in \mathbb{N}$ and $\varepsilon, \delta > 0$. Let $\mathcal{A}_n$ be an $(\varepsilon, \delta)$-DP algorithm. Then, if $\mathcal{A}_n$ is $(\xi, m)$-traceable, it holds that*

$$m \leq n \exp(\varepsilon)\xi + n\delta.$$

## A.3 Concentration inequalities

First, we collect lemmata on the subgaussian norm, introduced in Section 3, which we use to derive concentration inequalities. The following is Equation (2.14) in [Ver18], and shows that a bound on subgaussian norm immediately leads to concentration inequalities.

**Lemma A.6** (Subgaussian concentration)**.** *There exists a universal constant $C$ such that the following holds for every random variable $X$ with $\|X\|_{\psi_2} < \infty$: for every $t \geq 0$,*

$$\Pr\left[|X| \geq t\right] \leq 2\exp\left(-\frac{ct^2}{\|X\|_{\psi_2}^2}\right)$$

The subgaussian norm behaves nicely under the summation of independent random variables. The following is Proposition 2.6.1 in [Ver18].

**Lemma A.7** (Sum of subgaussian variables)**.** *Let $C > 0$ be a universal constant. Let $X_1, \ldots, X_n$ be a collection of arbitrary independent real random variables. Then,*

$$\left\| \sum_{i=1}^n X_i \right\|_{\psi_2}^2 \leq C \sum_{i=1}^n \|X_i\|_{\psi_2}^2.$$

Subgaussian norm also behaves nicely under mixtures. In particular, we have the following proposition.

**Proposition A.8** (Subgaussian mixtures)**.** *Let $\{X_\alpha\}_{\alpha \in A}$ be $\sigma$-subgaussian random variables, and let $\pi$ be a distribution over the index set $A$. Then, a mixture of $\{X_\alpha\}_{\alpha \in A}$ under $\alpha \sim \pi$ is also $\sigma$-subgaussian.*

*Proof.* Let $Y$ be such mixture. Then, for every $t > 0$, we have

$$\mathbb{E}[\exp(Y^2/t^2)] = \mathbb{E}_{\alpha \sim \pi}\mathbb{E}[\exp(X_\alpha^2/t^2)].$$

Plugging in $t = \sigma$ into above, and using that $\|X_\alpha\|_{\psi_2} \leq \sigma$ for all $\alpha$, we have

$$\mathbb{E}[\exp(Y^2/\sigma^2)] = \mathbb{E}_{\alpha \sim \pi}\mathbb{E}[\exp(X_\alpha^2/t^2)] \leq 2,$$

i.e., $\|Y\|_{\psi_2} \leq \sigma$, as desired. $\qquad\square$

It is well-known that bounded random variables are subgaussian (Equation (2.17) of [Ver18]).

**Proposition A.9.** *Suppose $X$ is a random variable such that $X \in [-b, b]$ almost surely. Then,*

$$\|X\|_{\psi_2} \leq Cb,$$

*for some universal constant $C > 0$.*

We will heavily use the following result for the supremum of subgaussian processes (which follows from [Ver18, Theorem 8.1.6]). Let $\mathcal{N}(\Theta, \|\cdot\|, \varepsilon)$ denote the covering number of $\Theta$ in norm $\|\cdot\|$ at scale $\varepsilon > 0$.

**Proposition A.10.** *Let $\{X_\theta\}_{\theta \in \Theta}$ be a $\sigma$-subgaussian process w.r.t. a metric space $(\Theta, \|\cdot\|)$ as per Definition 3.1, and further assume that $\Theta$ is contained in the unit ball of $\|\cdot\|$. Let $t \geq 0$ be arbitrary. Then, with probability at least $1 - 4\exp(-t^2)$*

$$\sup_\theta X_\theta \leq C\sigma \left[ \int_0^1 \sqrt{\log \mathcal{N}(\Theta, \|\cdot\|, \varepsilon)} d\varepsilon + t \right],$$

*for some universal constant $C > 0$.*

*Proof.* Fix an arbitrary $\theta_0 \in \Theta$. Using Theorem 8.1.6 [Ver18], we obtain the following bound for the *increment* of the subgaussian process $\{X_\theta\}$,

$$\Pr \left[ \sup_{\theta \in \Theta} |X_\theta - X_{\theta_0}| \leq C\sigma \left( \int_0^\infty \sqrt{\log \mathcal{N}(\Theta, \|\cdot\|, \varepsilon)} d\varepsilon + t \right) \right] \geq 1 - 2\exp(-t^2).$$

First, note that for $\varepsilon \geq 1$, $\mathcal{N}(\Theta, \|\cdot\|, \varepsilon) = 1$, since $\Theta$ lies in the unit ball of $\|\cdot\|$. Thus,

$$\Pr \left[ \sup_{\theta \in \Theta} |X_\theta - X_{\theta_0}| \leq C\sigma \left( \int_0^1 \sqrt{\log \mathcal{N}(\Theta, \|\cdot\|, \varepsilon)} d\varepsilon + t \right) \right] \geq 1 - 2\exp(-t^2). \qquad (9)$$

Note that, by triangle inequality, we have

$$\sup_{\theta \in \Theta} |X_\theta - X_{\theta_0}| \geq \sup_{\theta \in \Theta} |X_\theta| - X_{\theta_0}. \qquad (10)$$

Since $\{X_\theta\}_{\theta \in \Theta}$ satisfies Definition 3.1, we have

$$\|X_{\theta_0}\|_{\psi_2} \leq 2\sigma.$$

From Lemma A.6, we then have

$$\Pr\left[|X_{\theta_0}| \leq c\sigma t\right] \geq 1 - 2\exp(-t^2),$$

for some constant $c > 0$. Combining this with Equation (10) and taking a union bound with Equation (9), we get

$$\Pr \left[ \sup_{\theta \in \Theta} |X_\theta| \leq C'\sigma \left( \int_0^1 \sqrt{\log \mathcal{N}(\Theta, \|\cdot\|, \varepsilon)} d\varepsilon + t \right) \right] \geq 1 - 4\exp(-t^2),$$

for some absolute constant $C' > 0$. $\qquad \square$

The following lemma is an anti-concentration inequality based on Paley–Zygmund inequality. It shows that if the sum of variables is large, one can conclude that many of them are large given an appropriate control over their sum of squares. It is given as Lemma A.4 in [ADHLR24], and it is also similar to Lemma 25 in [DSSUV15].

**Lemma A.11.** *Fix $n \in \mathbb{N}$ and $(a_1, \ldots, a_n) \in \mathbb{R}^n$. Let $A_1 := \sum_{i \in [n]} a_i$ and $A_2 := \sum_{i \in [n]} (a_i)^2$. Then, for every $\beta \in \mathbb{R}$, $\left| \{ i \in [n] : a_i \geq \beta/n \} \right| \geq \frac{(\max\{A_1 - \beta, 0\})^2}{A_2}$.*

### A.4 Beta distributions

Next definitions are the versions of beta distributions that we use in this paper. Recall that, classically, beta distribution is supported on $[0, 1]$. However, in our results, it is convenient to consider the rescaled and centered variants.

**Definition A.12.** Fix $\beta > 0$. A (symmetric) beta distribution denoted by s-beta $(\beta, \beta)$ is a continuous distribution, such that, if $X \sim$ s-beta $(\beta, \beta)$, then, for every $a \in [-1, 1]$, we have

$$\Pr(X \leq a) = \int_{-1}^a \frac{(1 - x^2)^{\beta-1}}{B(\beta)} dx,$$

where $B(\beta) = 2^{2\beta-1} \Gamma(\beta)^2 / \Gamma(2\beta)$.

**Definition A.13.** Fix $\beta > 0$ and $\gamma \in (0, 1]$. We define rescaled (symmetric) beta distribution, denoted by $\mathsf{s\text{-}beta}_{[-\gamma,\gamma]}(\beta, \beta)$, where for $a \in [-\gamma, \gamma]$, its distribution is given by

$$\Pr(X \leq a) = \frac{1}{\gamma B(\beta)} \int_{-\gamma}^{a} \left(1 - \left(\frac{x}{\gamma}\right)^2\right)^{\beta-1} dx,$$

where $B(\beta) = 2^{2\beta-1} \Gamma(\beta)^2 / \Gamma(2\beta)$.

We have the following result on the first moment of the beta distribution.

**Lemma A.14.** *Fix $\beta > 0$. Let $X \sim \mathsf{s\text{-}beta}(\beta, \beta)$ where $\beta \geq 1$. Then,*

$$\mathbb{E}|X| \geq \frac{1}{3\sqrt{\beta}}.$$

*Proof.* Let $B(\beta) = 2^{2\beta-1} \Gamma(\beta)^2 / \Gamma(2\beta)$ be the normalization constant. We have

$$\mathbb{E}|X| = \frac{1}{B(\beta)} \int_{-1}^{1} |x| (1 - x^2)^{\beta-1} dx$$

$$= \frac{1}{B(\beta)} \int_{0}^{1} 2x(1 - x^2)^{\beta-1} dx$$

$$= \frac{1}{\beta \cdot B(\beta)}.$$

It remains to upper bound $B(\beta)$. It follows from Theorem 1.5 of [Bat08] that, for every $x \geq 1$, we have

$$a \left(\frac{x - 1/2}{e}\right)^{x-1/2} \leq \Gamma(x) \leq b \left(\frac{x - 1/2}{e}\right)^{x-1/2},$$

where $a = \sqrt{2e}$ and $b = \sqrt{2\pi}$ are absolute constants. Thus,

$$B(\beta) = \frac{2^{2\beta-1} \Gamma(\beta)^2}{\Gamma(2\beta)} \leq \frac{2^{2\beta-1} b^2 \left(\frac{\beta-1/2}{e}\right)^{2\beta-1}}{a \left(\frac{2\beta-1/2}{e}\right)^{2\beta-1/2}}$$

$$= \frac{b^2 \sqrt{e}}{a} (2\beta - 1/2)^{-1/2} \left(\frac{2\beta - 1}{2\beta - 1/2}\right)^{2\beta-1}$$

$$\leq \frac{b^2 \sqrt{e}}{a} (2\beta - 1/2)^{-1/2}$$

$$= \frac{2\pi \sqrt{e}}{\sqrt{2e}} (2\beta - 1/2)^{-1/2}$$

$$= \pi (\beta - 1/4)^{-1/2}$$

$$\leq \pi \left(\frac{3}{4\beta}\right)^{1/2},$$

where in the last line we used $\beta - 1/4 \geq \frac{3}{4}\beta$ which holds as $\beta \geq 1$. Thus,

$$\mathbb{E}|X| \geq \frac{1}{\pi (3/4)^{1/2}} \frac{1}{\sqrt{\beta}} \geq \frac{1}{3\sqrt{\beta}},$$

as desired. $\qquad\square$

Since the density of the rescaled beta distribution is homogeneous w.r.t. $\gamma$, we have the following result.

**Corollary A.15.** *Fix $\beta \geq 1$ and $\gamma \in (0, 1]$. Let $X \sim \mathsf{s\text{-}beta}_{[-\gamma,\gamma]}(\beta, \beta)$. Then,*

$$\mathbb{E}|X| \geq \frac{\gamma}{3\sqrt{\beta}}.$$

# B Additional Related Work

**Necessity of memorization in learning.** A parallel line of work investigated memorization using the notion of *label memorization* in supervised setups. As per this definition, a learner is said to memorize its training samples if it "overfits" at these points. Feldman [Fel20] showed that, in some classification tasks, if the underlying distribution is *long-tailed*, then a learner is forced to memorize many training labels. Cheng, Duchi, and Kuditipudi [CDK22] showed this phenomenon also occurs in the setting of linear regression. While this framework is suitable to study memorization in supervised tasks, the notion of "labels" in SCO in not well-defined and thus calls for alternative definitions.

Another line of work studied memorization through the lens of information theoretic measures. Brown, Bun, Feldman, Smith, and Talwar [BBFST21] used *input-output mutual information* (IOMI) as a memorization metric and showed that IOMI can scale linearly with the training sample's entropy, indicating that a constant fraction of bits is memorized. In the context of SCO in $\ell_2$ geometry, lower bounds on IOMI have been studied in [HRTSMK23; Liv24]. Specifically [Liv24] demonstrated that, for every accurate algorithm, its IOMI must scale with dimension $d$. Our approach to the study of memorization is conceptually different since we focus on the number of *samples* memorized as opposed to the number of *bits*. Nevertheless, it can be shown using Lemma H.3 and [HNKRD20, Thm. 2.1] that the recall *lower bounds* IOMI of an algorithm (provided that soundness parameter $\xi$ is small enough, e.g., $\xi = 1/n^2$). However, because of the Lipschitzness of loss functions in $\ell_p$ SCO, we can use discretization of $\Theta$ and design algorithms with IOMI that is significantly smaller that the entropy of the training set, thus, memorization in the sense of [BBFST21] does not arise here.

**Membership inference.** Membership inference is an important practical problem [HSRDTMPSNC08; SSSS17; CCNSTT22]. In these works, the focus is on devising strategies for the tracer in modern machine learning settings, particularly neural networks. Our work takes a more fundamental perspective, aiming to determine whether membership inference is inherently unavoidable or simply a byproduct of specific training algorithms. An interesting aspect of our results is that, for $1 < p \leq 2$, the optimal strategy for tracing depends *only* on the loss function, which is in line with empirical studies [SDSOJ19].

**Private Stochastic Convex Optimization.** DP-SCO has been extensively studied in $\ell_2$ geometry (see, for instance, [CMS11; BST14; BFTG19; FKT20]). For $\ell_p$ with $p \in [1, 2)$, the optimal DP excess risk was established in [AFKT21; BGN21]. The best known upper bounds for DP-SCO in $\ell_p$ geometry for $p > 2$ are due to [BGN21; GLLST23]. In this setting, there is a long-standing gap between upper and lower bounds, and the best known lower bounds are due to [ABGMU22; LLL24], which our paper improves on.

## B.1 Detailed comparison with [DSSUV15; BST14].

One might hope that existing traceability results (such as [DSSUV15]) and a clever reduction to mean estimation (such as [BST14, Section 5.1]) might yield optimal results for SCO. Here, we will demonstrate rigorously that merely combining results and techniques of [DSSUV15; BST14] yields suboptimal results for the setup of SCO, even in the simple setting of $\ell_2$ geometry. [BST14] considers the following $\ell_2$ problem:

$$\Theta = \mathcal{B}_2(1), \quad \mathcal{Z} = \left\{ \pm \frac{1}{\sqrt{d}} \right\}^d, \quad f(\theta, Z) = -\langle \theta, Z \rangle.$$

To apply fingerprinting to establish traceability, we first need to posit a prior distribution over the unknown distribution. [DSSUV15] does so by considering product distributions over $\mathcal{Z}$, and placing a uniform prior over the mean $\mu \in [-1/\sqrt{d}, 1/\sqrt{d}]^d$. We now show that this (Bayesian) problem requires only $O(1/\alpha)$ samples to learn, and thus, tracing $\Omega(1/\alpha^2)$ samples is clearly impossible. Consider the ERM learner $\hat{\theta}$. It is easy to see that $\hat{\theta}$ can be written as:

$$\hat{\theta} = \frac{\hat{\mu}}{\|\hat{\mu}\|_2},$$

where $\hat{\mu}$ is the empirical mean of the dataset, that is, $\hat{\mu} = \frac{1}{n}\sum_{i=1}^{n} Z_i$. Similarly, the population risk minimizer $\theta^\star$ is $\mu/\|\mu\|_2$. The expected excess risk of $\hat{\theta}$ is then:

$$
\mathbb{E}\left\langle \frac{\mu}{\|\mu\|_2}, \mu \right\rangle - \left\langle \frac{\hat{\mu}}{\|\hat{\mu}\|_2}, \mu \right\rangle = \mathbb{E}\frac{\|\mu\|_2 \|\hat{\mu}\|_2 - \langle \hat{\mu}, \mu \rangle}{\|\hat{\mu}\|_2}
$$

$$
\leq^{(a)} \mathbb{E}\left[ \frac{\frac{1}{2}\|\mu\|_2^2 + \frac{1}{2}\|\hat{\mu}\|_2^2 - \langle \hat{\mu}, \mu \rangle}{\|\hat{\mu}\|_2} \wedge 2 \right]
$$

$$
= \frac{1}{2}\mathbb{E}\left[ \frac{\|\mu - \hat{\mu}\|_2^2}{\|\hat{\mu}\|_2} \wedge 4 \right],
$$

where in (a) we used the AM-GM inequality, and the fact that the expression on the preceding line is always bounded by 2. The intuition behind the rest of the argument is that, due to the uniform prior on $\mu$, we have $\|\mu\|, \|\hat{\mu}\| \in \Omega(1)$ with high probability. At the same time

$$
\mathbb{E}\left[ \|\mu - \hat{\mu}\|^2 \right] = \mathbb{E}\frac{1}{dn}\sum_{i=1}^{n} 2\mu_i(1 - \mu_i) \leq \frac{1}{2n},
$$

thus, expected risk will be on the order of $O(1/n)$. To formalize this, note that $\mathbb{E}\|\mu\|_2^2 = 1/3$, and $\|\mu\|^2$ is a sum of $d$ independent random variables bounded by $1/\sqrt{d}$ in absolute value. Hoeffding's inequality then yields that we have $\|\mu\|_2^2 \geq 1/6$ with very high probability. Similarly, we can obtain $\|\mu - \hat{\mu}\|_2^2 \leq 1/(2n) + 1/36 \leq 1/12$ for large enough $n$, with high probability. Then, with high probability, event

$$
\mathcal{E} := \left\{ \|\hat{\mu}\| \geq \frac{1}{\sqrt{6}} - \frac{1}{\sqrt{12}} \geq 0.1 \right\}
$$

holds Then, the excess risk is upper bounded by

$$
\mathbb{E}\left\langle \frac{\mu}{\|\mu\|_2}, \mu \right\rangle - \left\langle \frac{\hat{\mu}}{\|\hat{\mu}\|_2}, \mu \right\rangle \leq \frac{1}{2}\mathbb{E}\left[ \frac{\|\mu - \hat{\mu}\|_2^2}{\|\hat{\mu}\|_2} \wedge 4 \right]
$$

$$
= \frac{1}{2}\mathbb{E}\left[ \mathbb{1}(\mathcal{E})\left( \frac{\|\mu - \hat{\mu}\|_2^2}{\|\hat{\mu}\|_2} \wedge 4 \right) \right] + \frac{1}{2}\mathbb{E}\left[ \mathbb{1}(\mathcal{E}^c)\left( \frac{\|\mu - \hat{\mu}\|_2^2}{\|\hat{\mu}\|_2} \wedge 4 \right) \right]
$$

$$
\leq \frac{1}{2}\mathbb{E}\left[ 10\|\mu - \hat{\mu}\|_2^2 \right] + 2\Pr(\mathcal{E}^c)
$$

$$
\in O(1/n),
$$

as desired.

## C Proofs from Section 3

### C.1 Proof of Lemma 3.4

We first prove a slightly more general concentration statement to bound the supremum in Lemma 3.4, which will be useful to reuse in other proofs. Let $\mathcal{N}(\Theta, \|\cdot\|, \varepsilon)$ denote the size of the minimal cover of $\Theta$ in norm $\|\cdot\|$ at scale $\varepsilon > 0$. Then, the more general statement is given below.

**Lemma C.1.** *Fix $n, d \in \mathbb{N}$. Suppose $\Theta \subset \mathbb{R}^d$ is a subset of a unit ball in some norm $\|\cdot\|$. Let $\phi \colon \Theta \times \mathcal{Z} \to \mathbb{R}$ and $\mathcal{D} \in \mathcal{M}_1(\mathcal{Z})$ be such that, as $Z \sim \mathcal{D}$, $\{\phi(\theta, Z)\}$ is a $\sigma$-subgaussian process w.r.t. $(\Theta, \|\cdot\|)$ and for every $\theta \in \Theta$, $\mathbb{E}[\phi(\theta, Z)] = 0$. Let $(Z_1, \ldots, Z_n) \sim \mathcal{D}^{\otimes n}$. Then, there exist a universal constant $C > 0$, such that for every $t \geq 0$,*

$$
\Pr\left[ \sup_{\theta \in \Theta} \sqrt{\sum_{i=1}^{n} [\phi(\theta, Z_i)]^2} \leq C\sigma\left( \sqrt{n} + \int_0^1 \sqrt{\log \mathcal{N}(\Theta; \|\cdot\|, \varepsilon)}d\varepsilon + t \right) \right] \geq 1 - 4\exp(-t^2).
$$

*Proof.* Let $\Phi_\theta$ denote the following random vector

$$\Phi_\theta = \begin{bmatrix} \phi(\theta, Z_1) \\ \vdots \\ \phi(\theta, Z_n) \end{bmatrix}.$$

Then, observe that, the desired quantity is equal to

$$\sup_{\theta \in \Theta} \sqrt{\sum_{i=1}^n [\phi(\theta, Z_i)]^2} = \sup_{\theta \in \Theta} \|\Phi_\theta\|_2 = \sup_{\theta \in \Theta, x \in \mathbb{S}^{n-1}} \langle x, \Phi_\theta \rangle.$$

Then, $\langle x, \Phi_\theta \rangle$ can be seen to be a random process parameterized by a pair $(x, \theta)$. We will show that it is, in fact, a subgaussian process. Indeed, note that, by triangle inequality,

$$\|\langle x, \Phi_\theta \rangle - \langle x', \Phi_{\theta'} \rangle\|_{\psi_2} \le \|\langle x - x', \Phi_\theta \rangle\|_{\psi_2} + \|\langle x', \Phi_\theta - \Phi_{\theta'} \rangle\|_{\psi_2}. \tag{11}$$

Since $\Phi_\theta^i$ is $\sigma$-subgaussian for each $i$, we have by Lemma A.7,

$$\|\langle x - x', \Phi_\theta \rangle\|_{\psi_2} \le C\sigma \|x - x'\|_2,$$

for some universal constant $C > 0$. Now, for every $i$, $(\Phi_\theta - \Phi_{\theta'})^i$ is $\sigma \|\theta - \theta'\|$-subgaussian. Therefore, by Lemma A.7, we have

$$\|\langle x', \Phi_\theta - \Phi_{\theta'} \rangle\|_{\psi_2} = \left\| \sum_{i=1}^n (x')^i (\Phi_\theta^i - \Phi_{\theta'}^i) \right\|_{\psi_2} \le C\sigma \|\theta - \theta'\|.$$

Combining the two inequalities, we get

$$\|\langle x, \Phi_\theta \rangle - \langle x', \Phi_{\theta'} \rangle\|_{\psi_2} \le C\sigma \|\theta - \theta'\| + C\sigma \|x - x'\|_2$$
$$= 2C\sigma \cdot \frac{1}{2} \left[ \|\theta - \theta'\| + \|x - x'\|_2 \right].$$

Thus, $\langle x, \Phi_\theta \rangle$ is $(2C\sigma)$-subgaussian process w.r.t the norm $\gamma$, defined as

$$\gamma((x, \theta)) := \frac{1}{2} \left[ \|x\|_2 + \|\theta\| \right].$$

Moreover, we can see that $\Theta \times \mathbb{S}^{n-1}$ is a subset of a unit ball in $\gamma$. By definition of $\gamma$, we have

$$\mathcal{N}\left(\mathbb{S}^{n-1} \times \Theta; \gamma, \varepsilon\right) \le \mathcal{N}\left(\mathbb{S}^{n-1}; \|\cdot\|_2, \varepsilon\right) \cdot \mathcal{N}\left(\Theta; \|\cdot\|, \varepsilon\right). \tag{12}$$

Then, using Proposition A.10, we have, for some constant $K > 0$, that with probability $1 - 4\exp(-t^2)$

$$\sup_\theta \|\Phi_\theta\|_2 \le K\sigma \left[ \int_0^1 \sqrt{\log \mathcal{N}\left(\Theta \times \mathbb{S}^{n-1}; \gamma, \varepsilon\right)} d\varepsilon + t \right]$$
$$\le K\sigma \left[ \int_0^1 \sqrt{\log \mathcal{N}\left(\mathbb{S}^{n-1}; \|\cdot\|_2, \varepsilon\right) + \log \mathcal{N}\left(\Theta; \|\cdot\|, \varepsilon\right)} d\varepsilon + t \right]$$
$$\le^{(a)} K\sigma \left[ \int_0^1 \sqrt{n \log\left(1 + \frac{4}{\varepsilon}\right)} d\varepsilon + \int_0^1 \sqrt{\log \mathcal{N}\left(\Theta; \|\cdot\|, \varepsilon\right)} d\varepsilon + t \right]$$
$$\le K'\sigma \left[ \sqrt{n} + \int_0^1 \sqrt{\log \mathcal{N}\left(\Theta; \|\cdot\|, \varepsilon\right)} d\varepsilon + t \right],$$

as desired, where in (a) we used Example 5.8 from [Wai19], and $K' > 0$ is some other universal constant. $\qquad \square$

Using Example 5.8 from [Wai19] once again to upper bound $\sqrt{\log \mathcal{N}\left(\Theta; \|\cdot\|, \varepsilon\right)}$, we have the proof of Lemma 3.4.

**Lemma 3.4.** *Fix $n, d \in \mathbb{N}$. Suppose $\Theta \subset \mathbb{R}^d$ is a subset of a unit ball in some norm $\|\cdot\|$. Let $\phi \colon \Theta \times \mathcal{Z} \to \mathbb{R}$ and $\mathcal{D} \in \mathcal{M}_1(\mathcal{Z})$ be such that, as $Z \sim \mathcal{D}$, $\{\phi(\theta, Z)\}$ is a $\sigma$-subgaussian process w.r.t. $(\Theta, \|\cdot\|)$. Let $(Z_1, \ldots, Z_n) \sim \mathcal{D}^{\otimes n}$. Then, there is a constant $C > 0$, such that*

$$\Pr\left[\sup_{\theta \in \Theta} \sqrt{\sum_{i=1}^{n} [\phi(\theta, Z_i)]^2} \leq C\sigma\left(\sqrt{n} + \sqrt{d} + t\right)\right] \geq 1 - 4\exp(-t^2), \quad \forall t \geq 0.$$

*Proof.* From Example 5.8 in [Wai19], we have

$$\log \mathcal{N}\left(\Theta; \|\cdot\|, \varepsilon\right) \leq d \log\left(1 + \frac{2}{\varepsilon}\right).$$

Plugging this into the result of Lemma C.1, with probability at least $1 - 4\exp(-t^2)$, we have

$$\sup_{\theta \in \Theta} \sqrt{\sum_{i=1}^{n} [\phi(\theta, Z_i)]^2} \leq C\sigma\left(\sqrt{n} + \int_0^1 \sqrt{\log \mathcal{N}\left(\Theta; \|\cdot\|, \varepsilon\right)} d\varepsilon + t\right)$$

$$\leq C\sigma\left(\sqrt{n} + \int_0^1 \sqrt{d \log\left(1 + \frac{2}{\varepsilon}\right)} d\varepsilon + t\right)$$

$$\leq C'\sigma\left(\sqrt{n} + \sqrt{d} + t\right),$$

for some other universal constant $C' > 0$. $\qquad\square$

### C.2  Proof of Theorem 3.5

**Theorem 3.5.** *Fix $n \in \mathbb{N}$, $d \in \mathbb{N}$, $\kappa > 0$ and $\alpha \in [0, 1]$. Consider an arbitrary SCO problem $\mathcal{P} = (\Theta, \mathcal{Z}, f)$. Let $T = \mathrm{Tr}_\kappa(\mathcal{P}; n, \alpha)$ be the subgaussian trace value of $\mathcal{P}$. Then, for some constant $c > 0$, every $\alpha$-learner $\mathcal{A}_n$ is $(\xi, m)$-traceable with*

$$\xi = \exp(-cT^2), \quad m = c\left[\frac{n^2 T^2}{n + d} - \frac{16\kappa^2 n}{\exp(n + d)}\right].$$

*Proof.* We set

$$\lambda := \frac{T}{2}$$

First, we show that the soundness condition holds. Since $Z$ and $\hat{\theta}$ are independent, and using the subgaussian nature of $\phi(\hat{\theta}, Z)$, we have by Lemma A.6

$$\Pr_{Z \sim \mathcal{D}}\left[\phi(\hat{\theta}, Z) \geq \lambda\right] \leq \exp\left(-c\lambda^2\right) \leq \exp\left(-cT^2/4\right),$$

where $c > 0$ is some constant. For recall, let's define the set $\mathcal{I}$ as follows

$$\mathcal{I} = \{i \in [n] \colon \phi(\hat{\theta}, Z_i) \geq \lambda\}.$$

Using Lemma A.11, we have

$$\mathbb{E}\left[|\mathcal{I}|\right] = \mathbb{E}\left|\{i \in [n] \colon \phi(\hat{\theta}, Z_i) \geq \lambda\}\right|$$

$$\geq \mathbb{E}\left[\frac{\left(\sum_{i=1}^{n} \phi(\hat{\theta}, Z_i) - n\lambda\right)_+^2}{\sum_{i=1}^{n} \phi(\hat{\theta}, Z_i)^2}\right]$$

$$\geq \mathbb{E}\left[\frac{\left(\sum_{i=1}^{n} \phi(\hat{\theta}, Z_i) - n\lambda\right)_+^2}{\sup_\theta \|\{\phi(\theta, Z_i)\}_{i=1}^n\|_2^2}\right],$$

where for every $x \in \mathbb{R}$, we define $(x)_+ = \max\{x, 0\}$. Then, Lemma 3.4 tells us that, for $t := \sqrt{n+d}$, we have with probability $1 - 4\exp(-t^2)$,

$$\sup_{\theta \in \Theta} \left\| \left[ \phi(\hat{\theta}, Z_1), \ldots, \phi(\hat{\theta}, Z_n) \right]^\top \right\|_2^2 \leq C(n+d),$$

for some constant $C > 0$. Thus,

$$\Pr \left[ \mathcal{E} := \left\{ \sup_{\theta \in \Theta} \left\| \left[ \phi(\hat{\theta}, Z_1), \ldots, \phi(\hat{\theta}, Z_n) \right]^\top \right\|_2^2 \leq C(n+d) \right\} \right] \geq 1 - 4\exp(-t^2).$$

This implies,

$$
\begin{aligned}
\mathbb{E}|\mathcal{I}| &\geq \mathbb{E}\left[ \frac{\left( \sum_{i=1}^n \phi(\hat{\theta}, Z_i) - n\lambda \right)_+^2}{\sup_\theta \|\{\phi(\theta, Z_i)\}_{i=1}^n\|_2^2} \right] \\
&\geq \mathbb{E}\left[ \frac{\left( \sum_{i=1}^n \phi(\hat{\theta}, Z_i) - n\lambda \right)_+^2}{\sup_\theta \|\{\phi(\theta, Z_i)\}_{i=1}^n\|_2^2} \mathbb{1}(\mathcal{E}) \right] \\
&= \mathbb{E}\left[ \frac{\left( \sum_{i=1}^n \phi(\hat{\theta}, Z_i) - n\lambda \right)_+^2}{C(n+d)} \mathbb{1}(\mathcal{E}) \right] \\
&\geq \mathbb{E}\left[ \frac{\left( \sum_{i=1}^n \phi(\hat{\theta}, Z_i) - n\lambda \right)_+^2}{C(n+d)} \right] - \mathbb{E}\left[ \frac{\left( \sum_{i=1}^n \phi(\hat{\theta}, Z_i) - n\lambda \right)_+^2}{C(n+d)} \mathbb{1}(\mathcal{E}^c) \right].
\end{aligned}
$$

We know that, almost surely,

$$\phi(\hat{\theta}, Z)^2 \leq \kappa^2.$$

Thus, almost surely,

$$\left( \sum_{i=1}^n \phi(\hat{\theta}, Z_i) - n\lambda \right)_+^2 \leq \left( \sum_{i=1}^n \left( \phi(\hat{\theta}, Z_i) - \lambda \right)_+ \right)^2 \leq n \sum_{i=1}^n \phi(\hat{\theta}, Z_i)^2 \leq \kappa^2 n^2.$$

Thus,

$$
\begin{aligned}
\mathbb{E}\left[ \frac{\left( \sum_{i=1}^n \phi(\hat{\theta}, Z_i) - n\lambda \right)_+^2}{C(n+d)} \mathbb{1}(\mathcal{E}^c) \right] &\leq \Pr[\mathcal{E}^c] \cdot \frac{\kappa^2 n^2}{C(n+d)} \\
&\leq \frac{4\kappa^2 n}{C\exp(n+d)}.
\end{aligned}
$$

Hence,

$$
\begin{aligned}
\mathbb{E}\left[|\mathcal{I}|\right] &\geq \mathbb{E}\left[\frac{\left(\sum_{i=1}^{n}\phi(\hat{\theta},Z_i)-n\lambda\right)_{+}^{2}}{C\left(n+d\right)}\right] - \mathbb{E}\left[\frac{\left(\sum_{i=1}^{n}\phi(\hat{\theta},Z_i)-n\lambda\right)_{+}^{2}}{C\left(n+d\right)}\mathbb{1}(\mathcal{E}^{c})\right] \\
&\geq \mathbb{E}\left[\frac{\left(\sum_{i=1}^{n}\phi(\hat{\theta},Z_i)-n\lambda\right)_{+}^{2}}{C\left(n+d\right)}\right] - \frac{4\kappa^{2}n}{C\exp(n+d)} \\
&\geq^{(a)} \frac{\left(\sum_{i=1}^{n}\mathbb{E}\phi(\hat{\theta},Z_i)-n\lambda\right)_{+}^{2}}{C\left(n+d\right)} - \frac{4\kappa^{2}n}{C\exp(n+d)} \\
&\geq \frac{n^{2}T^{2}/4}{C\left(n+d\right)} - \frac{4\kappa^{2}n}{C\exp(n+d)} \\
&= c\left[\frac{n^{2}T^{2}}{n+d} - \frac{16\kappa^{2}n}{\exp(n+d)}\right],
\end{aligned}
$$

where $(a)$ follows by Jensen's inequality and $c = 1/4C$. $\qquad\square$

### C.3 Proof of Theorem 3.6

**Theorem 3.6.** *There exists a universal constant $c > 0$, such that the following holds. Fix $p \in [1,\infty)$, $n \in \mathbb{N}$, $d \in \mathbb{N}$, $\alpha \in [0,1]$, $\kappa > 0$ $\varepsilon > 0$, and $\delta \in [0,1]$. Consider an arbitrary SCO problem $\mathcal{P} = (\Theta, \mathcal{Z}, f)$ in $\mathbb{R}^d$. Let $T = \mathrm{Tr}_{\kappa}(\mathcal{P}; n, \alpha)$ be the subgaussian trace value of problem $\mathcal{P}$. Then, for every $(\varepsilon, \delta)$-DP $\alpha$-learner $\mathcal{A}_n$, we have $\exp(\varepsilon) - 1 \geq c\left(T - 2\delta\kappa\right)$.*

*Proof.* Consider an arbitrary distribution $\mathcal{D}$ and a function $\phi$ s.t. $\{\phi(\theta, Z)\}_{\theta \in \Theta}$ is a $1$-subgaussian process w.r.t. $(\Theta, \|\cdot\|_{\Theta})$ and $|\phi| \leq \kappa$ almost surely. Consider a sample $S_n = (Z_1, \ldots, Z_n)$ and let $Z_0$ be a freshly sampled point; let $S_n^{(i)}$ be a sample with $Z_i$ substituted by $Z_0$. Let $\hat{\theta}$ be a learner trained on $S_n$ and $\hat{\theta}^{(i)}$ be a learner trained on $S_n^{(i)}$. Then, since $\hat{\theta}$ is $(\varepsilon, \delta)$-DP and noting that $\phi(\theta, Z)$ is supported on $[-\kappa, \kappa]$, we may apply Lemma A.1 of [FS17] and get

$$
\left|\mathbb{E}\phi(\hat{\theta},Z_i) - \mathbb{E}\phi(\hat{\theta}^{(i)},Z_i)\right| \leq \mathbb{E}\left|\phi(\hat{\theta}^{(i)},Z_i)\right|(\exp(\varepsilon) - 1) + 2\delta\kappa.
$$

By independence of $\hat{\theta}$ and $Z_i$, we conclude that $\phi(\hat{\theta}, Z_i)$ is $1$-subgaussian random variable. It is well-known that $\mathbb{E}|X| \leq C\sigma$ if $X$ is $\sigma$-subgaussian for some constant $C$ (see part (ii) of Proposition 2.5.2 of [Ver18] for $p = 1$), thus the above gives

$$
\left|\mathbb{E}\phi(\hat{\theta},Z_i)\right| \leq C(\exp(\varepsilon) - 1) + 2\delta\kappa.
$$

Then, for every $\mathcal{D}$ and $\phi$ we get that,

$$
\mathbb{E}\frac{1}{n}\sum_{i=1}^{n}\phi(\hat{\theta},Z_i) \leq C(\exp(\varepsilon) - 1) + 2\delta\kappa.
$$

Thus,

$$
T \leq C(\exp(\varepsilon) - 1) + 2\delta\kappa,
$$

which, after rearranging, implies the desired result. $\qquad\square$

## D  Proofs of fingerprinting lemmas (Section 4)

### D.1 Proof of Lemma 4.2

**Lemma 4.2** (Sparse fingerprinting)**.** *Fix $d, n \in \mathbb{N}$ and let $k \in [d]$. For each $\mu \in [-k/d, k/d]^d$, let $\mathcal{Z}_k$ and $\mathcal{D}_{\mu,k}$ be as in Definition 4.1 . Let $\pi = \mathsf{s\text{-}beta}_{[-k/d,k/d]}(\beta,\beta)^{\otimes d}$ be a prior and set*

$$
\phi_{\mu}(\theta, Z) := \left\langle \theta, \left(Z - \frac{d}{k}\mu\right)\right\rangle_{\mathrm{supp}(Z)}.
$$

*Then, for every learning algorithm $\mathcal{A}_n : \mathcal{Z}^n \to \mathcal{M}_1(\mathbb{R}^d)$ with sample $S_n = (Z_1, \ldots, Z_n)$,*

$$\mathbb{E}_{\mu \sim \pi} \mathbb{E}_{S_n \sim \mathcal{D}_{\mu,k}^{\otimes n}, \hat{\theta} \sim \mathcal{A}_n(S_n)} \left[ \sum_{i=1}^n \phi_\mu(\hat{\theta}, Z_i) \right] = \frac{2\beta d}{k} \mathbb{E}_{\mu \sim \pi} \left\langle \mu, \mathbb{E}_{S_n \sim \mathcal{D}_{\mu,k}^{\otimes n}, \hat{\theta} \sim \mathcal{A}_n(S_n)}[\hat{\theta}] \right\rangle.$$

*Proof.* For each $j \in [d]$, let $\mathcal{I}_j := \{i \in [n] : Z_i^j \neq 0\}$ as the index of the training points such that their $j$-th coordinate is non-zero. Then, we have

$$\mathbb{E} \left[ \sum_{i=1}^n \phi_\mu(\hat{\theta}, Z_i) \right] = \mathbb{E} \left[ \sum_{j=1}^d \sum_{i \in \mathcal{I}_j} \left( (\hat{\theta})^j \left( Z_i^j - \frac{d}{k} \mu^j \right) \right) \right] \tag{13}$$

$$= \sum_{j=1}^d \mathbb{E} \left[ \sum_{i \in \mathcal{I}_j} (\hat{\theta})^j \left( Z_i^j - \frac{d}{k} \mu^j \right) \right]. \tag{14}$$

Then, define the following function

$$g^j(\mu^j) := \mathbb{E} \left[ (\hat{\theta})^j \, \middle| \, \{\mathcal{I}_r\}_{r \in [d]}, \{Z_i^m\}_{m \neq j, i \in [n]} \right].$$

We claim

$$\mathbb{E} \left[ \sum_{i \in \mathcal{I}_j} (\hat{\theta})^j \cdot \left( Z_i^j - \frac{d}{k} \mu^j \right) \, \middle| \, \{\mathcal{I}_r\}_{r \in [d]}, \{Z_i^m\}_{m \neq j, i \in [n]} \right] = \frac{k}{d} \left( 1 - \left( \frac{d}{k} \mu^j \right)^2 \right) \frac{d}{d\mu^j} g^j(\mu^j). \tag{15}$$

The proof is based on the following two observations: 1) conditioned on $\{Z_i^m\}_{m \neq j, i \in [n]}$, $\hat{\theta}$ is a function of $\{Z_i^j\}_{i \in [n]}$, 2) conditioned on $\{\mathcal{I}_r\}_{r \in [d]}$ the non-zero elements in $\{Z_i^j\}_{i \in [n]}$ are sampled i.i.d from $\{\pm 1\}$ with mean $\frac{d}{k} \mu^j$. Then, based on these observations Equation (15) follows as an straightforward application of [Ste16, Lemma 4.3.7].

Recall the definition of $\pi$ and notice that $\pi$ is a product measure. Let $\pi^j$ be the distribution on the $j$-th coordinate. By the definition of the prior distribution, we can write

$$\mathbb{E}_{\mu^j \sim \pi^j} \left[ \frac{k}{d} \left( 1 - \left( \frac{d}{k} \mu^j \right)^2 \right) \frac{d}{d\mu^j} g^j(\mu^j) \right]$$

$$= \frac{1}{C} \int_{-\frac{k}{d}}^{+\frac{k}{d}} \frac{k}{d} \left( 1 - \left( \frac{d}{k} v \right)^2 \right) \frac{d}{dv} g^j(v) \left( 1 - \left( \frac{d}{k} v \right)^2 \right)^{\beta-1} dv$$

$$= \frac{1}{C} \int_{-\frac{k}{d}}^{+\frac{k}{d}} \frac{k}{d} \frac{d}{d\mu^j} g^j(v) \left( 1 - \left( \frac{d}{k} v \right)^2 \right)^{\beta} dv \tag{16}$$

$$= \frac{2}{C} \int_{-\frac{k}{d}}^{+\frac{k}{d}} \frac{d}{k} \beta v \left( 1 - \left( \frac{d}{k} v \right)^2 \right)^{\beta-1} g^j(v) dv$$

$$= 2\beta \frac{d}{k} \mathbb{E}_{\mu^j \sim \pi^j} \left[ g^j(\mu^j) \mu^j \right].$$

Therefore, we have

$$\mathbb{E}_{\mu \sim \pi} \mathbb{E}_{S_n \sim \mathcal{D}_{\mu,k}^{\otimes n}, \hat{\theta} \sim \mathcal{A}_n(S_n)} \left[ \sum_{i=1}^{n} \phi_\mu(\hat{\theta}, Z_i) \right]$$

$$= \sum_{j=1}^{d} \mathbb{E} \left[ \sum_{i \in \mathcal{I}_j} (\hat{\theta})^j \left( Z_i^j - \frac{d}{k} \mu^j \right) \right]$$

$$= \sum_{j=1}^{d} \mathbb{E} \left[ \mathbb{E} \left[ \sum_{i \in \mathcal{I}_j} (\hat{\theta})^j \left( Z_i^j - \frac{d}{k} \mu^j \right) \middle| \{\mathcal{I}_r\}_{r \in [d]}, \{Z_i^m\}_{m \neq j, i \in [n]} \right] \right]$$

$$= 2\beta \frac{d}{k} \sum_{j=1}^{d} \mathbb{E} \left[ g^j(\mu^j) \cdot \mu^j \right],$$

where the last step follow from Equations (15) and (16). Then, notice that

$$\mathbb{E} \left[ g^j(\mu^j) \cdot \mu^j \right] = \mathbb{E} \left[ \mathbb{E} \left[ (\hat{\theta})^j \cdot \mu^j \middle| \{\mathcal{I}_r\}_{r \in [d]}, \{Z_i^m\}_{m \neq j, i \in [n]} \right] \right]$$

$$= \mathbb{E} \left[ (\hat{\theta})^j \cdot \mu^j \right].$$

Therefore, by the definition of inner product in $\mathbb{R}^d$, we have

$$2\beta \frac{d}{k} \sum_{j=1}^{d} \mathbb{E} \left[ g^j(\mu^j) \cdot \mu^j \right] = 2\beta \frac{d}{k} \mathbb{E} \left[ \left\langle \hat{\theta}, \mu \right\rangle \right],$$

as was to be shown. $\qquad \square$

## D.2    Fingerprinting for $\ell_1$ setup.

Additionally, to prove Theorem 2.6, we will need the following fingerprinting lemma. It can be seen as a generalization of beta-fingerprinting lemma in [SU17] using the scaling matrix technique of [KLSU18].

**Lemma D.1** (Fingerprinting lemma with a scaling matrix). *Fix $d \in \mathbb{N}$. Let $\mathcal{Z} = \{\pm 1\}^d$ and let $\beta > 0$ be arbitrary. Consider arbitrary $0 < \gamma \leq 1$. For every $\mu \in [-\gamma, \gamma]^d$, let $\mathcal{D}_\mu$ be the product distribution on $\mathcal{Z}$ with mean $\mu$, i.e., for every $z \in \mathcal{Z}$, we have $\mathcal{D}_\mu = \prod_{k=1}^{d} \left( \frac{1 + z^k \mu^k}{2} \right)$ let $\Lambda_\mu$ be a diagonal matrix of size $d$ where the $i$-th diagonal element is given by $\Lambda_\mu^{ii} = \frac{1 - (\mu^i/\gamma)^2}{1 - (\mu^i)^2}$, and let $\phi_\mu(\theta, z) = \langle \theta, \Lambda_\mu(z - \mu) \rangle$. Let $\pi = \mathsf{s\text{-}beta}_{[-\gamma, \gamma]} (\beta, \beta)^{\otimes d}$ be a prior. Then, for every algorithm $\mathcal{A}_n : \mathcal{Z}^n \to \mathcal{M}_1(\mathbb{R}^d)$, we have*

$$\mathbb{E}_{\mu \sim \pi} \mathbb{E}_{S_n \sim \mathcal{D}_\mu^{\otimes n}, \hat{\theta} \sim \mathcal{A}_n(S_n)} \sum_{Z \in S_n} \phi_\mu(\hat{\theta}, Z) = \frac{2\beta}{\gamma^2} \mathbb{E}_{\mu \sim \pi} \left\langle \mu, \mathbb{E}_{S_n \sim \mathcal{D}_\mu^{\otimes n}, \hat{\theta} \sim \mathcal{A}_n(S_n)}[\hat{\theta}] \right\rangle.$$

This fingerprinting lemma is handy for the following reason. To ensure the problem is hard to learn, entries of $\mu$ typically need to inversely scale with $\alpha$. To achieve this, one can select small $\gamma$ in the above to shrink the beta-prior to a smaller scale, while simultaneously having the freedom to set $\beta$ to any value. In particular, this allows us to choose $\beta \in \Theta(\log(d))$ in the proof of Theorem 2.6 to leverage the anti-concentration result of [SU17, Prop. 5].

Before we proceed with the proof, we state the necessary lemmata. Throughout this section, for a real number $p \in [-1, 1]$, we will write $X \sim p$ to denote the fact that $X$ is a random variable on $\{\pm 1\}$ with mean $p$. The following is a classical fingerprinting result.

**Lemma D.2** (Lemma 5 of [DSSUV15]). *Let $f : \{\pm 1\}^n \to \mathbb{R}$ be arbitrary. Define $g : [-1, 1] \to \mathbb{R}$ by*

$$g(p) = \mathbb{E}_{X \sim p^{\otimes n}}[f(X)].$$

*Then,*

$$\mathbb{E}_{X \sim p^{\otimes n}}\left[f(X) \sum_{i \in [n]} (X_i - p)\right] = (1 - p^2)g'(p).$$

Armed with the above result, we proceed to the proof of Lemma D.1. We will first prove a *per-coordinate version* of Lemma D.1. We make a note that the proofs combine techniques for beta-fingerprinting results of [SU17] and the scaling matrix technique of [KLSU19; ADHLR24].

**Lemma D.3** (Per-coordinate version of Lemma D.1). *Let* $f : \{\pm 1\}^n \to \mathbb{R}$ *be arbitrary. Let* $\pi = \mathsf{s\text{-}beta}_{[-\gamma,\gamma]}(\beta, \beta)$ *be a prior distribution. Then,*

$$\mathbb{E}_{p \sim \pi} \mathbb{E}_{X \sim p^{\otimes n}}\left[\frac{1 - (p/\gamma)^2}{1 - p^2} f(X) \sum_{i=1}^{n} (X_i - p)\right] = \frac{2\beta}{\gamma^2} \mathbb{E}_{p \sim \pi}\left[p \cdot \mathbb{E}_{X \sim p^{\otimes n}} f(X)\right].$$

*Proof.* Let

$$g(p) = \mathbb{E}_{X \sim p^{\otimes n}}[f(X)].$$

Then, by Lemma D.2, we have for every $p \in [-1, 1]$,

$$\mathbb{E}_{X \sim p^{\otimes n}}\left[\frac{1 - (p/\gamma)^2}{1 - p^2} f(X) \sum_{i \in [n]} (X_i - p)\right] = \left(1 - \left(\frac{p}{\gamma}\right)^2\right)g'(p).$$

Recalling the definition of scaled symmetric beta distribution from Definition A.13, we have

$$\mathbb{E}_{p \sim \pi} \mathbb{E}_{X \sim p^{\otimes n}}\left[\frac{1 - (p/\gamma)^2}{1 - p^2} f(X) \sum_{i \in [n]} (X_i - p)\right]$$

$$= \mathbb{E}_{p \sim \pi}\left[\left(1 - \left(\frac{p}{\gamma}\right)^2\right)g'(p)\right]$$

$$= \frac{1}{\gamma B(\beta)} \int_{-\gamma}^{\gamma} \left(1 - \left(\frac{p}{\gamma}\right)^2\right)^{\beta-1} \cdot \left(1 - \left(\frac{p}{\gamma}\right)^2\right) g'(p) dp$$

$$= \frac{1}{\gamma B(\beta)} \int_{-\gamma}^{\gamma} \left(1 - \left(\frac{p}{\gamma}\right)^2\right)^{\beta} g'(p) dp$$

$$=^{(a)} \frac{1}{\gamma B(\beta)} \left[\left(1 - \left(\frac{p}{\gamma}\right)^2\right)^{\beta} g(p)\Big|_{-\gamma}^{\gamma} - \int_{-\gamma}^{\gamma} \left(\left(1 - \left(\frac{p}{\gamma}\right)^2\right)^{\beta}\right)' g(p) dp\right]$$

$$= \frac{1}{\gamma B(\beta)} \left[\int_{-\gamma}^{\gamma} \left(1 - \left(\frac{p}{\gamma}\right)^2\right)^{\beta-1} \cdot \frac{2\beta p}{\gamma^2} g(p) dp\right]$$

$$= \frac{2\beta}{\gamma^2} \mathbb{E}_{p \sim \pi}\left[p \cdot g(p)\right],$$

where in (a) we used integration by parts. This concludes the proof. □

Applying the above results to each coordinate and summing the equalities gives Lemma D.1.

**Lemma D.1** (Fingerprinting lemma with a scaling matrix). *Fix* $d \in \mathbb{N}$. *Let* $\mathcal{Z} = \{\pm 1\}^d$ *and let* $\beta > 0$ *be arbitrary. Consider arbitrary* $0 < \gamma \leq 1$. *For every* $\mu \in [-\gamma, \gamma]^d$, *let* $\mathcal{D}_\mu$ *be the product distribution on* $\mathcal{Z}$ *with mean* $\mu$, *i.e., for every* $z \in \mathcal{Z}$, *we have* $\mathcal{D}_\mu = \prod_{k=1}^{d} \left(\frac{1 + z^k \mu^k}{2}\right)$ *let* $\Lambda_\mu$ *be a diagonal matrix of size* $d$ *where the i-th diagonal element*

*is given by* $\Lambda_\mu^{ii} = \frac{1-(\mu^i/\gamma)^2}{1-(\mu^i)^2}$, *and let* $\phi_\mu(\theta, z) = \langle \theta, \Lambda_\mu(z-\mu) \rangle$. *Let* $\pi = \mathsf{s\text{-}beta}_{[-\gamma,\gamma]}(\beta, \beta)^{\otimes d}$
*be a prior. Then, for every algorithm* $\mathcal{A}_n : \mathcal{Z}^n \to \mathcal{M}_1(\mathbb{R}^d)$, *we have*

$$\mathbb{E}_{\mu\sim\pi}\mathbb{E}_{S_n\sim\mathcal{D}_\mu^{\otimes n}, \hat{\theta}\sim\mathcal{A}_n(S_n)} \sum_{Z\in S_n} \phi_\mu(\hat{\theta}, Z) = \frac{2\beta}{\gamma^2}\mathbb{E}_{\mu\sim\pi}\left\langle \mu, \mathbb{E}_{S_n\sim\mathcal{D}_\mu^{\otimes n}, \hat{\theta}\sim\mathcal{A}_n(S_n)}[\hat{\theta}]\right\rangle.$$

*Proof.* For a sample $S_n = (Z_1, \ldots, Z_n)$ and $j \in [j]$, we will use $S_n^j \in \mathbb{R}^n$ to denote a vector $(Z_1^j, \ldots, Z_n^j)$ of $j^{\text{th}}$ coordinates. For each coordinate $j \in [d]$, let $f_j \colon \{\pm 1\}^n \to \mathbb{R}$ the function such that

$$f_j(S_n^j) = \mathbb{E}_{\mu\sim\pi}\mathbb{E}_{S_n\sim\mathcal{D}_\mu^{\otimes n}, \hat{\theta}\sim\mathcal{A}_n(S_n)}\left[\hat{\theta}^i \mid S_n^j\right]$$

In other words, $f_j(X)$ is the expected value of $\theta^j$, given that $j^{\text{th}}$ coordinates of samples in $S_n$ are given by $X$. Applying the result of Lemma D.1 to $f_j$, we have

$$\mathbb{E}_{\mu^j\sim\pi^j}\mathbb{E}_{S_n^j\sim(\mu^j)^{\otimes n}}\left[\Lambda_\mu^{jj} \cdot f_j(S_n^j)\sum_{i=1}^n(Z_i^j - \mu^j)\right]$$

$$= \mathbb{E}_{\mu^j\sim\pi^j}\mathbb{E}_{S_n^j\sim(\mu^j)^{\otimes n}}\left[\frac{1-(\mu^j/\gamma)^2}{1-(\mu^j)^2}f_j(S_n^j)\sum_{i=1}^n(Z_i^j - \mu^j)\right]$$

$$= \frac{2\beta}{\gamma^2}\mathbb{E}_{\mu^j\sim\pi^j}\left[\mu^j \cdot \mathbb{E}_{S_n^j\sim(\mu^j)^{\otimes n}}f_j(S_n^j)\right].$$

By the law of total expectation, we get

$$\mathbb{E}_{\mu\sim\pi}\mathbb{E}_{S_n\sim\mathcal{D}_\mu^{\otimes n}, \hat{\theta}\sim\mathcal{A}_n(S_n)}\left[\Lambda_\mu^{jj} \cdot \hat{\theta}^j\sum_{i=1}^n(Z_i^j - \mu^j)\right] = \frac{2\beta}{\gamma^2}\mathbb{E}_{\mu\sim\pi}\left[\mu^j \cdot \mathbb{E}_{S_n\sim\mathcal{D}_\mu^{\otimes n}, \hat{\theta}\sim\mathcal{A}_n(S_n)}\hat{\theta}^j\right].$$

Finally, summing the above over all coordinates $j \in [d]$, we obtain

$$\mathbb{E}_{\mu\sim\pi}\mathbb{E}_{S_n\sim\mathcal{D}_\mu^{\otimes n}, \hat{\theta}\sim\mathcal{A}_n(S_n)}\left[\left\langle\Lambda_\mu\hat{\theta}, \sum_{i=1}^n(Z_i - \mu)\right\rangle\right] = \frac{2\beta}{\gamma^2}\mathbb{E}_{\mu\sim\pi}\left[\left\langle\mu, \mathbb{E}_{S_n\sim\mathcal{D}_\mu^{\otimes n}, \hat{\theta}\sim\mathcal{A}_n(S_n)}\hat{\theta}\right\rangle\right],$$

as desired. □

## E  Hard problem constructions and proofs of subgaussian trace value lower bounds

### E.1  Proofs for $\ell_p$-geometries (Theorem 5.1)

First, recall here the construction of the hard problems $\mathcal{P}_{k,p}$ in Equation (7), parameterized by $k \in [d]$

$$\Theta = \mathcal{B}_\infty(d^{-1/p}), \quad \mathcal{Z} = \{z \in \{0, \pm 1\}^d \colon \|z\|_0 = k\}, \quad f(\theta, z) = -k^{-1/q}\langle\theta, z\rangle. \qquad (\mathcal{P}_{k,p})$$

First, we show in the simple proposition below that $\alpha$-learners for linear problems must agree with the distribution mean.

**Proposition E.1.** *Let* $\mathcal{A}_n$ *be an* $\alpha$-learner for $\mathcal{P}_{k,p}$. *Let* $\mathcal{D} \in \mathcal{M}_1(\mathcal{Z})$ *be a distribution with mean* $\mu = \mathbb{E}_{Z\sim\mathcal{D}}[Z]$. *Then, we have*

$$\mathbb{E}_{S_n\sim\mathcal{D}^{\otimes n}, \hat{\theta}\sim\mathcal{A}_n(S_n)}\left[\left\langle\mu, \hat{\theta}\right\rangle\right] \geq d^{-1/p}\|\mu\|_1 - k^{1/q}\alpha.$$

*Proof.* Since $\mathcal{A}_n$ is an $\alpha$-learner, we have

$$\alpha \geq \mathbb{E}\left[F_\mathcal{D}(\hat{\theta})\right] - \inf_{\theta\in\Theta}F_\mathcal{D}(\theta)$$

$$= \mathbb{E}_{S_n\sim\mathcal{D}^{\otimes n}, \hat{\theta}\sim\mathcal{A}_n(S_n)}\mathbb{E}_{Z\sim\mathcal{D}}\left[f(\hat{\theta}, Z)\right] - \inf_{\theta\in\Theta}\mathbb{E}_{Z\sim\mathcal{D}}f(\theta, Z)$$

$$= k^{-1/q}\left[\sup_{\theta\in\Theta}\mathbb{E}_{Z\sim\mathcal{D}}\langle\theta, Z\rangle - \mathbb{E}_{S_n\sim\mathcal{D}^{\otimes n}, \hat{\theta}\sim\mathcal{A}_n(S_n)}\mathbb{E}_{Z\sim\mathcal{D}}\left\langle\hat{\theta}, Z\right\rangle\right]$$

$$= k^{-1/q}\left[\sup_{\theta\in\Theta}\langle\theta, \mu\rangle - \mathbb{E}_{S_n\sim\mathcal{D}^{\otimes n}, \hat{\theta}\sim\mathcal{A}_n(S_n)}\left\langle\hat{\theta}, \mu\right\rangle\right],$$

which, after rearranging, becomes

$$\mathbb{E}_{S_n \sim \mathcal{D}^{\otimes n}, \hat{\theta} \sim \mathcal{A}_n(S_n)} \left[ \left\langle \mu, \hat{\theta} \right\rangle \right] \geq \sup_{\theta \in \Theta} \langle \mu, \theta \rangle - k^{1/q} \cdot \alpha$$

$$= d^{-1/p} \|\mu\|_1 - k^{1/q}\alpha,$$

where in the last transition we used duality of $\ell_\infty$ and $\ell_1$ norms and the fact that $\Theta = \mathcal{B}_\infty(d^{-1/p})$. This concludes the proof. $\qquad\square$

In the next lemma, we show that every $\alpha$-learner for $\mathcal{P}_{k,p}$ needs to have a large correlation with the training samples in order to achieve small excess risk. The proof is an application of Lemma 4.2 combined with Proposition E.1.

**Lemma E.2.** *Let $\alpha \leq 1/6$, and suppose $k \in [d]$ is such that $k \geq (6\alpha)^p d$. Then, for every $\alpha$-learner $\mathcal{A}_n$ for $\mathcal{P}_{k,p}$, there exists $\mu \in [-k/d, k/d]^d$ and distribution $\mathcal{D} \in \mathcal{M}_1(\mathcal{Z}_k)$ with mean $\mu$ such that the following holds: let*

$$\phi(\theta, Z) := \frac{d^{1/p}}{\sqrt{k}} \left\langle \theta, \left( Z - \frac{d}{k}\mu \right) \right\rangle_{\text{supp}(Z)},$$

*then,*

$$\mathbb{E}_{S_n \sim \mathcal{D}^{\otimes n}, \hat{\theta} \sim \mathcal{A}_n(S_n)} \left[ \sum_{i=1}^n \phi(\hat{\theta}, Z_i) \right] \geq \frac{d^{1-1/p}}{18 k^{1/2-1/p}\alpha}$$

*Proof.* Let

$$\beta = \left( \frac{k^{1/p}}{6 d^{1/p}\alpha} \right)^2 \geq 1,$$

and $\pi = \mathsf{s\text{-}beta}_{[-k/d, k/d]}(\beta, \beta)$. Then, using Corollary A.15, we have

$$\mathbb{E}_{\mu \sim \pi}[\|\mu\|_1] = d\mathbb{E}_{\mu \sim \pi}|\mu^1|$$

$$\geq d \cdot \frac{k/d}{3\sqrt{\beta}}$$

$$= \frac{k}{3(k^{1/p}/6 d^{1/p}\alpha)}$$

$$= 2\alpha d^{1/p} k^{1-1/p}$$

$$= 2\alpha d^{1/p} k^{1/q}.$$

Then, using Lemma 4.2 we have

$$\mathbb{E}_{\mu \sim \pi}\mathbb{E}_{S_n \sim \mathcal{D}_{\mu,k}^{\otimes n}} \sum_{i=1}^n \left\langle \hat{\theta}, \left( Z_i - \frac{d}{k}\mu \right)_{\text{supp}(Z_i)} \right\rangle = \frac{2d\beta}{k} \mathbb{E}_{\mu \sim \pi}\mathbb{E}_{S_n \sim \mathcal{D}_{\mu,k}^{\otimes n}, \hat{\theta} \sim \mathcal{A}_n(S_n)} \left[ \left\langle \mu, \hat{\theta} \right\rangle \right]$$

$$\geq^{(a)} \frac{2d\beta}{k} \mathbb{E}_{\mu \sim \pi} \left[ d^{-1/p} \|\mu\|_1 - k^{1/q}\alpha \right]$$

$$\geq^{(b)} \frac{2d\beta}{k} \left[ d^{-1/p} \cdot 2\alpha d^{1/p} k^{1/q} - k^{1/q}\alpha \right]$$

$$= \frac{2d}{k} \cdot \left( \frac{k^{1/p}}{6 d^{1/p}\alpha} \right)^2 \cdot k^{1/q}\alpha$$

$$= \frac{2d}{k} \cdot \left( \frac{k^{1/p}}{6 d^{1/p}\alpha} \right)^2 \cdot k^{1/q}\alpha$$

$$= \frac{d^{1-2/p} k^{1/p}}{18\alpha}.$$

Since the above holds in expectation over draws of $\mu$, there exists at least one value of $\mu$ for which the above holds; let $\mathcal{D} = \mathcal{D}_{k,\mu}$. Then, letting

$$\phi(\theta, Z) := \frac{d^{1/p}}{\sqrt{k}} \left\langle \theta, \left( Z - \frac{d}{k}\mu \right) \right\rangle_{\text{supp}(Z)},$$

we obtain

$$\mathbb{E}_{S_n \sim \mathcal{D}^{\otimes n}, \hat{\theta} \sim \mathcal{A}_n(S_n)} \left[ \sum_{i=1}^n \phi(\hat{\theta}, Z_i) \right] = \mathbb{E}_{\mu \sim \pi} \mathbb{E}_{S_n \sim \mathcal{D}_{\mu,k}^{\otimes n}} \sum_{i=1}^n \frac{d^{1/p}}{\sqrt{k}} \left\langle \hat{\theta}, \left( Z_i - \frac{d}{k}\mu \right)_{\mathrm{supp}(Z_i)} \right\rangle$$

$$\geq \frac{d^{1-1/p}}{18 k^{1/2-1/p}\alpha},$$

as desired. $\qquad \square$

We now argue that the pair $(\phi, \mathcal{D})$ from the lemma above (with $\phi$ scaled by some constant) constitutes a valid subgaussian tracer. In particular, the lemma below shows that $\{\phi(\theta, Z)\}_{\theta \in \Theta}$ induces a $O(1)$-subgaussian process w.r.t. $(\Theta, \|\cdot\|_\Theta)$ norm.

**Lemma E.3.** *Fix $d \in \mathbb{N}$. Let $\mu \in [-k/d, k/d]^d$ be arbitrary. Let $\phi : \Theta \times \mathcal{Z}_k \to \mathbb{R}$ be as in Lemma E.2. Let $\mathcal{D}_{\mu,k}$ be the data distribution from Definition 4.1 for some $\mu$, and consider $Z \sim \mathcal{D}_{\mu,k}$. Then, $\{\phi(\theta, Z)\}_{\theta \in \Theta}$ is a $C$-subg process w.r.t. to $(\Theta, \|\cdot\|_\Theta)$ for some universal constant $C > 0$.*

*Proof.* Let $J \in \binom{[d]}{k}$ be an arbitrary coordinate subset of size $k$, and, recalling Definition 4.1, let $Z_J$ be a random variable with PMF given by $P_{\mu,k,J}$. Then, $Z$ is a uniform mixture of $\{Z_J\}_{J \in \binom{[d]}{k}}$.

Fix $J \in \binom{[d]}{k}$, and let $\theta_1, \theta_2 \in \Theta$ be two arbitrary points. First, we upper bound a subgaussian norm of $\phi(\theta_1, Z_J) - \phi(\theta_2, Z_J)$. We have

$$\|\phi(\theta_1, Z_J) - \phi(\theta_2, Z_J)\|_{\psi_2} = \frac{d^{1/p}}{\sqrt{k}} \|\langle \theta_1 - \theta_2, Z_J \rangle_J\|_{\psi_2}$$

$$= \frac{d^{1/p}}{\sqrt{k}} \left\| \sum_{j \in J} (\theta_1^j - \theta_2^j) Z_J^j \right\|_{\psi_2}$$

$$\leq^{(a)} \frac{d^{1/p}}{\sqrt{k}} \sqrt{C_1 \sum_{j \in J} \left\| (\theta_1^j - \theta_2^j) Z_J^j \right\|_{\psi_2}^2}$$

$$\leq^{(b)} \frac{d^{1/p}}{\sqrt{k}} \sqrt{C_1 \sum_{j \in J} C_2 |\theta_1^j - \theta_2^j|^2}$$

$$= \frac{d^{1/p}}{\sqrt{k}} \cdot \sqrt{C_1 C_2} \sqrt{k} \|\theta_1 - \theta_2\|_\infty$$

$$=^{(c)} \sqrt{C_1 C_2} \|\theta_1 - \theta_2\|_\Theta,$$

where $C_{1,2} > 0$ are universal constants, in (a) we apply Lemma A.7, in (b) we apply Proposition A.9, and in (c) we use that, since $\Theta = \mathcal{B}_\infty(d^{-1/p})$, we have $\|\cdot\|_\Theta = d^{1/p} \|\cdot\|_\infty$. Thus, letting $C = \sqrt{C_1 C_2}$, we have

$$\|\phi(\theta_1, Z_J) - \phi(\theta_2, Z_J)\|_{\psi_2} \leq C \|\theta_1 - \theta_2\|_\Theta.$$

Now, note that $\phi(\theta_1, Z) - \phi(\theta_2, Z)$ has the same distribution as a uniform mixture of $\{\phi(\theta_1, Z_J) - \phi(\theta_2, Z_J)\}_{J \in \binom{[d]}{k}}$. Then, by Proposition A.8, we also have

$$\|\phi(\theta_1, Z) - \phi(\theta_2, Z)\|_{\psi_2} \leq C \|\theta_1 - \theta_2\|_\Theta,$$

which satisfies the first condition in Definition 3.1. Finally, by plugging $\theta_2 = 0$ into the above, we have

$$\|\phi(\theta_1, Z)\|_{\psi_2} \leq C \|\theta_1\|_\Theta \leq C,$$

which satisfies the second condition in Definition 3.1. Thus, $\{\phi(\theta, Z)\}_{\theta \in \Theta}$ is a $C$-subgaussian process w.r.t. $(\Theta, \|\cdot\|_\Theta)$, as desired. $\qquad \square$

Finally, we lower bound the subgaussian trace value of $\mathcal{P}_{k,p}$.

**Lemma E.4.** *Let $\alpha \leq 1/6$ and $d \in \mathbb{N}$ be arbitrary, and let $k \in [d]$ be such that $k \leq (6\alpha)^p d$. Let $\mathcal{P}_{k,p}$ be as in Equation* (7). *Then, the following subgaussian trace value lower bounds hold for every $p \in [1, \infty)$ and some $\kappa \leq c_1\sqrt{d}$,*

$$\mathrm{Tr}_\kappa(\mathcal{P}_{k,p}; n, \alpha) \geq c_2 \frac{d^{1-1/p}}{k^{1/2-1/p}n\alpha}$$

*where $c_{1,2} > 0$ are universal constants.*

*Proof.* Let $C > 0$ be the constant from Lemma E.3, and $\phi$ be as in Lemma E.2. Then, $\{\phi(\theta, Z)/C\}_{\theta \in \Theta}$ is a 1-subgaussian process w.r.t. $(\Theta, \|\cdot\|_\Theta)$. Moreover, $\phi(\theta, Z)/C \leq \sqrt{k}/C \leq \sqrt{d}/C$. Then, letting $\kappa = \sqrt{d}/C$ and using Lemma E.2, the subgaussian trace value of $\mathcal{P}_{k,p}$ can be lower bounded by

$$\mathrm{Tr}_\kappa(\mathcal{P}_{k,p}; n, \alpha) \geq \frac{1}{C}\mathbb{E}_{S_n \sim \mathcal{D}^{\otimes n}, \hat\theta \sim \mathcal{A}_n(S_n)} \frac{1}{n}\left[\sum_{i=1}^n \phi(\hat\theta, Z_i)\right] \geq \frac{d^{1-1/p}}{18C \cdot k^{1/2-1/p}n\alpha},$$

as desired. $\square$

Now we are ready to prove Theorem 5.1.

**Theorem 5.1.** *Let $\mathcal{P}_{k,p}$ be the family of problems described in Equation* (7). *There exist universal constants $c_1, c_2 > 0$ such that, for all $\alpha \in (0, 1/6]$ and $d \in \mathbb{N}$, the following subgaussian trace value lower bounds hold for all $p \in [1, \infty)$ and $\kappa \leq c_1\sqrt{d}$:*

*(i) For $p \leq 2$ and $k = d$, we have $\mathrm{Tr}_\kappa(\mathcal{P}_{k,p}; n, \alpha) \geq c_2 \frac{\sqrt{d}}{n\alpha}$.*

*(ii) For $p \geq 2$ and $k = (6\alpha)^p d \vee 1$, we have $\mathrm{Tr}_\kappa(\mathcal{P}_{k,p}; n, \alpha) \geq c_2 \left[\frac{\sqrt{d}}{n(6\alpha)^{p/2}} \wedge \frac{d^{1-1/p}}{n\alpha}\right]$.*

*Proof.* The theorem is a direct consequence of Lemma E.4. For $p \leq 2$, plug in $k = d$ into the statement of Lemma E.4. We obtain

$$\mathrm{Tr}_\kappa(\mathcal{P}_{k,p}; n, \alpha) \geq c_2 \frac{d^{1-1/p}}{k^{1/2-1/p}n\alpha} = c_2 \frac{\sqrt{d}}{n\alpha}.$$

For $p \geq 2$, plug in $k = (6\alpha)^p \vee 1$. We obtain,

$$\begin{aligned}
\mathrm{Tr}_\kappa(\mathcal{P}_{k,p}; n, \alpha) &\geq c_2 \frac{d^{1-1/p}}{k^{1/2-1/p}n\alpha} \\
&= c_2 \left[\frac{d^{1-1/p}}{n\alpha} \wedge \frac{\sqrt{d}}{(6\alpha)^{p(1/2-1/p)}n\alpha}\right] \\
&= c_2 \left[\frac{d^{1-1/p}}{n\alpha} \wedge \frac{\sqrt{d}}{(6\alpha)^{p/2}n}\right],
\end{aligned}$$

as desired. $\square$

## E.2 Proofs for $\ell_1$-geometry

### E.2.1 Intuition

**Refinement for $p = 1$.** While the above construction also yields a traceability result for $p = 1$, it is suboptimal for the following simple reason: for $k = d$, the problem in Equation (7) only requires $\Theta(1/\alpha^2)$ samples to learn, thus, it is impossible to trace out $\Omega(\log(d)/\alpha^2)$ samples. On the other hand, the problem in Equation (6) requires $\Theta(\log(d)/\alpha^2)$ samples to learn but is not traceable. The intuition we follow here is to modify the construction in Equation (7) to make $\Theta$ "look" more like an $\ell_1$-ball to drive up the sample complexity while still avoiding the counterexample with an ERM learner from the beginning of the section. In particular, we consider the following $\ell_1$-problem,

$$\Theta = \mathcal{B}_1(1) \cap \mathcal{B}_\infty(1/s), \quad \mathcal{Z} = \{\pm 1\}^d, \quad f(\theta, z) = -\langle z, \theta\rangle, \tag{17}$$

for a suitably chosen $s \in [d]$. Note that, if we choose $s \gg 1$, $\Theta$ above is a polytope with much more vertices $(2^s\binom{d}{s})$ than an $\ell_1$ ball $(2d)$, which would intuitively force a learner like an ERM to reveal more information about the training sample. On a technical level, selecting large $s$ improves the subgaussian constant of a tracer; however, selecting $s$ that is too large shrinks the diameter of the set, and thus, the problem becomes easier to learn. We must trade off these two aspects, and carefully set the value of $s$. As it turns out, the optimal choice is $s \propto d^{1-c}$ for an arbitrary small $c > 0$ in order to establish Theorem 2.6. The remainder of the proof is rather technical and hence is deferred to Appendix F.2.

### E.2.2 Formal Proof

For technical reasons, we will need the following refinement of Lemma 3.4 for the special case when the function $\phi$ is convex and $\Theta$ is a polytope. In the proof, we use Lemma C.1.

**Lemma E.5.** *Fix $n, d \in \mathbb{N}$. Suppose $\Theta \subset \mathbb{R}^d$ is (i) a subset of a unit ball in some norm $\|\cdot\|$, and (ii) $\Theta$ is a polytope with $N$ vertices. Let $\phi \colon \Theta \times \mathcal{Z} \to \mathbb{R}$ be a measurable function that is convex in its first argument. Let $\mathcal{D} \in \mathcal{M}_1(\mathcal{Z})$ be such that $\phi(\theta, Z)$ is a $\sigma$-subgaussian process w.r.t. $(\Theta, \|\cdot\|)$. Let $(Z_1, \ldots, Z_n) \sim \mathcal{D}^{\otimes n}$. Then, for every $t \geq 0$,*

$$\Pr\left( \sup_{\theta \in \Theta} \sqrt{\sum_{i=1}^{n} [\phi(\theta, Z_i)]^2} \leq C\sigma \left[ \sqrt{n} + \sqrt{\log(N)} + t \right] \right) \geq 1 - 2\exp(-t^2)$$

*where $C > 0$ is some universal constant.*

*Proof.* Similarly to the proof of Lemma C.1, let $\Phi_\theta$ denote the following random vector

$$\Phi_\theta = \begin{bmatrix} \phi(\theta, Z_1) \\ \vdots \\ \phi(\theta, Z_n) \end{bmatrix}.$$

Then, observe that, the desired quantity is equal to

$$\sup_{\theta \in \Theta} \sqrt{\sum_{i=1}^{n} [\phi(\theta, Z_i)]^2} = \sup_{\theta \in \Theta} \|\Phi_\theta\|_2 = \sup_{\theta \in \Theta, x \in \mathbb{S}^{n-1}} \langle x, \Phi_\theta \rangle.$$

Let $V$ be the set of vertices of $\Theta$ with $|V| \leq N$. Since $\phi$ is convex in its first argument, the supremum above is attained in one of the vertices of $\Theta$. Thus,

$$\sup_{\theta \in \Theta} \sqrt{\sum_{i=1}^{n} [\phi(\theta, Z_i)]^2} = \sup_{\theta \in V, x \in \mathbb{S}^{n-1}} \langle x, \Phi_\theta \rangle = \sup_{\theta \in V} \sqrt{\sum_{i=1}^{n} [\phi(\theta, Z_i)]^2}.$$

Thus, we may apply Lemma C.1 to $V$ instead of $\Theta$. Trivially, we have

$$\mathcal{N}(V, \|\cdot\|, \varepsilon) \leq |V| = N.$$

Then, we have with probability $1 - 2\exp(-t^2)$,

$$\sup_{\theta \in \Theta} \sqrt{\sum_{i=1}^{n} [\phi(\theta, Z_i)]^2} \leq C\sigma \left( \sqrt{n} + \int_0^1 \sqrt{\log \mathcal{N}(\Theta; \|\cdot\|, \varepsilon)} d\varepsilon + t \right)$$

$$\leq C\sigma \left( \sqrt{n} + \sqrt{\log(N)} + t \right),$$

as desired. $\square$

Intuitively, in the special case when $\Theta$ is a polytope, the log-number of vertices becomes "effective dimension" instead of $d$, due to the fact that $\phi$ satisfies the convexity requirement. In some cases, we can have $d \gg \log(N)$, in which the above gives a tighter concentration. In particular, this is a case in our construction for $\ell_1$ geometry in Equation (17). With the above result, we can also establish the following refinement of Theorem 3.5.

**Theorem E.6.** *Fix* $n \in \mathbb{N}$, $d \in \mathbb{N}$, $\kappa > 0$ *and* $\alpha \in [0,1]$. *Consider an arbitrary SCO problem* $\mathcal{P} = (\Theta, \mathcal{Z}, f)$, *and suppose* $\Theta$ *is a polytope with* $N > 0$ *vertices. Let* $T$ *be defined as,*

$$T := \mathrm{Tr}_\kappa(\mathcal{P}; n, \alpha).$$

*Then, for some constant* $c > 0$, *every* $\alpha$-*learner* $\mathcal{A}_n$ *is* $(\xi, m)$-*traceable with*

$$\xi = \exp(-cT^2), \quad m = c\left[\frac{n^2 T^2}{n + \log(N)} - \frac{16\kappa^2 n}{\exp(n + \log(N))}\right].$$

*Proof.* The proof is identical to Theorem 3.5, but using Lemma E.5 instead of Lemma 3.4, and thus, replacing $d$ with $\log(N)$ everywhere. We omit the details. $\qquad\square$

Now, recall the construction from Equation (17),

$$\mathcal{Z} = \{\pm 1\}^d, \quad \Theta = \mathcal{B}_1(1) \cap \mathcal{B}_\infty(1/s), \quad f(\theta, Z) = -\langle \theta, Z \rangle. \tag{18}$$

It is easy to see that $f(\cdot, Z)$ above is 1-Lipschitz w.r.t. $\ell_1$ as $\mathcal{Z} \subset \mathcal{B}_\infty(1)$. We have the following claim.

**Lemma E.7.** *Let* $\mathcal{P}_s$ *be as in* (18), $n \in \mathbb{N}$ *and* $1/8 > \alpha > 0$. *Then,*

$$\mathrm{Tr}_\kappa(\mathcal{P}_s; n, \alpha) \geq \frac{c\sqrt{s}\log\left(\frac{d}{16(s \vee 14)}\right)}{n\alpha},$$

*where* $\kappa \leq c'\sqrt{s}$ *for some constants* $c, c' > 0$

*Proof.* We aim to use Lemma D.1 to characterize the subgaussian trace value. Consider the construction of the prior in Lemma D.1 with the following parameters: $\gamma = 8\alpha \leq 1$ and $\beta = 1 + \frac{1}{2}\log\left(\frac{d}{16(s \vee 14)}\right)$. Then, by combining Lemma D.1 with and Proposition E.1, there exist a prior $\pi$ and a family $\{\Lambda_\mu\}$ of diagonal matrices with non-negative diagonal entries bounded by 1 from above, such that

$$\mathbb{E}_{\mu \sim \pi}\mathbb{E}_{Z \sim \mu^{\otimes n}}\left\langle \hat{\theta}, \sum_{i=1}^n \Lambda_\mu(Z_i - \mu)\right\rangle = \frac{2\beta}{\gamma^2}\mathbb{E}_{\mu \sim \pi}\left[\sup_\theta \langle \theta, \mu \rangle - \alpha\right]$$

$$\geq \frac{2\beta}{\gamma^2}\mathbb{E}_{\mu \sim \pi}\left[\frac{1}{s}\sup_{\substack{I \subset [d] \\ |I| = s}}\sum_{i \in I}|\mu_i| - \alpha\right]$$

$$\geq^{(a)} \frac{\beta}{\gamma^2}\left(\frac{\gamma}{2} - 2\alpha\right)$$

$$\geq \frac{\log\left(\frac{d}{16(s \vee 14)}\right)}{32\alpha^2} \cdot 2\alpha$$

$$\geq \frac{\log\left(\frac{d}{16(s \vee 14)}\right)}{16\alpha},$$

where in (a) we used Proposition 5 of [SU17]. Since this holds in expectation over $\mu$, it holds for at least one choice of $\mu$. Let $\mu$ be that value. Now, let $\phi(\theta, Z) = C^{-1/2}\sqrt{s}\left\langle \hat{\theta}, \Lambda(Z - \mu)\right\rangle$, where $C$ is the absolute constant from Lemma A.7. For all $\theta, \theta' \in \Theta$, we have

$$\|\phi(\theta, Z) - \phi(\theta', Z)\|_{\psi_2} \leq C^{-1/2}\sqrt{s}\|\langle \theta' - \theta, \Lambda(Z - \mu)\rangle\|_{\psi_2}$$

$$\leq^{(a)} C^{-1/2}\sqrt{s}\|\theta' - \theta\|_2 \cdot C^{1/2}\max_i\|\Lambda^i(Z^i - \mu^i)\|_{\psi_2}$$

$$\leq \sqrt{s}\|\theta' - \theta\|_2$$

$$\leq^{(b)} \|\theta' - \theta\|_\Theta,$$

where in (a) we apply Lemma A.7, and in (b) we use that for every $\theta \in \Theta$, we have $\|\theta\|_2 \leq 1/\sqrt{s}$, thus, $\sqrt{s}\|\cdot\|_2 \leq \|\cdot\|_\Theta$. Plugging $\theta' = 0$ gives

$$\|\phi(\theta, Z)\|_{\psi_2} \leq \|\theta\|_\Theta \leq 1.$$

Thus, $\{\phi(\theta, Z)\}_{\theta \in \Theta}$ is a 1-subg process w.r.t. $(\Theta, \|\cdot\|_\Theta)$. Finally, we have

$$|\phi(\theta, Z)| \leq C^{-1/2}\sqrt{s}$$

Therefore, setting $\kappa = C^{-1/2}\sqrt{s}$ and noting that $\phi$ is linear (and therefore convex) in its first argument, we have

$$\mathrm{Tr}_\kappa(\mathcal{P}_s; n, \alpha) \geq \mathbb{E}_{\mu \sim \pi} \mathbb{E}_{Z \sim \mu^{\otimes n}} \sum_{i=1}^n \frac{1}{n} \phi(\theta, Z_i)$$

$$\geq \frac{C^{-1/2}\sqrt{s} \log\left(\frac{d}{16(s \vee 14)}\right)}{n\alpha},$$

as desired.  $\square$

## F  Proofs of the main results (Section 2.2)

### F.1  Proof of Theorem 2.5

**Theorem 2.5.** *There exists a universal constant $c > 0$ such that, for all $p \in [1, 2)$, if $d$, $n$, $\xi \in (0, 1/e)$, and $\alpha > 0$ are such that*

$$\frac{c}{\sqrt{n}} \leq \alpha \leq \min\left\{c \cdot \sqrt{\frac{d}{n^2 \log(1/\xi)}}, \frac{1}{6}\right\}, \tag{3}$$

*then there exist an $\ell_p$ SCO problem that every $\alpha$-learner is $(\xi, m)$-traceable with $m \in \Omega\left(\alpha^{-2}\right)$.*

*Proof.* We begin by noting that the interval for $\alpha$ in Equation (3) is non-empty only if

$$\frac{c}{\sqrt{n}} \leq \min\left\{c \cdot \sqrt{\frac{d}{n^2 \log(1/\xi)}}, \frac{1}{6}\right\} \leq c \cdot \sqrt{\frac{d}{n^2}},$$

where we used $\xi < 1/e$ in the last transition. Via straightforward algebra, the above implies $d \geq n$, thus, we may without loss of generality assume $d \geq n$ in the remainder of the proof.

Let $\mathcal{P}_{k,p}$ be as in Equation (7), and set $k$ as in Theorem 5.1. Then, Theorem 5.1 gives the following lower bound on the subgaussian trace value in this case:

$$T := \mathrm{Tr}_\kappa(\mathcal{P}_{k,p}; n, \alpha) \geq c_2 \frac{\sqrt{d}}{n\alpha} \geq \frac{c_2}{c}\sqrt{\log\left(\frac{1}{\xi}\right)},$$

for some $\kappa \leq c_1\sqrt{d}$. Thus, provided $c$ is small enough ($c \leq c_2$), by Theorem 3.5, every $\alpha$-learner is $(\xi, m)$-traceable with $m$ satisfying, for some universal constant $c' > 0$,

$$m \geq c'\left[\frac{n^2 T^2}{n+d} - \frac{16\kappa^2 n}{\exp(n+d)}\right]$$

$$\geq c'\left[\frac{c_2^2 d}{n+d} \cdot \frac{1}{\alpha^2} - \frac{16c_1^2 dn}{\exp(n+d)}\right]$$

$$\geq c'\left[\frac{c_2^2}{2} \cdot \frac{1}{\alpha^2} - \frac{16c_1^2 d^2}{\exp(d)}\right], \tag{19}$$

where we used $d \geq n$ in the last inequality. Then, we have

$$m \in \Omega\left(\frac{1}{\alpha^2}\right),$$

as desired.

$\square$

## F.2 Proof of Theorem 2.6

Then, the proof of Theorem 2.6 follows.

**Theorem 2.6.** *There exists a universal constant $c > 0$ such that, if $d$ is large enough and $n$, $\xi \in (0, 1/e)$, and $\alpha > 0$ are such that*

$$c \cdot \sqrt{\frac{\log(d)}{n}} \le \alpha \le \min\left\{ c \cdot \frac{d^{0.49}}{n\sqrt{\log(1/\xi)}}, \frac{1}{8} \right\}, \textit{ then} \tag{4}$$

*there exists a $\ell_1$ SCO problem that every $\alpha$-learner is $(\xi, m)$-traceable with $m \in \Omega\left(\log(d)/\alpha^2\right)$.*

*Proof.* We begin by noting that the interval for $\alpha$ in Equation (4) is non-empty only if

$$c \cdot \sqrt{\frac{\log(d)}{n}} \le c \cdot \frac{d^{0.49}}{n\sqrt{\log(1/\xi)}} \le c \cdot \frac{d^{0.49}}{n}$$

where we used $\xi < 1/e$ and $c < 1/6$ in the last transition. Via straightforward algebra, the above implies $d^{0.98}/\log(d) \ge n$, thus, we may without loss of generality assume $d^{0.98}/\log(d) \ge n$ in the remainder of the proof.

Let $s = d^{0.98}$. Note that, letting $V$ be the set of vertices of the polytope $\Theta = \mathcal{B}_1(1) \cap \mathcal{B}_\infty(1/s)$, we have

$$V = \left\{ z \in \left\{ 0, \pm\frac{1}{s} \right\} : \|z\|_0 = s \right\}.$$

The proof of this fact is straightforward and it is based on showing that every point in $\Theta$ can be written as a convex combination of the points in $V$. Thus,

$$\log|V| = \log\left( 2^s \binom{d}{s} \right) \le s\log(de/s) + s = s\log(de^2/s).$$

By the choice of $s$ and using Lemma E.7, we have, for some $\kappa \le c_1\sqrt{s}$,

$$T := \mathrm{Tr}_\kappa(\mathcal{P}_s; n, \alpha) \ge c_2 \frac{d^{0.49}\log\left(\frac{d}{16(s \vee 14)}\right)}{n\alpha}$$

$$\ge^{(a)} c_2 \frac{d^{0.49}\log\left(\frac{d}{16s}\right)}{n\alpha}$$

$$= c_2 \frac{d^{0.49}\log\left(\frac{d^{0.02}}{16}\right)}{n\alpha}$$

$$= c_2 \frac{d^{0.49}(0.02 \cdot \log(d) - \log(16))}{n\alpha}$$

$$\ge^{(b)} c_2 \frac{d^{0.49} \cdot 0.01 \cdot \log(d)}{n\alpha}$$

$$\ge^{(c)} \sqrt{\log(1/\xi)},$$

where (a) and (b) hold provided $d$ (and thus $s$) is large enough, and (c) holds provided $c > 0$ in Equation (4) is small enough. By Theorem E.6, every $\alpha$-learner is $(\xi, m)$-traceable, where

$$m \ge c'\left[ \frac{n^2 T^2}{n + \log|V|} - \frac{16\kappa^2 n}{\exp(n + \log|V|)} \right]$$

$$\ge c'\left[ \frac{0.01^2 c_2^2 \cdot d^{0.98}\log(d)}{n + \log|V|} \cdot \frac{\log(d)}{\alpha^2} - \frac{16c_1^2 \cdot sn}{\exp(n + \log|V|)} \right].$$

Recall that $d^{0.98} \ge n\log(d) \ge n$ (for $d \ge 3$). Moreover, $\log|V| \le s\log(de^2/s) = d^{0.98}\log(e^2 d^{0.02}) \le Cd^{0.98}\log(d)$, for some universal $C > 0$. Thus, for $d$ large enough,

$$m \in \Omega\left( \frac{\log(d)}{\alpha^2} \right),$$

as desired. $\square$

### F.3 Proof of Theorem 2.7

**Theorem 2.7.** *There exists a universal constant $c > 0$ such that, for all $p \in [2, \infty)$, if $d$, $n$, $\xi \in (0, 1/e)$, and $\alpha > 0$ are such that*

$$\frac{1}{6} \cdot \min\left\{\frac{1}{n^{1/p}}, \frac{d^{\frac{1}{2}-\frac{1}{p}}}{\sqrt{n}}\right\} \leq \alpha \leq \min\left\{c \cdot \left(\frac{d}{n^2 \log(1/\xi)}\right)^{1/p}, \frac{1}{6}\right\}, then \qquad (5)$$

*there exist an $\ell_p$ SCO problem such that every $\alpha$-learner is $(\xi, m)$-traceable with $m \in \Omega\left(1/(6\alpha)^p\right)$.*

*Proof.* Throughout, we assume $c$ is a sufficiently small constant. Assume $c < 1/6$. Then, we begin by noting that the interval for $\alpha$ in Equation (5) is non-empty only if

$$\frac{1}{6} \cdot \frac{1}{n^{1/p}} \leq c \cdot \left(\frac{d}{n^2 \log(1/\xi)}\right)^{1/p} \leq \frac{1}{6} \cdot \left(\frac{d}{n^2}\right)^{1/p},$$

where we used $\xi < 1/e$ and $c < 1/6$ in the last transition. Via straightforward algebra, the above implies $d \geq n$, thus, we may without loss of generality assume $d \geq n$ in the remainder of the proof.

Let $\mathcal{P}_{k,p}$ be as in Equation (7), and set $k$ as in Theorem 5.1. Then, Theorem 5.1 gives the following lower bound on the subgaussian trace value

$$T := \mathrm{Tr}_\kappa(\mathcal{P}_{k,p}; n, \alpha) \geq c_2 \left[\frac{d^{1-1/p}}{n\alpha} \wedge \frac{\sqrt{d}}{(6\alpha)^{p/2}n}\right] \qquad (20)$$

for some $\kappa \leq c_1\sqrt{d}$. Note that, from (5), we have

$$\alpha \geq \frac{1}{6} \cdot \frac{1}{n^{1/p}} \geq \frac{1}{6} \cdot \frac{1}{d^{1/p}}$$

Now, note that, the minimum in Equation (20) is achieved in the second term iff $\alpha \geq d^{-1/p}/6$. Then, the lower bound on the subgaussian trace value becomes

$$T \geq c_2 \frac{\sqrt{d}}{(6\alpha)^{p/2}n} \geq \frac{c_2}{(6c)^{p/2}} \cdot \sqrt{\log(1/\xi)} \geq \sqrt{\log(1/\xi)}$$

where the second transition follows from Equation (5) and the third transition holds whenever $c > 0$ is small enough (e.g., when $c \leq c_2^{2/p}/6$). Then, by Theorem 3.5 every $\alpha$-learner is $(\xi, m)$-traceable with $m$ satisfying, for some universal constant $c' > 0$,

$$
\begin{aligned}
m &\geq c' \left[\frac{n^2 T^2}{n+d} - \frac{16\kappa^2 n}{\exp(n+d)}\right] \\
&\geq c' \left[\frac{c_2^2 d}{n+d} \cdot \frac{1}{(6\alpha)^p} - \frac{16c_1^2 dn}{\exp(n+d)}\right] \\
&\geq c' \left[\frac{c_2^2}{2} \cdot \frac{1}{(6\alpha)^p} - \frac{16c_1^2 d^2}{\exp(d)}\right], \qquad (21)
\end{aligned}
$$

where we used $d \geq n$ in the last transition. Thus,

$$m \in \Omega\left(\frac{1}{(6\alpha)^p}\right),$$

as desired. $\qquad\square$

### F.4 Proof of Theorem 2.8

**Theorem 2.8.** *Let $p \in [2, \infty)$. There exist a universal constant $c > 0$ and an $\ell_p$ SCO problem $\mathcal{P} = (\Theta, \mathcal{Z}, f)$ such that every $(\varepsilon, \delta)$-DP learner of $\mathcal{P}$ with $\varepsilon \leq 1$ and $\delta \leq c/n$ satisfies,*

$$\alpha \geq c \cdot \min\left\{\left(\frac{d}{\varepsilon^2 n^2}\right)^{\frac{1}{p}}, \frac{d^{1-1/p}}{\varepsilon n}, 1\right\}.$$

*Proof.* By Theorem 5.1, for some problem $\mathcal{P}$, we have

$$T := \mathrm{Tr}_\kappa\left(\mathcal{P}; n, \alpha\right) \geq c'\left[\frac{d^{1-1/p}}{n\alpha} \wedge \frac{\sqrt{d}}{n\alpha^{p/2}}\right]$$

Then Theorem 3.6 implies

$$\exp(\varepsilon) - 1 \geq c''\left[T - 2\delta\kappa\right],$$

Note that for all $\varepsilon \leq 1$, we have $2\varepsilon \geq \exp(\varepsilon) - 1$. Thus,

$$2\varepsilon \geq c'\left[T - 2\delta\kappa\right].$$

which implies,

$$2(\varepsilon/c'' + \delta\kappa) \geq T \geq c'\left[\frac{d^{1-1/p}}{n\alpha} \wedge \frac{\sqrt{d}}{n\alpha^{p/2}}\right].$$

Then, for some $C > 0$,

$$C(\varepsilon \vee \delta\kappa) \geq \frac{d^{1-1/p}}{n\alpha} \wedge \frac{\sqrt{d}}{n\alpha^{p/2}}$$

Rearranging gives

$$\alpha \geq \frac{d^{1-1/p}}{Cn(\varepsilon \vee \delta\kappa)} \wedge \left(\frac{\sqrt{d}}{Cn(\varepsilon \vee \delta\kappa)}\right)^{2/p}$$

Note that if $\varepsilon \geq \delta\kappa$, the desired bound is immediate. For $\delta\kappa \geq \varepsilon$, we have, since $\delta \leq c/n$ and $\kappa \leq c'\sqrt{d}$,

$$\alpha \geq \frac{d^{1-1/p}}{C'\sqrt{d}} \wedge \left(\frac{\sqrt{d}}{C'\sqrt{d}}\right)^{2/p} \geq (C')^{-2/p},$$

for some $C' > 0$, as desired. $\qquad\square$

## F.5   Proof of Corollary 2.9

**Corollary 2.9.** *Let $\mathcal{Z} = \{\pm 1\}^d$, and suppose an estimator is given such that, given access to i.i.d. samples $Z_1, \ldots, Z_n \in \mathcal{Z}$, outputs $\hat{\mu}$ with $\mathbb{E}\left\|\hat{\mu} - \mathbb{E}[Z_1]\right\|_\infty \leq \alpha/2$. Then, there exists a universal constant $c > 0$ such that, if $d$ is large enough and $n$, $\xi \in (0, 1/e)$, and $\alpha > 0$ satisfy Equation (4), then the estimator $\hat{\mu}$ is $(\xi, m)$-traceable with $m \in \Omega\left(\log(d)/\alpha^2\right)$.*

*Proof.* We will first show that we can use the mean estimation algorithm to solve the corresponding hard problem for $\ell_1$-SCO. Specifically, consider the SCO problem as in Equation (17), and define the following learning algorithm based on the mean estimation. Let $\hat{\mu}$ be the output of mean estimator based on the samples $Z_1, \ldots, Z_n$, and let

$$\hat{\theta} := \arg\max_{\theta \in \Theta} \langle\theta, \hat{\mu}\rangle.$$

Let $\theta^\star$ be the population risk minimizer, and let $\mu$ be the true mean, that is, $\mu = \mathbb{E}[Z_1]$. Then, the excess risk of $\hat{\theta}$ can be upper bounded as:

$$\begin{aligned}
\langle\theta^\star, \mu\rangle - \left\langle\hat{\theta}, \mu\right\rangle &= \left\langle\theta^\star - \hat{\theta}, \mu\right\rangle \\
&= \left\langle\theta^\star - \hat{\theta}, \mu - \hat{\mu}\right\rangle + \left\langle\theta^\star - \hat{\theta}, \hat{\mu}\right\rangle \\
&\leq \left\|\theta^\star - \hat{\theta}\right\|_1 \|\mu - \hat{\mu}\|_\infty + \left\langle\theta^\star - \hat{\theta}, \hat{\mu}\right\rangle \\
&\leq 2\|\mu - \hat{\mu}\|_\infty + \left\langle\theta^\star - \hat{\theta}, \hat{\mu}\right\rangle,
\end{aligned}$$

where the last transition follows since $\hat{\theta}, \theta^\star$ both lie inside $\Theta \subset \mathcal{B}_1$. Now, by the choice of $\hat{\theta}$, the second term is non-positive. Thus,

$$\langle\theta^\star, \mu\rangle - \left\langle\hat{\theta}, \mu\right\rangle \leq 2\|\mu - \hat{\mu}\|_\infty.$$

Taking expectations on both sides, we get

$$\mathbb{E}\langle\theta^\star, \mu\rangle - \left\langle\hat{\theta}, \mu\right\rangle \leq 2\mathbb{E}\|\mu - \hat{\mu}\|_\infty \leq \alpha.$$

Applying the result of Theorem 2.6 to $\hat{\theta}$, we conclude the proof. $\qquad\square$

# G Connection between subgaussian trace value and non-private sample complexity

From the main part of the paper, we observe that the subgaussian trace value is typically inversely proportional to $\alpha$. We start by proving two innocuous results (Propositions G.1 and G.3) that establish an absolute upper bound on subgaussian trace value. It will then allow us to extract lower bounds on $\alpha$ by plugging in our lower bounds on subgaussian trace value (Theorems G.2 and G.4). We start with the $p \in (1, \infty)$ case, and then consider $p = 1$.

## G.1 Lower bounds for $p \in (1, \infty)$

**Proposition G.1.** *Fix $n \in \mathbb{N}$, $d \in \mathbb{N}$, and $\alpha \in [0,1]$. Consider an arbitrary SCO problem $\mathcal{P} = (\Theta, \mathcal{Z}, f)$ in $\mathbb{R}^d$. Let $\mathrm{Tr}_\kappa(\mathcal{P}; n, \alpha)$ be the subgaussian trace value of problem $\mathcal{P}$. Then, we have*

$$\mathrm{Tr}_\kappa(\mathcal{P}; n, \alpha) \cdot \sqrt{n} \leq C \cdot \sqrt{d},$$

*for some universal constant $C > 0$.*

*Proof.* Let $(\phi, \mathcal{D})$ be an arbitrary subgaussian tracer. Consider the process $\{X_\theta\}_{\theta \in \Theta}$ defined as

$$X_\theta := \frac{1}{n} \sum_{i=1}^{n} \phi(\theta, Z_i),$$

where $S_n = (Z_1, \ldots, Z_n) \sim \mathcal{D}^{\otimes n}$. We will argue that $\{X_\theta\}_{\theta \in \Theta}$ is $O(1/\sqrt{n})$-subgaussian process w.r.t. $(\Theta, \|\cdot\|_\Theta)$. First, consider arbitrary $\theta_1, \theta_2 \in \Theta$. We have

$$
\begin{aligned}
\|X_{\theta_1} - X_{\theta_2}\|_{\psi_2} &= \frac{1}{n} \left\| \sum_{i=1}^{n} \left( \phi(Z_i, \theta_1) - \phi(Z_i, \theta_2) \right) \right\|_{\psi_2} \\
&\leq^{(a)} \frac{C}{n} \sqrt{ \sum_{i=1}^{n} \|\phi(Z_i, \theta_1) - \phi(Z_i, \theta_2)\|_{\psi_2}^2 } \\
&\leq^{(b)} \frac{C}{n} \sqrt{ \sum_{i=1}^{n} \|\theta_1 - \theta_2\|_\Theta } \\
&= \frac{C}{\sqrt{n}} \|\theta_1 - \theta_2\|_\Theta,
\end{aligned}
$$

where in (a) we applied Lemma A.7, and in (b) we used the fact that $\{\phi(\theta, Z_i)\}_{\theta \in \Theta}$ is 1-subgaussian process w.r.t. $(\Theta, \|\cdot\|_\Theta)$ for every $i \in [n]$. Moreover, for every $\theta \in \Theta$, we similarly have

$$
\begin{aligned}
\|X_\theta\|_{\psi_2} &= \frac{1}{n} \left\| \sum_{i=1}^{n} \phi(Z_i, \theta) \right\|_{\psi_2} \\
&\leq^{(a)} \frac{C}{n} \sqrt{ \sum_{i=1}^{n} \|\phi(Z_i, \theta)\|_{\psi_2}^2 } \\
&\leq^{(b)} \frac{C}{n} \sqrt{ \sum_{i=1}^{n} \|\theta\|_\Theta } \\
&= \frac{C}{\sqrt{n}} \|\theta\|_\Theta,
\end{aligned}
$$

where in (a) we applied Lemma A.7, and in (b) we used the fact that $\{\phi(\theta, Z_i)\}_{\theta \in \Theta}$ is 1-subgaussian process w.r.t. $(\Theta, \|\cdot\|_\Theta)$ for every $i \in [n]$. Thus, $\{X_\theta\}_{\theta \in \Theta}$ is $C/\sqrt{n}$-subgaussian

process w.r.t. $(\Theta, \|\cdot\|)$ as per Definition 3.1. Therefore, by Proposition A.10, we have, with probability at least $1 - 4\exp(-t^2)$

$$
\begin{aligned}
\sup_{\theta \in \Theta} \left| \frac{1}{n} \sum_{i=1}^{n} \phi(\theta, Z_i) \right| &\leq \frac{Ct}{\sqrt{n}} + \frac{C}{\sqrt{n}} \int_0^1 \sqrt{\log \mathcal{N}\left(\Theta; \|\cdot\|_\Theta, \varepsilon\right)} d\varepsilon \\
&\leq \frac{Ct}{\sqrt{n}} + \frac{C}{\sqrt{n}} \int_0^1 \sqrt{d \log\left(1 + \frac{2}{\varepsilon}\right)} d\varepsilon \\
&\leq \frac{Ct}{\sqrt{n}} + \frac{C\sqrt{d}}{\sqrt{n}} \\
&= \frac{C}{\sqrt{n}}\left[\sqrt{d} + t\right],
\end{aligned}
\tag{22}
$$

where in second inequality we use [Wai19, Example 5.8]. Hence,

$$
\begin{aligned}
\mathbb{E} \sup_{\theta \in \Theta} \left| \frac{1}{n} \sum_{i=1}^{n} \phi(\theta, Z_i) \right| &= \int_0^\infty \Pr\left[\sup_{\theta \in \Theta} \left| \frac{1}{n} \sum_{i=1}^{n} \phi(\theta, Z_i) \right| > u\right] du \\
&\leq C\sqrt{\frac{d}{n}} + \int_0^\infty \Pr\left[\sup_{\theta \in \Theta} \left| \frac{1}{n} \sum_{i=1}^{n} \phi(\theta, Z_i) \right| > C\sqrt{\frac{d}{n}} + u\right] du \\
&\leq^{(a)} C\sqrt{\frac{d}{n}} + \frac{C}{\sqrt{n}} \int_0^\infty \Pr\left[\sup_{\theta \in \Theta} \left| \frac{1}{n} \sum_{i=1}^{n} \phi(\theta, Z_i) \right| > C\sqrt{\frac{d}{n}} + \frac{Ct}{\sqrt{n}}\right] dt \\
&\leq^{(b)} C\sqrt{\frac{d}{n}} + \frac{C}{\sqrt{n}} \int_0^\infty 4\exp(-t^2) dt \\
&\leq C\sqrt{\frac{d}{n}} + \frac{C'}{\sqrt{n}} \\
&\leq 2(C \vee C')\sqrt{\frac{d}{n}},
\end{aligned}
$$

where $C' > 0$ is some universal constant, (a) follows by a change of variables $u = Ct/\sqrt{n}$, and (b) follows from Equation (22). By Definition 2.3, we can write

$$
\begin{aligned}
\mathrm{Tr}_\kappa(\mathcal{P}; n, \alpha) &= \inf_{\alpha\text{-learner}\,\mathcal{A}_n} \sup_{\mathcal{T} \in \mathfrak{T}_\kappa} \mathbb{E}_{S_n = (Z_1, \ldots, Z_n) \sim \mathcal{D}^{\otimes n}, \hat{\theta} \sim \mathcal{A}_n(S_n)} \left[\frac{1}{n} \sum_{i \in [n]} \phi(\hat{\theta}, Z_i)\right] \\
&\leq \sup_{\mathcal{T} \in \mathfrak{T}_\kappa} \mathbb{E}_{S_n = (Z_1, \ldots, Z_n) \sim \mathcal{D}^{\otimes n}} \left[\sup_{\theta \in \Theta} \frac{1}{n} \sum_{i \in [n]} \phi(\theta, Z_i)\right] \\
&\leq 2(C \vee C')\sqrt{\frac{d}{n}}.
\end{aligned}
$$

By rearranging the terms we obtain the desired result. $\qquad\square$

We now show that Theorem 5.1 implies lower bounds on the sample complexity of learning $\ell_p$-Lipshitz-bounded problems for every $p \in [1, \infty)$. In particular, we show that the problems considered in Theorem 5.1 require many samples to learn (equivalently, we show a lower bound on optimal excess risk $\alpha$).

**Theorem G.2.** *Let $\alpha > 0$, $p \in [1, \infty)$ and $n \in \mathbb{N}$ be arbitrary. Let $\mathcal{P}_{k,p}$ be as in Equation (7), and set $k$ as in Theorem 5.1. Suppose there exist an $\alpha$-learner for $\mathcal{P}_{k,p}$. Then,*

*(i) for $p \in [1, 2]$, we have*

$$
\alpha \geq \frac{c}{\sqrt{n}},
$$

*(ii) for $p \in (2, \infty)$, we have*

$$\alpha \geq c \left[ \frac{1}{n^{1/p}} \wedge \frac{d^{1/2-1/p}}{\sqrt{n}} \right],$$

*for some universal constant $c > 0$.*

*Proof.* First, consider an arbitrary $k \in [d]$. We apply Proposition G.1 to the result of Lemma E.4. We then have the following double inequality

$$c_2 \frac{d^{1-1/p}}{k^{1/2-1/p}n\alpha} \leq \mathrm{Tr}_\kappa(\mathcal{P}_{k,p}; n, \alpha) \leq C\sqrt{\frac{d}{n}}.$$

Solving for $\alpha$ in the above, we have

$$\alpha \geq \frac{c_2}{C} \cdot \frac{(d/k)^{1/2-1/p}}{\sqrt{n}}.$$

First, consider the case $p \in [1, 2]$. Then $k = d$, and we have

$$\alpha \geq \frac{c_2}{C} \cdot \frac{1}{\sqrt{n}},$$

as desired. Now, consider the case $p \in (2, \infty)$. Then $k = (6\alpha)^p d \vee 1$, and we have

$$\alpha \geq \frac{c_2}{C} \cdot \frac{(d/k)^{1/2-1/p}}{\sqrt{n}} = \frac{c_2}{C} \left[ \frac{(1/6\alpha)^{p/2-1}}{\sqrt{n}} \wedge \frac{d^{1/2-1/p}}{\sqrt{n}} \right].$$

Solving for $\alpha$, we have

$$\alpha \geq \left[ \left( \frac{c_2}{C6^{p/2-1}} \right)^{2/p} \cdot \frac{1}{\sqrt{n}} \right] \wedge \left[ \frac{c_2}{C} \cdot \frac{d^{1/2-1/p}}{\sqrt{n}} \right].$$

Note that, since $2/p \leq 1$, we have

$$\left( \frac{c_2}{C6^{p/2-1}} \right)^{2/p} = \frac{1}{6} \left( \frac{6c_2}{C} \right)^{2/p} \geq \frac{1}{6} \left[ 1 \wedge \frac{6c_2}{C} \right].$$

Thus,

$$\alpha \geq \left[ \frac{1}{6} \wedge \frac{c_2}{C} \right] \cdot \left[ \frac{1}{\sqrt{n}} \wedge \frac{d^{1/2-1/p}}{\sqrt{n}} \right],$$

as desired. $\square$

## G.2    Lower bounds for $p = 1$.

Now, consider the case $p = 1$. For $p = 1$, we consider the problem as in Equation (17). We will need the following refinement of Proposition G.1 in a special case when $\Theta$ is a polytope with few vertices. Intuitively, $d$ in the statement Proposition G.1 can be replaced by $\log N$ where $N$ is the number of vertices of $\Theta$.

**Proposition G.3.** *Fix $n \in \mathbb{N}$, $d \in \mathbb{N}$, and $\alpha \in [0, 1]$. Consider an arbitrary SCO problem $\mathcal{P} = (\Theta, \mathcal{Z}, f)$ in $\mathbb{R}^d$, where $\Theta$ is a polytope with $N$ vertices. Let $\mathrm{Tr}_\kappa(\mathcal{P}; n, \alpha)$ be the subgaussian trace value of problem $\mathcal{P}$. Then, we have*

$$\mathrm{Tr}_\kappa(\mathcal{P}; n, \alpha) \cdot \sqrt{n} \leq C \cdot \sqrt{\log N},$$

*for some universal constant $C > 0$.*

*Proof.* Let $(\phi, \mathcal{D})$ be an arbitrary subgaussian tracer. Similarly to the proof of Proposition G.1, consider the process $\{X_\theta\}_{\theta \in \Theta}$ defined as

$$X_\theta := \frac{1}{n} \sum_{i=1}^n \phi(\theta, Z_i),$$

where $S_n = (Z_1, \ldots, Z_n) \sim \mathcal{D}^{\otimes n}$. As in the proof of Proposition G.1, $\{X_\theta\}_{\theta \in \Theta}$ is a $C/\sqrt{n}$-subgaussian process w.r.t. $(\Theta, \|\cdot\|_\Theta)$ for some universal constant $C > 0$.

Let $V$ be the set of vertices of $\Theta$; then $|V| = N$, as per the proposition statement. Since $\phi$ is convex in its first argument, the mapping $\theta \mapsto X_\theta$ is also convex (almost surely). Then,

$$\sup_{\theta \in \Theta} |X_\theta| = \sup_{\theta \in V} |X_\theta|.$$

Therefore, by Proposition A.10, we have, with probability at least $1 - 4\exp(-t^2)$

$$\sup_{\theta \in \Theta} \left| \frac{1}{n} \sum_{i=1}^n \phi(\theta, Z_i) \right| \leq \frac{Ct}{\sqrt{n}} + \frac{C}{\sqrt{n}} \int_0^1 \sqrt{\log \mathcal{N}\left(V; \|\cdot\|_\Theta, \varepsilon\right)} d\varepsilon$$

$$\leq \frac{Ct}{\sqrt{n}} + \frac{C}{\sqrt{n}} \int_0^1 \sqrt{\log N} d\varepsilon$$

$$\leq \frac{Ct}{\sqrt{n}} + \frac{C\sqrt{\log N}}{\sqrt{n}}.$$

Hence,

$$\mathbb{E} \sup_{\theta \in \Theta} \left| \frac{1}{n} \sum_{i=1}^n \phi(\theta, Z_i) \right| = \int_0^\infty \Pr\left[ \sup_{\theta \in \Theta} \left| \frac{1}{n} \sum_{i=1}^n \phi(\theta, Z_i) \right| > u \right] du$$

$$\leq C\sqrt{\frac{\log N}{n}} + \int_0^\infty \Pr\left[ \sup_{\theta \in \Theta} \left| \frac{1}{n} \sum_{i=1}^n \phi(\theta, Z_i) \right| > C\sqrt{\frac{\log N}{n}} + u \right] du$$

$$\leq^{(a)} C\sqrt{\frac{\log N}{n}} + \frac{C}{\sqrt{n}} \int_0^\infty \Pr\left[ \sup_{\theta \in \Theta} \left| \frac{1}{n} \sum_{i=1}^n \phi(\theta, Z_i) \right| > C\sqrt{\frac{\log N}{n}} + \frac{Ct}{\sqrt{n}} \right] dt$$

$$\leq^{(b)} C\sqrt{\frac{\log N}{n}} + \frac{C}{\sqrt{n}} \int_0^\infty 4\exp(-t^2) dt$$

$$\leq C\sqrt{\frac{\log N}{n}} + \frac{C'}{\sqrt{n}}$$

$$\leq 2(C \vee C')\sqrt{\frac{\log N}{n}},$$

where $C' > 0$ is some universal constant, (a) follows by a change of variables $u = Ct/\sqrt{n}$, and (b) follows from Equation (22). By Definition 2.3, we can write

$$\mathrm{Tr}_\kappa(\mathcal{P}; n, \alpha) = \inf_{\alpha\text{-learner}\mathcal{A}_n} \sup_{\mathcal{T} \in \mathfrak{T}_\kappa} \mathbb{E}_{S_n = (Z_1, \ldots, Z_n) \sim \mathcal{D}^{\otimes n}, \hat{\theta} \sim \mathcal{A}_n(S_n)} \left[ \frac{1}{n} \sum_{i \in [n]} \phi(\hat{\theta}, Z_i) \right]$$

$$\leq \sup_{\mathcal{T} \in \mathfrak{T}_\kappa} \mathbb{E}_{S_n = (Z_1, \ldots, Z_n) \sim \mathcal{D}^{\otimes n}} \left[ \sup_{\theta \in \Theta} \frac{1}{n} \sum_{i \in [n]} \phi(\theta, Z_i) \right]$$

$$\leq 2(C \vee C')\sqrt{\frac{\log N}{n}}.$$

By rearranging the terms we obtain the desired result. $\qquad\square$

Then, sample complexity lower bounds for $\ell_1$ geometry follow.

**Theorem G.4.** *Let $\alpha > 0$ and $n \in \mathbb{N}$ be arbitrary. Let $\mathcal{P}_s$ be as in Equation (17) and set $s = d^{0.99}$. Suppose there exists an $\alpha$-learner for $\mathcal{P}_s$. Then, for large enough $d$, we have*

$$\alpha \geq c\sqrt{\frac{\log(d)}{n}},$$

*for some universal constant $c > 0$.*

*Proof.* Lemma E.7 gives the following lower bound on the subgaussian trace value of $\mathcal{P}_s$,

$$\mathrm{Tr}_\kappa(\mathcal{P}_s; n, \alpha) \geq \frac{c\sqrt{s}\log\left(\frac{d}{16(s\vee 14)}\right)}{n\alpha}.$$

At the same time, noting that $\Theta$ is a polytope with vertices given by

$$V = \left\{z \in \left\{0, \pm\frac{1}{s}\right\}^d : \|z\|_0 = s\right\},$$

which has cardinality

$$|V| = \binom{d}{s} \leq \left(\frac{de}{s}\right)^s,$$

we have by Proposition G.3

$$\mathrm{Tr}_\kappa(\mathcal{P}_s; n, \alpha) \leq C\sqrt{\frac{s\log(de/s)}{n}},$$

for some universal $C > 0$. Combining this with the lower bound on subgaussian trace value, we have

$$\frac{c\sqrt{s}\log\left(\frac{d}{16(s\vee 14)}\right)}{n\alpha} \leq C\sqrt{\frac{s\log(de/s)}{n}},$$

which gives

$$\alpha \geq \frac{c}{C}\frac{\log\left(\frac{d}{16(s\vee 14)}\right)}{\sqrt{\log(de/s)n}}.$$

Recall that $s = d^{0.99}$. For large enough $d$, we have $s \vee 14 = s$. Also, note that $\log(d/s) \geq \log(d)/100$. Then, for for large enough $d$, we have

$$\alpha \geq c'\sqrt{\frac{\log(d)}{n}},$$

for some universal $c' > 0$, as desired. $\qquad\square$

# H   Traceability of VC classes (Section 1.1.1)

First, we state the main result.

**Theorem H.1.** *Fix $n \in \mathbb{N}$ and $\xi < 0.1$. Let $\mathcal{H}$ be an arbitrary VC concept class with VC dimension $d_{\mathsf{vc}}$. Then, there exists an optimal algorithm in terms of number of samples such that it is $(\xi/(n\log(n)), m)$-traceable with $m \leq O\left(d_{\mathsf{vc}}\log^2(n)\right)$. Moreover, when $\mathcal{H}$ is the class of thresholds, we have $m \in O(1)$.*

To prove it we use an information-theoretic notion that controls the difficulty of tracing from [SZ20].

**Definition H.2.** Fix $n \in \mathbb{N}$. Let $\mathcal{D}$ be a data distribution, and $\mathcal{A}_n$ be a learning algorithm. For every $n \in \mathbb{N}$, let $\boldsymbol{Z} = (Z_{j,i})_{j\in\{0,1\}, i\in[n]}$ be an array of i.i.d. samples drawn from $\mathcal{D}$, and $\boldsymbol{U} = (U_1, \ldots, U_n) \sim \mathrm{Ber}\left(\frac{1}{2}\right)^{\otimes n}$, where $\boldsymbol{U}$ and $\boldsymbol{Z}$ are independent. Define training set $S_n = (Z_{U_i,i})_{i\in[n]}$. Then, define

$$\mathrm{CMI}_\mathcal{D}(\mathcal{A}_n) := I\left(\mathcal{A}_n(S_n); \boldsymbol{U}\big|\boldsymbol{Z}\right).$$

In the next theorem, we show that the existence of a tracer for a learning algorithm provides a lower bound on the CMI of the algorithm. A similar observation is made in [ADHLR24].

**Lemma H.3.** *Fix $n \in \mathbb{N}$ such that $n \geq 2$ and $\xi < 1/2$. Let $\mathcal{A}_n$ be an arbitrary learning algorithm that is $(\xi/(n\log(n)), m)$-traceable. Then, it holds $\sup_\mathcal{D} \mathrm{CMI}_\mathcal{D}(\mathcal{A}_n) \geq m - 3\xi$.*

The following two results from [SZ20] and [HDMR21] provide upper bounds on the CMI of sample compression schemes. We skip the formal definitions of sample compression schemes and refer the reader to [LW86; MY16; BHMZ20].

**Lemma H.4** (Thm 4.2. [SZ20]). *Let $\mathcal{H}$ be an arbitrary concept class with VC dimension $d_{vc}$. Then, there exists an algorithm such that for every data distribution $\mathcal{D}$, $\mathrm{CMI}_\mathcal{D}(\mathcal{A}_n) \leq O(d_{vc} \log^2(n))$.*

**Lemma H.5** (Thm 3.4. [HDMR21]). *Let $\mathcal{H}$ be the concept class of threshold in $\mathbb{R}$. Then, there exists an algorithm such that for every data distribution $\mathcal{D}$ and $n \geq 2$, $\mathrm{CMI}_\mathcal{D}(\mathcal{A}_n) \leq 2 \log 2$.*

*Proof of Theorem H.1.* The proof is simply by combining Lemma H.4 with Lemma H.3. For the case of the class of thresholds, we use Lemma H.5. □

*Proof of Lemma H.3.* Assume there exists a tracer with recall of $m$ and soundness parameter of $\xi/(n \log(n))$. Let $\mathcal{D}$ denote the distribution used by the tracer (see Definition 2.3). Define the following random set

$$\mathcal{V}^1 = \{i \in [n] : \exists j \in \{0,1\}\ \phi(\hat{\theta}, Z_{j,i}) \geq \lambda \text{ and } \phi(\hat{\theta}, Z_{1-j,i}) < \lambda\}.$$

Also, for every $i \in [n]$, define the following random variable

$$\mathcal{G}_i = \mathbb{1}\{\phi(\hat{\theta}, Z_{\bar{U}_i, i}) < \lambda\}.$$

By the definition of mutual information and $\boldsymbol{U} \perp\!\!\!\perp \boldsymbol{Z}$, we have

$$\mathrm{CMI}_\mathcal{D}(\mathcal{A}_n) = \mathrm{H}(\boldsymbol{U}) - \mathrm{H}(\boldsymbol{U}|\boldsymbol{Z}, \hat{\theta}) \tag{23}$$

$$= n - \mathrm{H}(\boldsymbol{U}|\boldsymbol{Z}, \hat{\theta}). \tag{24}$$

Therefore, to lower bound $\mathrm{CMI}_\mathcal{D}(\mathcal{A}_n)$, we need to upper bound $\mathrm{H}(\boldsymbol{U}|\boldsymbol{Z}, \hat{\theta})$. By the sub-additivity of the entropy, we have

$$\mathrm{H}(\boldsymbol{U}|\boldsymbol{Z}, \hat{\theta}) \leq \sum_{i=1}^n \mathrm{H}(U_i|\boldsymbol{Z}, \hat{\theta}).$$

Then, by monotonicity of the entropy and the chain rule, we have

$$\mathrm{H}(U_i|\boldsymbol{Z}, \hat{\theta}) \leq \mathrm{H}(U_i, \mathcal{G}_i|\boldsymbol{Z}, \hat{\theta})$$

$$\leq \mathrm{H}(\mathcal{G}_i) + \mathrm{H}(U_i|\mathcal{G}_i, \boldsymbol{Z}, \hat{\theta}). \tag{25}$$

In the next step, by the definition of conditional entropy,

$$\mathrm{H}(U_i|\boldsymbol{Z}, \hat{\theta}, \mathcal{G}_i) \leq \mathrm{H}(U_i|\boldsymbol{Z}, \hat{\theta}, \mathcal{G}_i = 1) + \Pr(\mathcal{G}_i = 0).$$

Define the random variable $Y_i = \mathbb{1}(i \in \mathcal{V}^1)$. Notice that $Y_i$ is $(\hat{\theta}, \boldsymbol{Z})$-measurable random variable. Then, using the notations for the disintegrated conditional entropy from [HDMR21], we have

$$\mathrm{H}(U_i|\boldsymbol{Z}, \hat{\theta}, \mathcal{G}_i = 1) = \mathbb{E}\left[\mathrm{H}^{\boldsymbol{Z}, \hat{\theta}, \mathcal{G}_i = 1}(U_i)\right]$$

$$= \mathbb{E}\left[\mathrm{H}^{\boldsymbol{Z}, \hat{\theta}, \mathcal{G}_i = 1}(U_i)\, \mathbb{1}\{Y_i = 1\}\right] + \mathbb{E}\left[\mathrm{H}^{\boldsymbol{Z}, \hat{\theta}, \mathcal{G}_i = 1}(U_i)\, \mathbb{1}\{Y_i = 0\}\right]. \tag{26}$$

The main observation is that under the events $Y_i = 1$ and $\mathcal{G}_i = 1$, $U_i$ is deterministically known from $(\boldsymbol{Z}, \hat{\theta})$. It follows because

$$\{Y_i = 1\} \wedge \{\mathcal{G}_i = 1\} \Leftrightarrow \{\phi(\hat{\theta}, Z_{j,i}) \geq \lambda \text{ and } \phi(\hat{\theta}, Z_{1-j,i}) < \lambda\} \wedge \{\phi(\hat{\theta}, Z_{\bar{U}_i, i}) < \lambda\} \Rightarrow U_i = j.$$

Therefore, since $\mathrm{H}^{\boldsymbol{Z}, \hat{\theta}, \mathcal{G}_i = 1}(U_i) \leq 1$ with probability one, by Equation (26)

$$\mathrm{H}(U_i|\boldsymbol{Z}, \hat{\theta}, \mathcal{G}_i = 1) \leq \Pr(Y_i = 0).$$

Thus, from Equations (25) and (26), we obtain the following upper bound

$$\mathrm{H}(U_i|\boldsymbol{Z}, \hat{\theta}) \leq \Pr(Y_i = 0) + \mathrm{H}_b(\Pr(\mathcal{G}_i = 0)) + \Pr(\mathcal{G}_i = 0),$$

where $H_b(\cdot) : [0,1] \to [0,1]$ is the binary entropy function defined as $H_b(x) = -x\log(x) - (1-x)\log(1-x)$. We can lower bound $\mathrm{CMI}_{\mathcal{D}}(\mathcal{A}_n)$ as follows

$$
\begin{aligned}
\mathrm{CMI}_{\mathcal{D}}(\mathcal{A}_n) &\geq n - \sum_{i=1}^{n} \mathrm{H}(U_i | \boldsymbol{Z}, \hat{\theta}) \\
&\geq n - \sum_{i=1}^{n} \left[ \Pr(Y_i = 0) + H_b(\Pr(\mathcal{G}_i = 0)) + \Pr(\mathcal{G}_i = 0) \right] \\
&= \sum_{i=1}^{n} \Pr(Y_i = 1) - \sum_{i=1}^{n} \left( H_b(\Pr(\mathcal{G}_i = 0)) + \Pr(\mathcal{G}_i = 0) \right),
\end{aligned}
$$

where the last step follows because $n - \sum_{i=1}^{n} \Pr(Y_i = 0) = \sum_{i=1}^{n} \Pr(Y_i = 1)$ as $Y_i$ is an indicator random variable. In the next step, by the definition of soundness from Definition 2.3 and the definition of CMI in Definition H.2, we have

$$
\begin{aligned}
\Pr(\mathcal{G}_i = 0) &= \mathbb{E}\left[ \Pr(\mathcal{G}_i = 0 | \boldsymbol{U}) \right] \\
&= \Pr_{S_n \sim \mathcal{D}^{\otimes n}, Z \sim \mathcal{D}, \hat{\theta} \sim \mathcal{A}_n(S_n)} \left( \phi(\hat{\theta}, Z) \geq \lambda \right) \\
&\leq \xi / (n.\log(n)),
\end{aligned}
$$

Therefore,

$$
\mathrm{CMI}_{\mathcal{D}}(\mathcal{A}_n) \geq \sum_{i=1}^{n} \Pr(Y_i = 1) - \frac{\xi}{\log(n)} - n H_b\left( \frac{\xi}{n\log(n)} \right)
$$

By the recall condition from Definition 2.3 and the definition of CMI in Definition H.2, we have

$$
\begin{aligned}
m &= \mathbb{E}_{S_n = (Z_1, \ldots, Z_n) \sim \mathcal{D}^{\otimes n}, \hat{\theta} \sim \mathcal{A}_n(S_n)} \left[ \left| i \in [n] : \phi(\hat{\theta}, Z_i) \geq \lambda \right| \right] \\
&= \sum_{i=1}^{n} \Pr\left( \phi(\hat{\theta}, Z_i) \geq \lambda \right) \\
&\leq \sum_{i=1}^{n} \Pr\left( \phi(\hat{\theta}, Z_{U_i,i}) \geq \lambda \wedge \mathcal{G}_i \right) + \Pr(\mathcal{G}_i^c) \\
&= \sum_{i=1}^{n} \Pr(Y_i = 1) + \sum_{i=1}^{n} \Pr(\mathcal{G}_i^c) \\
&\leq \sum_{i=1}^{n} \Pr(Y_i = 1) + \frac{\xi}{\log(n)}.
\end{aligned}
$$

We also use the following well-known inequality, $H_b(x) \leq -x\log(x) + x$ for $x \in [0,1]$. As a result, we obtain

$$
\begin{aligned}
\mathrm{CMI}_{\mathcal{D}}(\mathcal{A}_n) &\geq \sum_{i=1}^{n} \Pr(Y_i = 1) - \frac{\xi}{\log(n)} - n H_b\left( \frac{\xi}{n\log(n)} \right) \\
&\geq m - \frac{2\xi}{\log(n)} - n\left( \frac{\xi}{n.\log(n)} - \frac{\xi}{(n.\log(n))} \log\left( \frac{\xi}{(n.\log(n))} \right) \right) \\
&\geq m - \xi\left( \frac{1}{e} + \frac{3}{\log(n)} \right),
\end{aligned}
$$

where the last step follows because $-x\log(x) \leq 1/e$ for $x \in [0,1]$.

$\square$

