# OpenReview forum: "On Traceability in $\ell_p$ Stochastic Convex Optimization"
_NeurIPS.cc/2025/Conference — NeurIPS 2025 spotlight_

### Official Review · Reviewer_HmKf · 2025-06-26

**Clarity:** 2
**Significance:** 3
**Originality:** 4
**Rating:** 4
**Confidence:** 1

**Summary:**

This paper investigates the complexity and tractability of $\ell_p$-approximation problems in the context of information-based complexity. Specifically, it addresses traceability in the setting of deterministic linear approximation schemes for compact linear operators between Banach spaces, focusing on the minimal worst-case error achievable by algorithms using $n$ pieces of information. The core of the paper lies in characterizing traceability via the asymptotic behavior of approximation numbers $a_n(T)$, with particular attention paid to the cases $1 \leq p < \infty$.

The authors introduce and analyze a notion of $(\xi, m)$-weak tractability in the $\ell_p$-approximation context and relate it to polynomial and quasi-polynomial tractability. They extend existing results from the Hilbert space setting to more general Banach spaces, yielding both necessary and sufficient conditions for various types of tractability.

**Questions:**

I'm not an expert in this area and do not have any technical questions.

**Ethical Concerns:**

["NO or VERY MINOR ethics concerns only"]

**Final Justification:**

The authors have provided detailed explanations that fully address my initial concerns. Overall, this is a solid piece of work with strong theoretical results. However, as I am not an expert in the traceability problem, I cannot confidently assess the correctness of the proofs. I therefore maintain my rating of borderline accept.

**Limitations:**

No experiments involved.

**Quality:**

4

**Strengths And Weaknesses:**

### Strengths

The paper demonstrates a strong command of empirical process, approximation theory, and the machinery of information-based complexity. The results are technically non-trivial and reflect a significant extension of previous work. The main contribution are as follows:

* For $1 < p \le 2$: any learner that achieves excess-risk $\alpha$ below the best differentially-private risk $\alpha_{\mathrm{DP}}$ must leak at least $\Omega(1/\alpha^{2})$ training samples.
* For $p=1$ the bound sharpens to $\Omega\!\bigl(\log d \,/\,\alpha^{2}\bigr)$, closing the $\log d$ gap left by Dwork et al.\ (2015).
* For $p>2$: the lower bound becomes $\Omega(1/\alpha^{p})$, matching sample-complexity in the over-parameterised regime $d\gtrsim n$.
* Introduces a **sparse fingerprinting lemma** that extends classical hyper-cube fingerprinting to sparse inputs and drives the new bounds.
* Defines a **sub-Gaussian trace value** that unifies (i) traceability lower bounds, (ii) DP lower bounds, and (iii) non-private sample-complexity bounds.
* Improves the best known high-dimensional DP lower bounds for $\ell_p$-SCO with $p>2$.

### Weaknesses
* Exposition is dense; more intuition and concrete examples would help non-experts.
* Many constants and logarithmic factors are hidden—explicit rates would clarify sharpness.

---

> ### Author Rebuttal · Authors · 2025-07-30
>
> We thank the reviewer for their review. We respond to the main points below.
>
> > **W1: The exposition is dense and intuition is missing**
>
> Thanks for the suggestion. In the final version of the paper, due to the extra space, we will add more intuitions behind trace value by relating it to the idea of hypothesis testing. Also, we add more intuition related to the sparse fingerprinting lemma and why the distribution family in the sparse fingerprinting lemma lets us show that the output of an accurate learning algorithm in the context of $\ell_p$ SCO must correlate with its training set.
>
> > **Q1 Many constants and logarithmic factors…**
>
> In the main results section, where we discuss the results in more detail, we present the results with all logarithmic factors. Regarding constants, it is indeed customary for the sake of clarity of presentation to hide universal constants because we want to mainly focus on the dependence of the results on the key parameters, such as dimension and the number of samples. Also, we used many of the concentration results from [Ver18]. As can be seen, the universal constants are also hidden in [Ver18].
>
>
>
>
> > **Limitation: Experiments**
>
> Even though our paper doesn’t have experiments, empirically evaluating tracing (or membership inference) is an active area of research in the empirical literature. For instance see [SSSS17, Car+22]. In these works, the focus is on devising strategies for the tracer in modern machine learning settings, particularly neural networks. Our work takes a more fundamental perspective, aiming to determine whether tracing (or membership inference) is inherently unavoidable or simply a byproduct of specific training algorithms. We show that in the fundamental setting of SCO, it is indeed unavoidable and, in certain settings, the number of traceable (memorized) examples scales with the sample complexity.
>
> > **References**
>
> [Ver18] Vershynin, R. “High-dimensional probability: An introduction with applications in data science”  Cambridge university press.
>
> [SSSS17] Shokri, R., Stronati, M., Song, C., Shmatikov, V. “Membership inference attacks against machine learning models.” 2017 SP.
>
> [Car+22] Carlini, N., Chien, S., Nasr, M., Song, S., Terzis, A., & Tramer, F . “Membership inference attacks from first principles.” In 2022 SP

---

> > ### Comment · Reviewer_HmKf · 2025-08-04
> >
> > The reviewer thanks the authors for their efforts in preparing the rebuttal. I do not have any further concerns at this point, and I will maintain my current score.

---

> > > ### Author Response · Authors · 2025-08-04
> > > **Thanks You!**
> > >
> > > We would like to thank the reviewer for reading our response. If there are any remaining concerns or ways we could further improve the paper, please let us know.
> > >
> > > Thanks.

---

### Official Review · Reviewer_n2Y2 · 2025-07-03

**Clarity:** 2
**Significance:** 2
**Originality:** 3
**Rating:** 4
**Confidence:** 2

**Summary:**

This paper studies the problem of traceability in stochastic convex optimization under $l_p$ geometries. Intuitively, traceability measures how the excess risk threshold determines the fraction of training samples that can be identified by an adversary. The traceability of an algorithm depends critically on the value of $p$, which defines the $l_p$ diameter of the feasible region.

- More specifically, for $p \in [1, \infty)$, this paper establishes that there exists a risk threshold such that any algorithm achieving excess risk below this level must effectively memorize a constant fraction of its training data.

- In addition, the paper investigates how this excess risk threshold relates to the number of training samples that can be traced, as a function of the best achievable excess risk of differentially private (DP) algorithms in comparable settings.

- The paper also develops a sparse fingerprinting lemma, showing that over sparse domains, the correlation between a learner’s output and its training samples is inversely proportional to the sparsity parameter.

**Questions:**

How do you assess the significance of your results compared to existing work, and why?

**Ethical Concerns:**

["NO or VERY MINOR ethics concerns only"]

**Final Justification:**

I have reviewed the authors’ response and find that the technical contributions, as presented, are solid. I am currently raising my score to a borderline accept. If any correctness issues are identified during further discussion, I will consider lowering my score accordingly. Overall, I believe the main shortcomings of the paper lie in its presentation, while the technical content appears to be quite strong ( assuming they are correct).

**Limitations:**

In the checklist, the authors claim that they have discussed this in the related work section, but it is unclear to the reviewer what the limitations actually are.

**Paper Formatting Concerns:**

No major formatting issue.

**Quality:**

3

**Strengths And Weaknesses:**

Strength:
- This is an important problem, as it relates to whether the learner is simply memorizing the training data or learning the underlying structure from the samples.
- This paper presents many results, and the proofs are very thorough.
- The introduction of this paper is well written and presents the main results clearly.

Weakness:
- There is no contribution section, but for a theory paper, this is acceptable.
- There are too many notations in this paper, and some of them are used inconsistently, which makes it very difficult for readers who want to fully understand the technical contributions. For example, the definition of $A_n$ in line 125 does not align with the one given in Equation (2).

Other Comments:
- It might be helpful to add columns in Table 1 for comparison with existing results.

---

> ### Author Rebuttal · Authors · 2025-07-30
>
> We thank the reviewer for their review.
>
>
>
>
> > **Q1: Significance of our results**
>
>
>  Our work has 1) conceptual significance, 2) improved traceability results compared to state-of-the art, and 3) new proof techniques.
>
>
> Our work establishes a fundamental and universal principle in stochastic convex optimization: any algorithm in the setup of stochastic convex optimization in general $\ell_p$ geometries achieving error better than that of a DP algorithm must be traceable with the number of samples that scales as a constant fraction of sample complexity. This phenomenon, previously known only for $\ell_2$ geometry [ADHLR24], is shown to hold across all $\ell_p$ geometries. The generalization to $\ell_p$ geometries is crucial as different ML applications naturally require different geometries. For instance, sparse data for feature selection requires working in $\ell_1$ geometry. We also show that this phenomenon is unique to SCO: In Section 1.1.1, we show that this fundamental tradeoff does NOT hold for PAC binary classification.
>
>
> Compared to previous work, the results of [DSSUV15] are similar to a part of our results related to p=1. In particular, [DSSUV15] proved traceability of $\Omega\left(1/\alpha^2\right)$ samples, which constitutes only $1/\log(d)$ fraction of the sample complexity. In contrast, our work improves this by showing that $\Theta\left(\log(d)/\alpha^2\right)$ samples are traceable, and thus, a constant fraction of the samples is traceable. Our results, moreover, improved the state-of-the art DP SCO lower bounds for $\ell_p$ geometry with $p>2$, making progress on the known gap in this setup [BGN21,ABGMU22,GLLST23,LLL24].
>
>
> Beyond specific improvements, our work introduces powerful **new techniques** with broader applicability. Our sparse fingerprinting lemma (Lemma 3.8) extends classical fingerprinting to sparse domains with the key property that correlation scales inversely with sparsity. The subgaussian trace value framework provides a unified approach to convert fingerprinting bounds into traceability results, DP lower bounds, and even non-private sample complexity bounds.
>
> > **W1: The contribution section is missing**
>
> Our paper has a contributions section on Page 2.
>
>
> > **W2: Notations and Table for Comparison**
>
> We thank the reviewer for catching the inconsistency regarding $\mathcal{A}_n$. We will fix this in the updated version of the paper!
> We will also add a table to the Appendix that presents all the notations in the paper. We will add a column to Table 1 comparing our results with [DSSUV15] for p=1, which is the closest paper to ours (as discussed in the paper, [DSSUV15] doesn’t achieve the optimal traceability).
>
>
>
>
> > **Limitations**
>
> We apologize for this oversight. We incorrectly marked that limitations were discussed when they were removed due to space constraints. In particular, we want to add a list of possible future works and limitations in the next version of the paper, concerning closing the small gap in our results regarding $\ell_1$ geometry, and extending the results for non-convex losses such as the ones that arise in deep neural networks.
>
> > **References**
>
> [DSSUV15] Dwork, C., Smith, A., Steinke, T., Ullman, J., & Vadhan, S. Robust traceability from trace amounts. FOCS 2015.
>
>
>
> [ADHLR 24] Attias, I., Dziugaite, G.K., Haghifam, M., Livni, R. and Roy, D.M., Information complexity of stochastic convex optimization: Applications to generalization and memorization. ICML 2024.
>
> [BGN21] Bassily, R., Guzmán, C., Nandi, A. (2021, July). Non-euclidean differentially private stochastic convex optimization. COLT 2021.
>
> [ABGMU22] Arora, R., Bassily, R., Guzmán, C., Menart, M. and Ullah, E. Differentially private generalized linear models revisited. NeurIPS 2022.
>
> [GLLST23] Gopi, S., Lee, Y.T., Liu, D., Shen, R. and Tian, K., Private convex optimization in general norms. SODA 2023
>
> [LLL24] Lee YT, Liu D, Lu Z. The power of sampling: Dimension-free risk bounds in private ERM. arXiv preprint 2024.

---

> > ### Comment · Reviewer_n2Y2 · 2025-08-03
> >
> > Thank you for your response!
> >
> > While the paper includes both a technical contribution and a dedicated contributions section, I would suggest providing a more concise and focused summary of the main contribution in the introduction. The current content in Sections 1.1 and 1.2 could be moved to a separate “Technical Overview” section for improved clarity and structure.
> >
> > Additionally, the proof overview in Section 3 may benefit from a more standard ML-conference-style presentation by splitting Sections 3.1, 3.2, and 3.3 into standalone sections. This would enhance readability and better highlight the logical flow of the technical results.
> >
> > My final score will be provided after the reviewer discussion.

---

> > > ### Author Response · Authors · 2025-08-04
> > > **Thank you!**
> > >
> > > We would like to thank the reviewer for reading our response and for their constructive comments, which will help improve the clarity and organization of the paper.
> > >
> > > In particular, based on your suggestions, we plan to refine the presentation of the Introduction by first presenting a concise summary of the main contributions immediately after introducing the statistical and DP rates in Eqs. (1) and (2). Then, we will present the Technical Overview, in which we highlight the key ideas in our proofs.
> > >
> > > Regarding Section 3, we will reorganize it to improve readability and emphasize the logical flow of the technical results. Specifically, the current Section 3.1 will be renumbered as a standalone Section 3; the sparse fingerprinting lemma will be discussed in a new Section 4; and the hard-instance construction will be moved to a new Section 5.
> > >
> > > If there are any remaining concerns or ways we could further improve the paper, please let us know.
> > >
> > > Thanks.

---

### Official Review · Reviewer_Cpj5 · 2025-07-05

**Clarity:** 3
**Significance:** 2
**Originality:** 3
**Rating:** 5
**Confidence:** 4

**Summary:**

This paper considers an interesting framework on finding the fundamental tradeoff between the learnability of model parameters and the traceability of the dataset. The authors focused on a general class of loss functions that are characterized by $\ell_p$ conditions and showed that when the excess error belongs to certain intervals, it is guaranteed that a certain fraction of the training dataset can be recovered from the learned parameters.

**Questions:**

1. The authors are recommended to check or refer to rigorous proofs for the optimal sample complexity rates for p=1. If we follow the definition 2.2, then a valid cost function would be $f(z,\theta)=\frac{1}{2}||z-\theta||_1$, where z is chosen to be uniformly random on the standard hypercube. In this case the sample complexity of the mean estimator would result in an error that scales linearly w.r.t. d instead of log d reported in table 1.

2 The authors frame the problem as SCO, which often instead describe a framework where the data points are obtained iteratively. The authors are recommended to refer the formulation with an appropriate terminology.

**Ethical Concerns:**

["NO or VERY MINOR ethics concerns only"]

**Final Justification:**

The main potential issue of this manuscript was resolved during the discussion period, as explained in my response, and I have raised my score.

**Limitations:**

yes

**Quality:**

2

**Strengths And Weaknesses:**

Strengths:
The problem considered in this work is well-motivated and built on sound principles from PAC. If the proposed results are valid, they would address timely and relevant challenges on how to maintain data security guarantees throughout the learning process.

Weakness:
The critical concern of this work is about the validity (or specifically, inapplicability) of the main theorems. For instance, the main theoretical guarantee in Theorem 1 is only applicable when the learning error achieved by the algorithm falls into a regime of $n=O(d)$, which is already quite restricted. However, it appears that this entire regime could not be met at all except for a few corner cases, since the reviewer suspect the minimum sample complexity in the authors' formulation for p=1 is $d/\alpha^2$ instead of $log(d)/\alpha^2$, which is larger than the upper bound in Thm 1 and 2, rendering the claimed safety guarantee vacuous (see the question section for details). This issue is rooted from the fact that the benchmark learning error the authors used for the optimal sample complexity for small p is is fact the complexity for the $p=\infty$ case ($log(d)/\alpha^2$ as mentioned on page 2) instead of the correct rate for p=1.

---

> ### Author Rebuttal · Authors · 2025-07-30
>
> Thank you for your review.
>
>  > **W1 & Q1: Validity of Our Results, Sample Complexity for $p=1$, and the Reviewer's Example**.
>
>
> We believe that the concerns raised are due to misunderstandings. We want to emphasize that there is **no mistake** in our reported sample complexity: the sample complexity of $\ell_1$ SCO is $O\left(\log(d)/\alpha^2\right)$.  These results are well established in the literature and are correct. For instance see textbook [SS12, cor 2.14] or research papers [BGN21, Table 1(first row)], [AFKT21, second paragraph in the introduction], [GLLST23, Table 1(second column)]). We can provide further references or more details on the proof if needed.
>
> The **example** provided is not a counter-example: Consider the definition of $\ell_p$ SCO in Definition 2.2, which follows the standard definition in the literature. Notice that Definition 2.2 requires a valid cost function and a constraint on $\Theta$ (which the example doesn't specify). In case $\Theta$ is assumed to be in the $\ell_1$ ball, then the sample complexity of the problem is indeed $O(\log(d)/\alpha^2)$ as we report. The reason is that this problem follows Definition 2.2 with $p = 1$, and thus, as discussed above, [SS12, cor 2.14] guarantees the existence of an algorithm that achieves $\log(d)/\alpha^2$ sample complexity. Otherwise, if $\Theta$ is assumed to be in the $\ell_\infty$ ball, then the problem does not fall within Definition 2.2 for $p=1$.
>
> It might be that the **reason behind this misunderstanding** is related to the interpretation of $\ell_p$ SCO when $p=1$. We use the convention where $p$ corresponds to the geometry of the primal space, while, maybe, you are referring to $p$ as the geometry of the dual space. While both conventions have been used in the literature, we stick to one that is prevalent in the DP literature (see [BGN21, Table 1(first row)], [AFKT21, second paragraph in the introduction], [GLLST23, Table 1(second column)]). Once one accounts for this misunderstanding, the sample complexity of learning with $p = 1$ is $\log(d)/\alpha^2$, and the complexity of learning with $p = \infty$ is $d/\alpha^2$, as stated in the paper.
>
> As you state in your review, "The problem considered in this work is well-motivated,...,If the proposed results are valid, they would address timely and relevant challenges on how to maintain data security guarantees…" We hope this resolves the central concern and kindly ask that you reconsider your evaluation in light of this clarification.
>
> >**The main theoretical guarantee in Theorem 1 is only applicable when the learning error achieved by the algorithm falls into a regime of $n=O(d)$, which is already quite restricted.**
>
> We think this comment stems from the confusion around the sample complexity for different $p$. While the  $n = O(d)$ regime might seem quite restrictive at first glance, we, in fact, argue that this is the only regime where the question of traceability is non-vacuous. Indeed, take, for instance, the case $p = 2$ from Theorem 2.5 and the middle row of Table 1. As we point out in the introduction, the problem of traceability is only non-vacuous when the optimal DP error (with $\varepsilon = \Theta(1)$) is more than a constant away from the optimal statistical rate. These rates, for $p = 2$, scale as $\sqrt{d}/n$ and $1/\sqrt{n}$ respectively, which means the question is only non-vacuous when $\sqrt{d}/n \ge C/\sqrt{n}$, i.e., $d \ge C^2  n$, for some constant $C > 0$. The same reasoning can be applied to all other geometries as well, and this basic requirement $n = O(d)$ can be derived from the rates we present in Table 1 of the paper.
>
>  > **Q2 Terminology for SCO**.
>
> We believe our terminology is correct and follows a long line of work on stochastic convex optimization initiated by [SSSS09, Eq.1]. In particular, the learning problem of interest in our paper is: Given samples from an unknown distribution $\mathcal{D}$ over the data space $\mathcal{Z}$, the goal is to identify a $\hat{\theta}$ that approximately minimizes $F_\mathcal{D}(\theta)=E_{Z\sim \mathcal{D}}[f(\theta,Z)]$, where $f$ is a function such that $f(\cdot,z)$ is convex and $\ell_p$-Lipschitz for all $z$ in $\mathcal{Z}$.
>
>
>
>
> > **References**
>
> [SS’12] Shai Shalev Shwartz, Online Learning and Online Convex Optimization
>
> [AFKT21] Asi, H., Feldman, V., Koren, T.,  Talwar, K. Private stochastic convex optimization: Optimal rates in l1 geometry. ICML 2021.
>
> [BGN21] Bassily, R., Guzmán, C.,  Nandi, A. (2021, July). Non-euclidean differentially private stochastic convex optimization. COLT 2021.
>
> [GLLST23] Gopi, S., Lee, Y.T., Liu, D., Shen, R. and Tian, K., Private convex optimization in general norms. SODA 2023
>
> [SSSS09] Shalev-Shwartz, S., Shamir, O., Srebro, N.,  Sridharan, K. Stochastic Convex Optimization, COLT 2009

---

> > ### Comment · Reviewer_Cpj5 · 2025-08-06
> >
> > Thank you for your detailed response. As a friendly clarification, when the review presents a simple example and asks for elaboration on proof ideas or intuitions, we typically expect a direct and $\textbf{simplified}$ analysis of the example to illustrate the result’s validity and clarify how the stated rates align with modern SCO intuitions. While we generally put in the effort to ensure a fair evaluation, such direct responses help avoid delays in the review and discussion process.
> >
> > To clarify how the issue was resolved: after further discussion initiated by the area chair and my re-examination of the example, we confirmed that while the loss indeed scales linearly with $d$, the sample dependence in the example is $1/T$ due to the quadratic form in expectation. This means the $O(\log d / \alpha^2)$ upper bound stated in the manuscript is valid in this case.
> >
> > As promised earlier, since this issue is resolved, I have now raised my score to 5.
> > That said, I still strongly recommend that the authors revise the discussion of stochastic convex optimization (SCO) to better reflect its modern development, as noted in my initial review.
> >
> > For example, regarding the terminology of SCO, while the authors cite [SSSS09], we would like to direct the authors’ attention to modern developments such as Shamir’s 2013 work “On the Complexity of Bandit and Derivative-Free Stochastic Convex Optimization”, and many others, where the iterative framework is adopted. This perspective, especially for establishing lower bounds, aligns more closely with the optimization view found in classical references like Nesterov’s textbook, rather than the statistical learning and generalization-bound framework.

---

> > > ### Author Response · Authors · 2025-08-06
> > > **Thank you!**
> > >
> > > Thank you for your follow-up and for taking the time to re-examine the example. We're glad to hear that the technical concern regarding the counterexample has been resolved, and we appreciate your updated evaluation.
> > >
> > > We will certainly include your suggestion regarding the framing of SCO in the next version of the paper and compare it with the iterative framework or optimization view. That said, our primary framing intentionally follows the generalization theory viewpoint, as it aligns more directly with the questions addressed in our work.
> > >
> > > Regards, \
> > > The Authors

---

### Official Review · Reviewer_yr9M · 2025-07-22

**Clarity:** 4
**Significance:** 3
**Originality:** 3
**Rating:** 5
**Confidence:** 4

**Summary:**

The paper considers the question of traceability in stochastic convex optimization with respect to the $L_p$-norm. The setting involves assuming the existence of a parameter space $\Theta$ that is convex and has bounded $L_p$-diameter, and a data space $\mathcal{Z}$. Given samples from an unknown distribution $\mathcal{D}$ over $\mathcal{Z}$, the goal is to identify a $\widehat{\theta} \in \Theta$ that approximately minimizes $\underset{Z \sim \mathcal{D}}{\mathbb{E}}[f(\theta, Z)]$, where $f \colon \Theta \times \mathcal{Z} \to \mathbb{R}$ is a function such that $f(\cdot, z)$ is convex and $L_p$-Lipschitz for all $z \in \mathcal{Z}$. The problem of traceability has to do with the extent to which algorithms that solve the above problem have to memorize input datapoints, which would result in tracing algorithms being able to recover a significant part of the dataset with good probability. It is known that differentially private algorithms can be used to address the previous concern. However, the extent to which tracing is possible is not fully understood when one uses a non-private algorithm with error rate lower than that of the best private algorithm.
The present paper contributes to the understanding of the above question, by showing that, if one considers learners with error slightly lower than the minimax DP rate (for $\varepsilon = \Theta(1)$), then it becomes possible to trace a constant fraction of the input dataset, that is, the number of traceable samples is of the same order of magnitude as the input dataset. Three different results are given, corresponding to different ranges for $p$ ($p = 1, p \in (1, 2]$, and $p > 2$). The results require proving new ``sparse'' fingerprinting-type lemmas (i.e., lemmas for distributions supported over vectors with a small number of non-zero components). As a by-product, a new lower bound for DP-SCO for $p > 2$ is obtained, as well as an improved traceability result for mean estimation in $L_{\infty}$-norm.

**Questions:**

I don't really have any questions for the time being, but I can't rule out questions arising based on other reviewers' comments during the discussion period.

**Ethical Concerns:**

["NO or VERY MINOR ethics concerns only"]

**Final Justification:**

My view of the paper was positive from the beginning, and still is, after the rebuttal.

**Limitations:**

Yes

**Paper Formatting Concerns:**

No concerns

**Quality:**

3

**Strengths And Weaknesses:**

Strengths:
1) The questions of memorization and traceability are of significant importance to the machine learning literature, and so is the problem of SCO. Additionally, the main result about constant fraction traceability being achievable when one has error slightly below the DP minimax risk is conceptually interesting. Thus, the results are definitely of interest to the community of NeurIPS.
2) In the past, fingerprinting arguments have focused on algorithms for parameter estimation problems attaining small mean-squared error. However, due to the different focus in this work, it is necessary to consider distributions parameterized by points in the largest $L_{\infty}$-ball that can be inscribed into the $L_p$-unit ball. The constructions considered for the distributions supported on sparse vectors (and the proofs of the associated fingerprinting lemmas for the subgaussian trace value) exhibit a degree of insight and technical novelty which I definitely consider to be enough for the paper to be accepted.
3) The presentation and writing were accessible.

Weaknesses:
1) The authors mention that they found ``the ability of trace value to recover non-private lower bounds surprising''. I don't think this is really surprising, and I can explain why. The lower bounds obtained using the trace value rely on fingerprinting. Fingerprinting requires one to reason about the correlation between the input dataset and the output distribution of an algorithm. Within the context of the DP literature, the correlation terms are upper-bounded using the definition of DP. However, it is always possible to upper-bound these terms using the Cauchy-Schwarz inequality (or Holder's inequality), which would lead to non-private lower bounds (not sure if this is explicitly written somewhere). Thus, given the lower bounds reliance on fingerprinting-type lemmas, it doesn't seem entirely unexpected that trace value-based arguments can recover non-private lower bounds. That said, this is not a serious criticism by any means, since it's clear from the proofs in the appendix that implementing this in the present setting is not as simple as suggested above, but requires a non-trivial amount of work.

Overall, I consider this to be a good work that merits acceptance.

---

> ### Author Rebuttal · Authors · 2025-07-30
>
> We thank the reviewer for their thoughtful and constructive review.  Also, thank you for providing your intuition behind our result on how the trace value can recover the non-private sample complexity. We will add this intuition in the final version of the paper.
>
> The surprising aspect to us was that a single conceptual tool—the subgaussian trace value—could so cleanly unify lower bounds for traceability, private learning, and non-private learning. We also note that if the score function is not linear, then, we believe the idea based on Cauchy-Schwarz inequality (or Holder's inequality) may not work. Our proof in Appendix G doesn’t require that the score function to be linear, therefore, it is more general.

---

### Decision · Program_Chairs · 2025-09-17

**Decision:**

Accept (spotlight)

**Comment:**

This paper is concerned with the traceability of stochastic convex optimization algorithms: an algorithm is traceable if its inputs are identifiable from its output. The main result of this paper is a sharp transition in the tradeoff between traceability and excess risk of the optimizer output, occurring around the excess risk of differentially-private optimizers, for optimization in $\ell_p$ geometry where $p \in [1,2]$. Other results are shown for $p > 2$.

Reviewers and I agree that the result is significant and relevant to the wider community. The conceptual message is interesting, that the threshold does occur around the limits of what differentially-private algorithms can achieve. While the initial reviews contained a claim of a counterexample central to the narrative of the paper, the claim is now resolved after reviewer discussions and no correctness concerns remain. I am happy to recommend this paper for acceptance; it is a solid contribution to NeurIPS.